# GOAL REACHING WITH EIKONAL-CONSTRAINED HIERARCHICAL QUASIMETRIC REINFORCEMENT LEARNING

**Vittorio Giammarino, Ahmed H. Qureshi**
Purdue University
{vgiammar,ahqureshi}@purdue.edu

## ABSTRACT

Goal-Conditioned Reinforcement Learning (GCRL) mitigates the difficulty of reward design by framing tasks as goal reaching rather than maximizing hand-crafted reward signals. In this setting, the optimal goal-conditioned value function naturally forms a quasimetric, motivating Quasimetric RL (QRL), which constrains value learning to quasimetric mappings and enforces local consistency through discrete, trajectory-based constraints. We propose *Eikonal-Constrained Quasimetric RL (Eik-QRL)*, a continuous-time reformulation of QRL based on the Eikonal Partial Differential Equation (PDE). This PDE-based structure makes Eik-QRL trajectory-free, requiring only sampled states and goals, while improving out-of-distribution generalization. We provide theoretical guarantees for Eik-QRL and identify limitations that arise under complex dynamics. To address these challenges, we introduce *Eik-Hierarchical QRL (Eik-HiQRL)*, which integrates Eik-QRL into a hierarchical decomposition. Empirically, Eik-HiQRL achieves state-of-the-art performance in offline goal-conditioned navigation and yields consistent gains over QRL in manipulation tasks, matching temporal-difference methods.

## 1 INTRODUCTION

In recent years, Reinforcement Learning (RL) has emerged as a powerful paradigm for solving complex decision-making tasks from experience, achieving success in domains such as game playing (Mnih et al., 2015; Silver et al., 2017), robotics (Tang et al., 2025), autonomous driving (Kiran et al., 2021; Sabouni et al., 2024; Ahmad et al., 2025), and industrial control (Mirhoseini et al., 2020). Despite these achievements, the aforementioned solutions fundamentally rely on human-designed rewards, a process that is notoriously difficult and time consuming. Reward engineering introduces hand-crafted structures, additional hyperparameters, and assumptions about task solutions, which ultimately undermines RL's promise to transcend human design and ability.

Goal-Conditioned RL (GCRL) provides an elegant way to address this challenge by framing problems in terms of reaching arbitrary goals (Schaul et al., 2015; Andrychowicz et al., 2017). By conditioning policies on goals, GCRL offers a scalable route toward flexible, goal-directed behavior without the need for manually designed reward functions. Beyond this practical advantage, recent work has uncovered a fundamental geometric property of GCRL: the optimal goal-conditioned value function $V^*(s, g)$ corresponds to the length of the shortest feasible path from a state $s$ to a goal $g$, and therefore naturally defines a *quasimetric* (Sontag, 1995; Liu et al., 2023).

This perspective underpins *Quasimetric RL (QRL)* (Wang et al., 2023), a recently proposed GCRL algorithm that restricts value functions to quasimetric mappings, reducing the hypothesis space from arbitrary functions to a structured subset aligned with goal-reaching tasks. In this framework, GCRL is recast as an optimization problem over quasimetrics, providing an alternative to the classical Temporal-Difference (TD) recursion (Sutton & Barto, 2018). To accurately represent the optimal value function, Wang et al. (2023) show that the formulated optimization problem must preserve *local distances*, i.e., the costs of transitions between successive states, while simultaneously *pushing apart* distant states. Consequently, QRL is framed as the maximization of a quasimetric function where local consistency is enforced through trajectory-based constraints.

Building on these foundations, we propose a novel formulation of QRL that enforces local constraints directly in continuous time rather than through discrete transitions. We argue that for many interesting use cases the discrete-time perspective offers no fundamental benefit beyond practical convenience, as the continuous-time formulation naturally leads to Partial Differential Equations (PDEs). Historically, the difficulties of solving PDEs has limited their role in RL, but recent advances in Physics-Informed Neural Networks (PINNs) (Raissi et al., 2019) have begun to overcome this barrier. By exploiting automatic differentiation, PINNs embed PDE constraints directly into the training objective, enabling neural networks to approximate PDE solutions.

This progress opens the door to a continuous-time perspective on QRL that brings new structural advantages. Most notably, our PDE formulation becomes *trajectory-free*: rather than relying on short rollouts, it only requires sampling random states and goals from the feasible space. Furthermore, beyond enforcing local consistency, the PDE acts as an implicit regularizer on the value function (Giammarino et al., 2025), improving both learning stability and out-of-distribution estimation accuracy.

**Contributions**   We summarize the contributions of our work as follows:

1. Our main contribution is the introduction of a novel PDE-constrained formulation of QRL that derives continuous local constraints from the Eikonal PDE (Noack & Clark, 2017), leading to Eik-QRL. We provide theoretical guarantees for Eik-QRL and highlight inherent limitations in complex dynamical settings (cf. Section 4).

2. Building on Eik-QRL, we propose Eik-Hierarchical QRL (Eik-HiQRL), a hierarchical algorithm that mitigates these limitations while preserving the strengths of Eik-QRL (cf. Section 5). Eik-HiQRL achieves state-of-the-art (SOTA) performance on the OGbench navigation suite (Park et al., 2024a) and delivers consistent improvements over QRL in manipulation tasks, while also matching the performance of TD-based algorithms.

3. Other minor contributions include an additional theoretical analysis supporting the hierarchical design of Eik-HiQRL (cf. Appendix B) and an experimental evaluation protocol that accounts not only for goal-reaching success but also for *collision avoidance*, an aspect often overlooked in the RL literature.

## 2   RELATED WORK

**Goal-Conditioned RL**   GCRL was introduced to address two longstanding challenges in classical RL (Sutton & Barto, 2018). The first is the difficulty of designing reward functions, a time consuming process, prone to unintended side effects. Reward shaping (Ng et al., 1999) has been proposed to ease this burden, but it remains subject to reward hacking, where agents exploit loopholes to maximize rewards without achieving the intended behavior (Amodei et al., 2016; Everitt et al., 2017; 2021; Pan et al., 2022; Di Langosco et al., 2022). Sparse rewards offer an alternative to hand-crafted shaping, yet they introduce complementary challenges due to the weakness of the learning signal. To this end, GCRL provides a natural way to cope with this difficulty by framing tasks in terms of goal reaching, which enables learning directly from sparse success indicators (Kaelbling, 1993; Schaul et al., 2015), and has motivated specialized algorithms to improve efficiency under sparse feedback (Andrychowicz et al., 2017). A second limitation of classical RL is its focus on single-task settings, which restricts generalization across objectives. By conditioning policies on goals, GCRL extends naturally to multi-goal learning and supports more general-purpose behaviors. Recent advances in this direction include Contrastive RL (Eysenbach et al., 2022), state-occupancy matching (Ma et al., 2022), and QRL (Wang et al., 2023).

**Offline GCRL**   Offline RL trains policies from pre-collected datasets without further interaction with the environment (Levine et al., 2020). This paradigm is appealing in domains where online data collection is expensive, risky, or impractical, but it suffers from distributional shift and extrapolation error when policies select actions outside the dataset support. Extending goal conditioning to this setting has given rise to *Offline GCRL*, which inherits the challenges of both paradigms: agents must generalize to unseen state-goal pairs while learning entirely from fixed data and without corrective feedback. These compounded difficulties have motivated a range of specialized methods (Chebotar et al., 2021; Yang et al., 2022; 2023; Mezghani et al., 2023; Sikchi et al., 2023; Park et al., 2024b;a) aimed at improving stability and generalization. Our work contributes to this line by introducing PDE-based constraints that improve generalization to out-of-distribution state-goal pairs, with particular benefits in large environments and in scenarios that require trajectory stitching.

**Quasimetrics in GCRL**   A key advance in GCRL is recognizing that the optimal goal-conditioned value function corresponds to the shortest feasible path between a state and a goal, and therefore naturally defines a quasimetric (Sontag, 1995; Liu et al., 2023). As mentioned above, QRL (Wang et al., 2023) builds on this observation by restricting value functions to quasimetric mappings, aligning the hypothesis space with shortest-path goal reaching. This perspective connects to earlier work on quasimetrics in representation learning (Sontag, 1995; Pitis et al.; Durugkar et al., 2021) and has inspired the development of differentiable architectures (Wang & Isola; Liu et al., 2023) that provide universal approximation guarantees for quasimetric parameterized functions. Our work establishes PDE-constrained QRL as a new direction, reformulating value learning in continuous time and revealing structural properties that extend beyond existing formulations.

**Continuous-time Optimal Control and PINNs**   The continuous-time view of optimal control is traditionally formulated through PDEs, most notably the Hamilton-Jacobi-Bellman (HJB) equation (Munos, 1997; Munos & Bourgine, 1997). Although these formulations capture both local and global properties of value functions, they are notoriously difficult to solve in practice, which has long limited their impact in RL. More recently, PINNs (Rudy et al., 2017; Raissi et al., 2019; Bansal & Tomlin, 2021) have leveraged automatic differentiation to embed PDE constraints directly into neural training, enabling progress in areas such as system identification and scientific computing. This paradigm has also been applied to RL: Shilova et al. (2023) used PINNs to approximate HJB solutions in continuous-time RL, though their results were limited to simple dynamics and low-dimensional state spaces, while other works have employed PDE-inspired regularizers to improve value estimation accuracy in Offline RL (Lien et al., 2024) and Offline GCRL (Giammarino et al., 2025). In contrast, our approach incorporates PDE constraints into the QRL framework, yielding a continuous-time formulation that offers structural benefits not explored in earlier work.

**Hierarchy in RL**   Temporal abstraction is a long-standing idea in RL, where policies are decomposed into high- and low-level components to improve exploration and credit assignment (Sutton et al., 1999b; Vezhnevets et al., 2017; Nachum et al., 2018; Giammarino & Paschalidis, 2021). Recent work has shown that this structure is particularly effective in GCRL, where subgoal decomposition mitigates the challenges of long horizons (Park et al., 2024b). In our framework, hierarchy is essential, as it both mitigates the practical limitations of Eik-QRL and amplifies its strengths, thereby further improving value learning and overall performance in long-horizon tasks.

## 3   PRELIMINARIES

**Goal-Conditioned RL**   We consider a finite-horizon discounted Markov Decision Process (MDP) defined by $(\mathcal{S}, \mathcal{G}, \mathcal{A}, \mathcal{T}, \mathcal{R}, \mathcal{P}_g, \rho_0, \gamma)$, where $\mathcal{S}$ is the state space, $\mathcal{G}$ the goal space, $\mathcal{A}$ the action space, $\mathcal{T} : \mathcal{S} \times \mathcal{A} \rightarrow \mathcal{P}(\mathcal{S})$ the transition map with $\mathcal{P}(\mathcal{S})$ the set of distributions over $\mathcal{S}$, and $\mathcal{R} : \mathcal{S} \times \mathcal{G} \rightarrow \mathbb{R}$ a goal-conditioned reward function such that $\mathcal{R}(s, g) = -1$ when $s \neq g$ and $\mathcal{R}(s, g) = 0$ otherwise. Furthermore, $\mathcal{P}_g \in \mathcal{P}(\mathcal{G})$ is the goal distribution, $\rho_0 \in \mathcal{P}(\mathcal{S})$ is the initial state distribution, and $\gamma \in (0, 1]$ is the discount factor. Unless specified, we assume $\mathcal{G} \equiv \mathcal{S}$ and $g \sim \mathcal{P}_g$ at the start of each episode. The agent aims to maximize $J(\pi) = \mathbb{E}_{\tau_\pi(g)}[\sum_{t=0}^T \gamma^t \mathcal{R}(s_t, g)]$, where $\tau_\pi(g) = (g, s_0, a_0, s_1, a_1, \ldots, s_T)$ are trajectories generated by the decision policy $\pi : \mathcal{S} \times \mathcal{G} \rightarrow \mathcal{P}(\mathcal{A})$. The value function induced by $\pi$ is defined as $V^\pi(s, g) = \mathbb{E}_{\tau_\pi(g)}[\sum_{t=0}^T \gamma^t \mathcal{R}(s_t, g) \mid S_0 = s, G = g]$. In the offline setting, the agent must optimize $J(\pi)$ using only a fixed dataset $\mathcal{D}$ of trajectories $\tau = (s_0, a_0, s_1, s_2, \ldots, s_T)$. Finally, we denote parameterized functions as $\pi_{\boldsymbol{\theta}}$ where $\boldsymbol{\theta} \in \Theta \subset \mathbb{R}^k$.

**Quasimetric RL**   Given $\mathcal{X} \subset \mathbb{R}^d$ a compact subset with non-empty interior, a quasimetric function is defined as $d : \mathcal{X} \times \mathcal{X} \rightarrow \mathbb{R}_{\geq 0}$ such that $\forall x_1, x_2, x_3$ the triangle inequality holds:

$$d(x_1, x_3) \leq d(x_1, x_2) + d(x_2, x_3);$$

and $\forall x, d(x, x) = 0$. We denote by $\mathcal{Q}\mathrm{met}(\mathcal{X})$ the space of quasimetrics over $\mathcal{X} \times \mathcal{X}$. QRL is formulated on the observation that, in markovian environments, the optimal goal-conditioned value function, $V^*(s, g) = -d^*(s, g)$, belongs to the space of quasimetrics. Hence, $-V^* \in \mathcal{Q}\mathrm{met}(\mathcal{S})$. As a consequence, GCRL can be re-framed as an optimization problem over $\mathcal{Q}\mathrm{met}(\mathcal{S})$. Wang et al. (2023) argue that, in order to capture both local and global relationships between states, this optimization should preserve *local distances* (i.e., transition costs) between nearby states while *maximally spreading out* distant ones. This motivates the following formulation:

$$\max_{\boldsymbol{\theta}} \underbrace{\mathbb{E}_{s \sim \mathcal{P}_{\mathrm{state}}, g \sim \mathcal{P}_{\mathrm{goal}}}[\zeta(d_{\boldsymbol{\theta}}(s, g))]}_{\text{Global Relationships (GRs)}} \quad \text{s.t.} \quad \underbrace{\mathbb{E}_{(s, a, s', \mathrm{cost}) \sim \mathcal{D}}[\max(d_{\boldsymbol{\theta}}(s, s') - \mathrm{cost}, 0)^2]}_{\text{Local Relationships (LRs)}} \leq \epsilon^2, \quad (1)$$

where: $\text{cost} = -\mathcal{R}(s, s')$, $\mathcal{P}_{\text{state}}$ and $\mathcal{P}_{\text{goal}}$ denote respectively the marginal state and goal distributions induced by the trajectories dataset, $\zeta$ is a monotonically increasing transformation that dampens large distances $d_{\boldsymbol{\theta}}(s, g)$, and $d_{\boldsymbol{\theta}}$ is a parameterized quasimetric model, i.e., $d_{\boldsymbol{\theta}} \in \mathcal{Q}\text{met}(\mathcal{S})$. A diagram for QRL is provided in Fig. 1. The optimization problem in (1) guarantees optimal value recovery under the assumption that quasimetric models are universal function approximators, i.e., capable of representing any function in $\mathcal{Q}\text{met}(\mathcal{S})$ up to an $\epsilon > 0$.

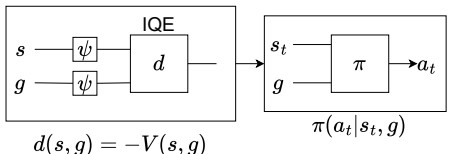

$$d(s, g) = -V(s, g)$$

Figure 1: Diagrams for QRL (Wang et al., 2020). QRL consists of a quasimetric value function parameterized by an Interval Quasimetric Embedding (IQE) model (Wang & Isola) and an actor $\pi$.

**Continuous-time Optimal Control** When in continuous-time, we consider a dynamical system defined by $\dot{s}(t) = f(s(t), a(t))$, with $t \geq 0$ and where $s(t)$ denotes the state of the system at time $t$, $\dot{s}(t)$ its time derivative and $a(t)$ the control action. Given the initial conditions $s(0) = s_0$, $a(0) = a_0$ and a goal $s(T) = g$, the undiscounted optimal control problem seeks to minimize the cumulative cost $J = \int_0^T c(s(t), a(t))dt$ where $c(s(t), a(t))$ is the instantaneous cost function. The optimal value function is defined as $d^*(s, g) = \inf_{a(\cdot)} \int_0^T c(s(t), a(t))dt$, and, for sufficiently small $\Delta t$, it satisfies the principle of optimality

$$d^*(s, g) = \inf_{a \in \mathcal{A}} [c(s, a)\Delta t + d^*(s(t + \Delta t), g)]. \tag{2}$$

Here we intentionally reuse the notation $d^*(s, g)$, as in the quasimetric setting above, and will later establish a direct connection between the two. Under standard regularity conditions (Assumptions 4.1, 4.2, 4.3 and 4.4), $d^*(s(t + \Delta t), g)$ in (2) can be approximated by its first-order Taylor expansion (Giammarino et al., 2025), yielding

$$\inf_{a \in \mathcal{A}} [c(s, a) + \nabla_s d^*(s, g)^{\mathsf{T}} f(s, a)] = 0. \tag{3}$$

Eq. (3) is the HJB PDE for the undiscounted problem defined above and highlights the relationship between the dynamics, cost, and value function in continuous time.

## 4 EIKONAL-CONSTRAINED QUASIMETRIC RL

In this section, we introduce and discuss our Eik-QRL algorithm. We begin by stating the assumptions underlying our formulation, then derive continuous-time variants of quasimetric value learning, including Eik-QRL as a novel perspective on QRL. We conclude with a theoretical analysis that highlights Eik-QRL structural properties and limitations motivating our hierarchical extension.

**Assumptions** We begin by stating the assumptions underlying our formulation.

**Assumption 4.1** (Compact state and action spaces). Let the state space $\mathcal{S}$ and the action space $\mathcal{A}$ be compact, and let $\mathcal{K} \subseteq \mathcal{S}$ denote the feasible (obstacle-free) subset.

**Assumption 4.2** (Lipschitz dynamics). The transition map $\mathcal{T} : \mathcal{S} \times \mathcal{A} \to \mathcal{S}$ is deterministic and Lipschitz continuous in $s$. That is, there exists a constant $L_{\mathcal{T}} > 0$ such that for every $s, s' \in \mathcal{K}$ and $a \in \mathcal{A}$: $\|\mathcal{T}(s, a) - \mathcal{T}(s', a)\| \leq L_{\mathcal{T}} \|s - s'\|$. Equivalently, in the continuous-time limit $\Delta t \to 0$, we have $\mathcal{T}(s, a) = s + f(s, a)\Delta t + o(\Delta t)$, where $f(\cdot, a)$ is Lipschitz continuous in $s$ (uniformly in $a$).

**Assumption 4.3** (Cost regularity). We consider a continuous, bounded, nonnegative running cost $c : \mathcal{S} \times \mathcal{G} \to \mathbb{R}_{\geq 0}$, where $c(s, g)$ denotes the instantaneous cost incurred when the agent visits state $s \in \mathcal{S}$ while pursuing goal $g \in \mathcal{G}$. Specifically, we assume a unit running cost away from the goal, that is, $c(s, g) = 0$ if $s = g$ and $c(s, g) = 1$ if $s \neq g$, for all $g \in \mathcal{G}$. The value function $d(s, g)$ is obtained by accumulating this cost along the trajectory until the goal is reached.

**Assumption 4.4** (Value function regularity). For each fixed goal $g \in \mathcal{K}$, the optimal value function $d^*(\cdot, g)$ exists, is finite on $\mathcal{K}$, and is locally Lipschitz.

These assumptions are standard optimal-control regularity conditions that ensure a sound HJB formulation (Munos & Bourgine, 1997; Munos, 1997; Kim et al., 2021). Assumption 4.1 guarantees that the minimization over actions in Eq. (3) is well posed, i.e., the $\arg\min$ in the HJB operator exists.

Assumption 4.2 ensures that the system dynamics admit unique and well-defined trajectories since Lipschitz continuity of $f(s, a)$ guarantees existence and uniqueness of ODE solutions (Coddington & Levinson, 1955). This in turn makes the value function and the associated HJB PDE well posed, and provides stability for local expansions and numerical approximations such as $\mathcal{T}(s, a) = s + f(s, a)\Delta t + o(\Delta t)$. Determinism is imposed mainly for analytical convenience and is consistent with prior work (Kumar et al., 2019; Wang et al., 2023); importantly, our method still applies in stochastic settings, as illustrated by our `teleport` experiments in Section 6. Finally, Assumption 4.3 ensures finite path costs, and Assumption 4.4 permits point-wise Taylor expansions of value functions.

**Formulation**    Let the dynamics be $\mathcal{T}(s, a) \equiv s' = s + f(s, a)\,\Delta t + o(\Delta t)$ and the instantaneous cost be $c(s, g)$, such that the one-step cost is equal to $c(s, g)\,\Delta t + o(\Delta t)$. Note that, following GCRL, the cost does not depend on actions. QRL in Eq. (1) imposes local consistency through the constraint

$$\mathbb{E}_{(s,a,s')\sim\mathcal{D}}\Big[\max\left(d_{\boldsymbol{\theta}}(s, s') - 1,\, 0\right)^2\Big] \;\leq\; \epsilon^2, \tag{4}$$

where, compared to Eq. (1), we set the cost equal to $1$ aligned with the GCRL framework. This constraint penalizes violations of the inequality $d(s, s') \leq c(s, g)\,\Delta t + o(\Delta t)$ along observed transitions $(s, s')$. We now establish a connection between Eq. (4) and the HJB PDE in (3).

Consider the triangle inequality $d(s, g) \leq d(s, s') + d(s', g)$, by replacing $d(s, s') \leq c(s, g)\,\Delta t + o(\Delta t)$, we get

$$d(s, g) \;\leq\; c(s, g)\,\Delta t \;+\; d(s', g) \;+\; o(\Delta t). \tag{5}$$

Assuming $d(\cdot, g)$ is differentiable at $s$, the first-order Taylor expansion of $d(s', g)$ becomes $d(s', g) = d(s, g) + \nabla_s d(s, g)^\top f(s, a)\,\Delta t + o(\Delta t)$. Replacing $d(s', g)$ in Eq. (5) yields $d(s, g) \leq c(s, g)\,\Delta t + d(s, g) + \nabla_s d(s, g)^\top f(s, a)\,\Delta t + o(\Delta t)$. By subtracting $d(s, g)$ from both sides, dividing by $\Delta t$ and taking the limit $\Delta t \to 0$, we get

$$0 \;\leq\; c(s, g) \;+\; \nabla_s d(s, g)^\top f(s, a), \tag{6}$$

which is the static HJB *inequality* and holds true for every $a \in \mathcal{A}$ and $d(\cdot)$. Under the assumptions stated above, by minimizing over $a$ and assuming $d(\cdot) = d^*(\cdot)$, Eq. (6) tightens to the HJB PDE

$$0 \;=\; c(s, g) + \nabla_s d^*(s, g)^\top f(s, a^*),$$

where $a^* = \arg\min_a \nabla_s d^*(s, g)^\top f(s, a)$.

In the hypothetical case that $a^*$ were available, it becomes natural to view the HJB residual in (6) as a surrogate for the discrete local constraint in (4). In practice, if the goal $g$ is chosen as a state visited along the same trajectory as the transition $(s, s')$ (Andrychowicz et al., 2017), one can reasonably approximate $f(s, a^*) \approx s' - s$, which leads to the following HJB-QRL formulation:

$$\max_{\boldsymbol{\theta}} \underbrace{\mathbb{E}_{s,g}\big[\zeta(d_{\boldsymbol{\theta}}(s, g))\big]}_{\text{Global Relationships (GRs)}} \quad \text{s.t.} \quad \underbrace{\mathbb{E}_{(s,s')\sim\mathcal{D},\, g\sim\mathcal{P}_{\text{goal}}}\Big[\big(\nabla_s d_{\boldsymbol{\theta}}(s, g)^\top (s' - s) + 1\big)^2\Big]}_{\text{HJB-Local Relationships (HJB-LRs)}} \;\leq\; \epsilon^2. \tag{7}$$

This formulation offers a PINN-style perspective on goal-conditioned value learning, which extends beyond purely additive regularizers explored in prior work (Lien et al., 2024; Giammarino et al., 2025). However, the problem in (7) becomes difficult to optimize in practice, as the inner product $\nabla_s d_{\boldsymbol{\theta}}(s, g)^\top (s' - s)$ is often ill-conditioned in high-dimensional state spaces, a challenge also noted in many PINN-based approaches to HJB (Shilova et al., 2023; Bansal & Tomlin, 2021). Furthermore, it retains dependence on transition tuples $(s, s')$, reflecting QRL's reliance on trajectories.

As a result, in order to tackle the aforementioned practical challenges, we propose to refine HJB-QRL by imposing explicit assumptions on the system dynamics $f(s, a)$. Specifically, we simplify the HJB PDE to an Eikonal PDE (Noack & Clark, 2017) by leveraging unit-speed, isotropic dynamics $f(s, a) = a$ with $\|a\| \leq 1$. In this setting, it follows that $a^* = -\nabla_s d(s, g)/\|\nabla_s d(s, g)\|$, reducing the local constraint to a unit-slope condition. This yields the following Eikonal-constrained QRL (Eik-QRL) formulation:

$$\max_{\boldsymbol{\theta}} \underbrace{\mathbb{E}_{s,g}\big[\zeta(d_{\boldsymbol{\theta}}(s, g))\big]}_{\text{Global Relationships (GRs)}} \quad \text{s.t.} \quad \underbrace{\mathbb{E}_{s\sim\mathcal{P}_{\text{state}},\, g\sim\mathcal{P}_{\text{goal}}}\Big[\big(\|\nabla_s d_{\boldsymbol{\theta}}(s, g)\| - 1\big)^2\Big]}_{\text{Eikonal-Local Relationships (Eik-LRs)}} \;\leq\; \epsilon^2. \tag{8}$$

*Remark* 4.5 (Benefits). Relative to QRL and HJB-QRL, Eik-QRL becomes *trajectory-free*: it requires only i.i.d. samples of states and goals $(s, g)$ rather than full rollouts or transition pairs $(s, s')$. This property is particularly appealing in applications such as $(i)$ navigation with maps, where $s$ and $g$ can be drawn uniformly from free poses of an occupancy map; $(ii)$ robotic manipulation, where target object or end-effector poses are sampled within a collision-free workspace; $(iii)$ autonomous driving and motion planning, where waypoints are sampled along lane centerlines or roadgraph nodes; and $(iv)$ industrial or process control, where operating setpoints are sampled within a certified envelope. In addition, Eik-QRL *improves state-space coverage*: for $\mathcal{S} \subset \mathbb{R}^k$, each sampled pair $(s, g)$ contributes a full gradient vector $\nabla_s d_{\boldsymbol{\theta}}(s, g) \in \mathbb{R}^k$, thereby coupling all coordinate directions at $s$. Aggregating such gradients across diverse pairs has a regularizing effect, promoting global consistency, whereas QRL's constraints act only along observed transitions in $\mathcal{D}$. Our experiments confirm these advantages, particularly in large environments and in stitched-data regimes.

*Remark* 4.6 (Limitations). A natural concern arises from assuming isotropic dynamics $f(s, a) = a$ with $\|a\| \leq 1$, which may appear restrictive. We emphasize that this choice primarily simplifies the problem numerically, and empirical evidence show that Eik-QRL outperforms HJB-QRL and QRL even under complex dynamics. We acknowledge, however, that this assumption commits us to a class of solutions that might not work optimally for all MDPs and, consequently, envision future work which extend beyond isotropic dynamics, while preserving similar numerical advantages. More broadly, we view both Eik-QRL and HJB-QRL as providing a hybrid regime in which simplified dynamical models impose explicit value-learning constraints that complement model-free RL. Unlike model-based RL (Hafner et al., 2019), where models are mainly used to generate imaginary rollouts, this perspective offers a potential bridge between model-free and model-based approaches.

**Theoretical Analysis** We now turn to guarantees for Eik-QRL. Formal statements and proofs for these results are deferred to Appendix A. Optimal value recovery guarantees follow from the existence of a quasimetric approximator $d_{\boldsymbol{\theta}^*} \in \mathcal{Q}\mathrm{met}(\mathcal{S})$ that satisfies $-V^*(s, g) \leq d_{\boldsymbol{\theta}^*}(s, g) \leq -(1 + \epsilon) V^*(s, g)$ for some small $\epsilon > 0$. Given such an approximator, a standard high-probability recovery bound can be derived. However, unlike classical QRL, establishing existence in Eik-QRL is more delicate due to the gradient constraint enforced through the Eik-LRs term in (8). To drive $\epsilon \to 0$ while respecting this constraint, the target $-V^*(s, g)$ must itself obey a matching gradient regularity (Anil et al., 2019). The next lemma formalizes this regularity, ensuring that a compatible $d_{\boldsymbol{\theta}^*}$ exists and closing the argument for optimal value recovery.

**Lemma 4.7** (1-Lipschitz optimal value). *Let $K \subset \mathcal{K}$ be compact and convex, fix any $g \in \mathcal{K}$ for which $d^*(\cdot, g)$ is finite on $K$, and suppose the dynamics on $K$ are the unit-speed integrator $\dot{s}(t) = a(t)$ with $\|a(t)\| \leq 1$ (equivalently, $\mathcal{T}(s, a) = s + a \Delta t + o(\Delta t)$ on $K$). Assume the running cost is constant, $c(s, g) \equiv 1$ on $\mathcal{K} \setminus \{g\}$ and $c(g, g) = 0$. Then, for all $s, s' \in K$, $d^*(\cdot, g)$ is 1-Lipschitz on $K$.*

By Lemma 4.7, we have that $d^*(\cdot, g) = -V^*(\cdot, g)$ is 1-Lipschitz on $K$. This implies the Eikonal condition $\|\nabla_s d^*(s, g)\| = 1$ on $K \setminus \{g\}$, so $d^*$ is *feasible* for the Eik-LRs constraint in (8). This alignment between the true geometry and the Eik-LRs allows a universal quasimetric approximator optimized by (8) to recover $d^*$. We leverage this fact to state our approximation guarantee.

**Theorem 4.8** (Approximate value recovery). *Let the conditions of Lemma 4.7 hold, and $\{d_{\theta}\}_{\theta}$ be the set of universal approximators of $\mathcal{Q}\mathrm{met}(\mathcal{K})$ in the $L_\infty$ norm. Suppose that $d_{\theta^*} \in \{d_{\theta}\}_{\theta}$ is a solution of Eq. (8) such that $d_{\theta^*} \in \mathcal{C}_{\mathrm{grad}}^{(\epsilon)} := \{d_{\theta} : \|\nabla_s d_{\theta}(s, g)\| \leq 1 + \epsilon\}$ for small $\epsilon > 0$. Then, for any $\eta > 0$ and $s \sim \mathcal{P}_{state}, g \sim \mathcal{P}_{goal}$, it holds with probability at least $1 - \mathcal{O}(\frac{\epsilon}{\eta}(-\mathbb{E}[V^*]))$ that $|d_{\theta^*}(s, g) + (1 + \epsilon)V^*(s, g)| \in [-\eta, 0]$.*

Thus, Theorem 4.8 establishes that a universal quasimetric approximator trained with Eik-QRL in (8) recovers $V^*(s, g)$ approximately with high probability, under the sufficient conditions of Lemma 4.7. While these conditions provide clean guarantees, they are not strictly required in practice. As shown in our experiments (Section 6), Eik-QRL remains competitive even when 1-Lipschitz property cannot be verified, as in the `antmaze` tasks. This observation aligns with the fact that if the target $d^*$ is Lipschitz, then approximating it with Lipschitz function, possibly with a different constant, preserves shortest-path geometry up to a positive scaling factor (Sethian, 1996), which is typically sufficient for policy gradient methods given their scale-invariance (Sutton et al., 1999a; Schulman et al., 2015).

In summary, Eik-QRL offers clear structural benefits, including trajectory-free estimation and PDE-based regularization, but at the cost of stronger regularity assumptions on the target value, in particular local Lipschitz continuity of the optimal value function. These assumptions are standard in continuous-

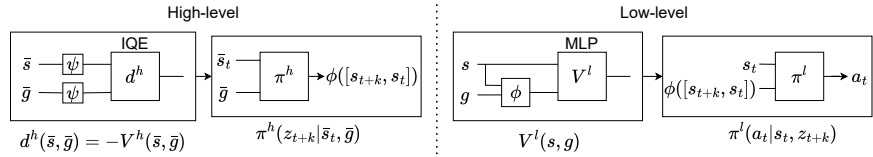

Figure 2: Diagram for Eik-HiQRL. The high-level module consists of a quasimetric model $d^h$, parameterized as an IQE (Wang & Isola), which is trained to optimize the Eik-QRL objective in Eq. (8). The low-level module comprises a value function $V^l$ and a goal representation network $\phi : \mathcal{S} \times \mathcal{G} \to \mathcal{Z}$ (Park et al., 2024b), both instantiated as MLPs and trained via TD-based recursion. Both high-level policy $\pi^h$ and the low-level policy $\pi^l$ are trained via advantage-weighted regression (Peng et al., 2019). Full details are provided in Appendix C.

time optimal control, yet they are not guaranteed in general MDPs. This raises a practical question: *can we retain the advantages of Eik-QRL while mitigating violations of these regularity conditions?*

## 5    EIKONAL-CONSTRAINED HIERARCHICAL QRL (EIK-HIQRL)

To retain the advantages of Eik-QRL while mitigating its limitations, we propose a hierarchical architecture, yielding the Eik-HiQRL algorithm illustrated in Fig. 2. This design is motivated by the following considerations. First, as highlighted in previous work (Park et al., 2024b), a key challenge in GCRL is the low signal-to-noise ratio in value estimates, especially in long-horizon tasks where the initial state and goal are far apart. Park et al. (2024b) propose a *hierarchical solution* in which a high-level actor outputs subgoals and a low-level controller attempts to reach them. Building on this, we show in Appendix B that quasimetric-parameterized value functions can have an even stronger effect than hierarchy alone on mitigating this signal-to-noise problem, further justifying the use of quasimetric value learning strategies such as Eik-QRL, HJB-QRL, and QRL together with hierarchy.

Given this result, a natural question is whether one could rely solely on a single quasimetric value function for both high- and low-level policies. Unfortunately, enforcing quasimetric structures in high-dimensional state spaces is itself challenging as the approximation error of quasimetric parameterizations grows exponentially with the state-space dimension[1]. A natural way to address this problem is to *reduce the state dimensionality* before applying quasimetric projection, either by projecting states into an embedding space or by selecting a subset of task-relevant state variables on which to define subgoals. If, in this process, we ensure that the resulting abstract space evolves under approximately isotropic dynamics, we can then leverage not only the benefits of the quasimetric structure but also *the advantages of the Eikonal formulation (Remark 4.5)*.

Eik-HiQRL is designed precisely to reconcile these aspects: (1) quasimetric projection for the high-level value in a low-dimensional abstract space, (2) PDE-based regularization via the Eikonal formulation, and (3) a hierarchical structure that improves the signal-to-noise ratio. At the high level, we define a lower-dimensional abstract space $\bar{\mathcal{S}}$ (e.g., the positions of the agent or task-relevant objects) where the regularity assumptions of Eik-QRL hold and the quasimetric projection is tractable. This enables learning a high-level value function that improves subgoal generation, going beyond the single-value design in HIQL (Park et al., 2024b). These subgoals then guide the low-level controller more effectively, further mitigating the signal-to-noise ratio issue in value function learning.

In the following section, we provide empirical results that demonstrate the benefits of this design. For all the implementation details for both Eik-HiQRL and Eik-QRL refer to Appendix C.

## 6    EXPERIMENTS

Our experiments are primarily conducted in the Offline GCRL setting, which offers a more controlled testbed than the online counterpart for evaluating value function learning and out-of-distribution generalization. In the offline case, the dataset remains fixed throughout training, making it possible to directly assess an algorithm's ability to generalize beyond the training distribution. By contrast, in the online setting it is difficult to disentangle whether performance fluctuations arise from fortunate

---

[1]This result can be readily derived by extending Wang & Isola, Theorem 3 and combining it with standard results on covering numbers (e.g., Shalev-Shwartz & Ben-David (2014, Ex. 27.1)).

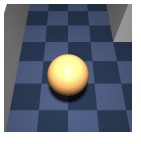
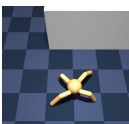
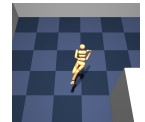
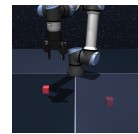
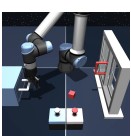

(a)
`pointmaze`    (b) `antmaze`    (c) `humanoid-`
maze    (d)
`antsoccer`    (e) `cube`    (f) `scene`

Figure 3: Environments from OGbench (Park et al., 2024a) used in our experiments.

exploration or genuine improvements in generalization. To this end, we base our experiments on the tasks from OGbench (Park et al., 2024a), summarized in Fig. 3. We provide in Appendix E more details on these environments, including a discussion of how closely their dynamics satisfy the assumptions stated in Section 4.

This section is organized as follows. First, Table 1 provides a controlled comparison between Eik-QRL in (8), HJB-QRL in (7), and QRL in (1), highlighting the pros and cons of introducing PDE-based constraints into quasimetric value learning. The same table also includes our hierarchical formulation Eik-HiQRL, showing how the proposed hierarchical design overcomes the limitations of Eik-QRL on the `antmaze` suite. Next, we compare Eik-HiQRL against strong Offline GCRL baselines on OGbench tasks in Fig. 4, and further stress-test it in non-regular environments (where our assumptions do not hold) in Table 2. Appendix H complements these main results with additional analyses and ablations. Specifically, Appendix H.1 and H.2 investigate the effects of different actors and optimization pipelines on empirical performance; Appendix H.3 reports a complete comparison between Eik-HiQRL and Offline GCRL baselines; Appendix H.4 isolates the impact of the Eikonal constraint within our hierarchy by comparing Eik-HiQRL to HiQRL (without the Eikonal term in the high-level value function). Finally, Appendix H.5 and H.6 respectively explore how to exploit the trajectory-free property of Eik-QRL in practice and evaluate its effectiveness in an online RL setting.

**Empirical Comparison of Quasimetric Formulations**    We begin with a controlled comparison of QRL, HJB-QRL, Eik-QRL, and Eik-HiQRL, intended to provide empirical support for the theoretical results established in the previous sections. We evaluate these algorithms on two benchmark suites: `pointmaze` and `antmaze`. The `pointmaze` environment offers an idealized setting with isotropic dynamics, so that the main assumptions in our analysis are satisfied. In contrast, the `antmaze` suite involves higher-dimensional state spaces and complex dynamics, where discontinuities at contact points challenge the Lipschitz continuity assumption on the dynamics (Assumption 4.2). All algorithms are compared across different maze dimensions and dataset variants. Performance is reported using two main metrics. The first is the success rate $\mathcal{R}$, defined as the percentage of episodes in which a randomly sampled goal is reached. The second is collision rate $\kappa$, measured as the percentage of time steps during which the agent is colliding with obstacles. These results are summarized in Table 1 and learning curves are provided in Appendix G.

A few notes on implementations are in order. Eik-QRL and HJB-QRL are both implemented with a hierarchical actor, consistent with our analysis in Appendix B. Whereas QRL follows the original implementation of Wang et al. (2023) as presented in Fig. 1. As mentioned, Appendix H.1 illustrates an additional comparison where QRL is combined with a hierarchical actor (denoted QRL:hi) in order to isolate the contribution of PDE constraints versus hierarchy. All the details for these variants are provided in Appendix D. Finally, Eik-HiQRL follows the design introduced in Section 5, where, for all the presented navigation tasks, the abstract space $\bar{\mathcal{S}}$ is defined as the agent's coordinate space.

Turning to the results in Table 1, in the `pointmaze` setting, we observe no tangible differences among the three PDE-constrained formulations, consistent with our theoretical analysis. All of them achieve improved accuracy in the estimated value function, which translates into lower collision rates and stronger performance compared to QRL. Refer for instance to `giant` mazes for both `navigate` and `stitch` datasets. QRL achieves competitive success rates, but the learned policies heavily rely on collisions with walls, a behavior that undermines performance in larger environments.

For the `antmaze` suite, we highlight the following observations. Compared to the idealized `pointmaze` setting, the performance of all purely quasimetric algorithms (Eik-QRL, HJB-QRL, and QRL) deteriorates, with particularly pronounced drops in the `giant` variants. This reflects the increased difficulty of learning accurate quasimetrics in higher-dimensional state space and with

Table 1: Summary of the comparison among the QRL formulations. All agents are trained for $10^5$ steps using 10 seeds and evaluated every $10^4$ steps. We report the mean and standard deviation across seeds for the last evaluation. For each seed, evaluations are conducted over 5 different random goals, with the learned policy tested for 50 episodes per goal. For success rate ($\mathcal{R}$), results within 95% of the best value are written in **bold**. For collision avoidance ($\kappa$) the lowest average is colored.

| Environment | Dataset | Dim | Eik-HiQRL $\mathcal{R}$ ($\uparrow$) | Eik-HiQRL $\kappa$ ($\downarrow$) | Eik-QRL $\mathcal{R}$ ($\uparrow$) | Eik-QRL $\kappa$ ($\downarrow$) | HJB-QRL $\mathcal{R}$ ($\uparrow$) | HJB-QRL $\kappa$ ($\downarrow$) | QRL $\mathcal{R}$ ($\uparrow$) | QRL $\kappa$ ($\downarrow$) |
|---|---|---|---|---|---|---|---|---|---|---|
| pointmaze | navigate | medium | **83 ± 9** | 8 ± 2 | **87 ± 7** | 10 ± 4 | **88 ± 6** | 10 ± 3 | 79 ± 4 | 46 ± 4 |
| | | large | 82 ± 12 | 5 ± 2 | **90 ± 8** | 11 ± 8 | **91 ± 7** | 11 ± 6 | **89 ± 7** | 59 ± 3 |
| | | giant | 73 ± 9 | 14 ± 8 | **82±12** | 18 ± 12 | **83 ± 8** | 17 ± 6 | 69 ± 8 | 60 ± 8 |
| | | teleport | **46±17** | 36 ± 6 | 40 ± 11 | 49 ± 14 | 40 ± 8 | 46±11 | 20 ± 9 | 85 ± 8 |
| | stitch | medium | **96 ± 4** | 10 ± 2 | 86 ± 5 | 10 ± 1 | 85 ± 9 | 10 ± 2 | 78 ± 11 | 50 ± 6 |
| | | large | 75 ± 10 | 5 ± 2 | 78 ± 7 | 5 ± 1 | 73 ± 4 | 9 ± 4 | **84±11** | 54 ± 5 |
| | | giant | 62 ± 22 | 28 ± 14 | **73±16** | 19 ± 11 | **70±17** | 19±12 | 51 ± 13 | 61±10 |
| | | teleport | **50±10** | 5 ± 5 | 37 ± 12 | 21 ± 12 | 29 ± 9 | 31±12 | 33 ± 13 | 78 ± 7 |
| antmaze | navigate | medium | **93 ± 2** | 18 ± 2 | 84 ± 3 | 25 ± 1 | 31 ± 4 | 35 ± 3 | 82 ± 7 | 25 ± 4 |
| | | large | **86 ± 2** | 25 ± 2 | 74 ± 3 | 25 ± 1 | 28 ± 8 | 36 ± 2 | 54 ± 9 | 38 ± 4 |
| | | giant | **59 ± 7** | 28 ± 3 | 8 ± 2 | 35 ± 1 | 0 ± 0 | 34 ± 2 | 0 ± 0 | 55 ± 4 |
| | | teleport | **49 ± 2** | 40 ± 3 | 45 ± 3 | 38 ± 2 | 28 ± 4 | 42 ± 2 | 32 ± 6 | 52 ± 5 |
| | stitch | medium | **94 ± 1** | 19 ± 2 | 70 ± 4 | 32 ± 2 | 37 ± 5 | 38 ± 2 | 66 ± 9 | 26 ± 2 |
| | | large | **81 ± 8** | 23 ± 2 | 23 ± 4 | 31 ± 3 | 13 ± 4 | 36 ± 3 | 15 ± 6 | 39 ± 3 |
| | | giant | **47±12** | 29 ± 2 | 0 ± 0 | 44 ± 2 | 0 ± 0 | 46 ± 3 | 0 ± 0 | 52 ± 1 |
| | | teleport | **51 ± 6** | 28 ± 2 | 28 ± 7 | 35 ± 3 | 13 ± 3 | 37 ± 4 | 27 ± 5 | 32 ± 3 |

complex and non-Lipschitz dynamics. Despite this degradation, Eik-QRL consistently outperforms HJB-QRL, underscoring the numerical challenges of optimizing (7). At the same time, Eik-QRL remains on par with QRL (and with QRL:hi in Table 5), indicating that the Eikonal PDE constraint does not excessively compromise effectiveness in this regime. Finally, Eik-HiQRL achieves the strongest performance across the `antmaze` tasks, highlighting the benefits of our hierarchical design.

**Evaluation in Challenging and Non-Regular Environments** This second set of experiments is divided into two parts. The first evaluates Eik-HiQRL against established Offline GCRL algorithms in the `humanoidmaze` navigation tasks. The second examines scenarios where the regularity assumptions do not hold, including MDPs with third-party objects and categorical variables in the state space. These experiments are designed to test performance in settings where our theoretical analysis is difficult to apply. The baselines considered in this section include Hierarchical Implicit Q-Learning (HIQL) (Park et al., 2024b), Contrastive RL (CRL) (Eysenbach et al., 2022), QRL (Wang et al., 2023), and Eikonal-regularized HIQL (Eik-HIQL) (Giammarino et al., 2025). More details about these baselines, including how they differ from our algorithm, are provided in Appendix F.

For the `humanoidmaze` tasks, the results in Fig. 4 provide strong evidence for the effectiveness of our PDE-constrained hierarchical design. Eik-HiQRL consistently outperforms all baselines, with especially large gains in the long-horizon `large` and `giant` settings. The advantage is most evident on `large-stitch` and `giant-stitch`, where Eik-HiQRL achieves statistically significant improvements over the best-performing baseline according to Welch's t-test ($t = 11.7$, $p \approx 10^{-9}$ and $t = 22$, $p \approx 10^{-14}$, respectively). As also highlighted in Table 1, when data stitching is required the PDE-constrained algorithms exhibit their clearest benefits due to the regularizing effect of the PDEs. To the best of our knowledge, Eik-HiQRL achieves SOTA performance on this benchmark. For additional experiments, as well as an ablation comparing Eik-HiQRL with HiQRL (which uses QRL instead of Eik-QRL in the high-level policy), see respectively Appendices H.3 and H.4.

For environments involving external object interactions, we evaluate the `antsoccer` suite, where an `ant` agent must move a ball to a designated target, as well as the `cube` and `scene` robotic manipulation tasks. In these settings, the abstract space $\bar{\mathcal{S}}$ is defined as the concatenation of the ant and ball coordinates for `antsoccer`, and as a latent space learned end-to-end for the two `manipulation` tasks. Specifically, to learn this abstract latent space we define an MLP $\nu : \mathcal{S} \to \mathcal{Z}$, which is trained end-to-end as part of the objective in Eq. (8). During training, we feed $\nu(s)$ and $\nu(g)$ to the quasimetric value network, and the gradients from both the Global Relationships loss and the Eikonal Local Relationships term in Eq. (8) are backpropagated through $\nu$. Importantly, we do not introduce any additional auxiliary loss to explicitly enforce our geometric assumptions in the embedding space. The results, reported in Table 2, show that Eik-HiQRL achieves performance comparable to the baselines. Unlike in the navigation domain, however, we do not observe the same level of performance gains. This reflects the non-trivial characteristics of dynamics in complex manipulation, where interactions with objects introduce additional challenges. In particular, contact

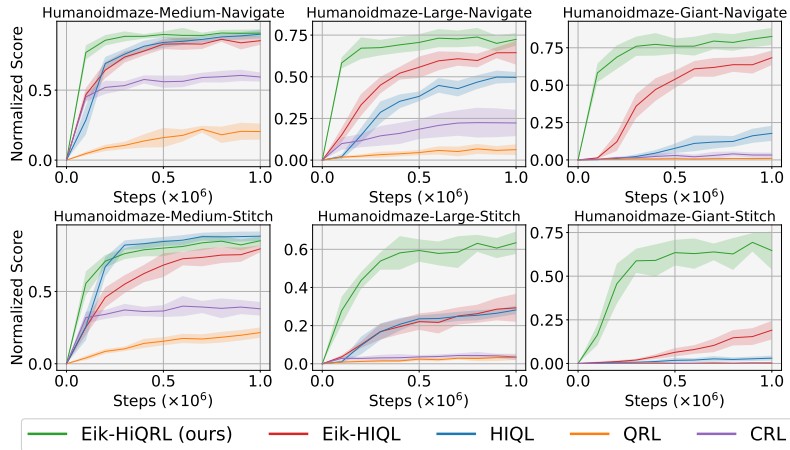

Figure 4: Learning curves for the `humanoidmaze` experiments. All agents are trained for $10^6$ steps using 10 seeds and evaluated every $10^5$ steps. For each seed, evaluations are conducted as in Table 1. Plots show the average success and standard deviation across seeds.

Table 2: Summary of the comparison on non-regular environments. All agents are trained for $10^6$ steps using 10 seeds and evaluated every $10^5$ steps. For each seed, evaluations are conducted as in Table 1 and we report the best evaluation. Results within $95\%$ of the best value are written in **bold**.

| Environment | Dataset | Dimension | Eik-HiQRL (ours) | Eik-HIQL | HIQL | QRL | CRL |
|---|---|---|---|---|---|---|---|
| antsoccer | navigate | arena | **61 ± 5** | 19 ± 2 | **60 ± 4** | 10 ± 3 | 24 ± 2 |
| | | medium | **13 ± 1** | 3 ± 2 | **13 ± 3** | 2 ± 2 | 4 ± 2 |
| | stitch | arena | **32 ± 4** | 2 ± 0 | 17 ± 3 | 2 ± 1 | 1 ± 1 |
| | | medium | **7 ± 2** | 1 ± 0 | 5 ± 1 | 0 ± 0 | 0 ± 0 |
| manipulation | cube-single-play | | 12 ± 3 | 25 ± 1 | **31 ± 4** | 11 ± 3 | **32 ± 2** |
| | scene-play | | **55 ± 14** | **52 ± 7** | **52 ± 3** | 8 ± 2 | 35 ± 6 |

events are often represented through categorical or mode-switching variables (e.g., in-contact vs. out-of-contact), which induce sharp discontinuities and hybrid structure in the underlying value function. These discontinuities are at odds with the smooth topology implicitly assumed by our PDE-based constraints (Assumption 4.4), so that enforcing Eikonal constraints in this regime introduces a bias that does not perfectly match the true optimal value landscape. We believe that these peculiarities make robotic manipulation a promising direction for follow-up work, with significant opportunities to design PDE-constrained algorithms explicitly tailored to contact-rich, hybrid dynamics. Learning curves for these experiments are provided in Appendix G.

## 7 CONCLUSION

In this work, we derived and analyzed Eik-QRL, a PDE-constrained algorithm that introduces a PINN-based perspective on value learning for model-free GCRL. Our analysis highlights both the strengths and limitations of this formulation, motivating its hierarchical extension, Eik-HiQRL, and outlining several avenues for future work. We complement this analysis with extensive experiments that provide empirical support for our claims. These results demonstrate the effectiveness of Eik-HiQRL while also underscoring the challenges of applying PDE-constrained algorithms in non-regular environments, pointing to practical and impactful directions for further research. An especially promising direction is to leverage our analysis more directly for representation design. By explicitly characterizing which geometric and regularity properties are beneficial for PDE-based value learning, our results clarify what structural features an embedding space should satisfy for PDE-based algorithms to be applicable. Designing representation-learning mechanisms that realize such embeddings while still supporting effective control remains an open challenge. Progress along this line could offer a compelling path to further bridge the gap between formal guarantees and real-world deployment. Taken together, our contributions establish both theoretical novelty and practical relevance, and we hope this work will serve as a solid foundation for subsequent advances in PDE-informed RL.

## 8 REPRODUCIBILITY STATEMENT

We have taken extensive measures to ensure the reproducibility of our results. The main algorithms are presented concisely in the paper, with design choices, loss formulations, and training procedures clearly specified. Additional implementation details, hyperparameters, and specifications are provided in Appendix C for Eik-HiQRL and in Appendix D for the other variants. To further support reproducibility, we include a preliminary version of our code with this submission, and we will release a cleaned-up and fully documented version with the camera-ready paper.

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

CONTENTS

# A   THEORETICAL RESULTS FOR SECTION 4

## A.1   ASSUMPTIONS RESTATED

We begin by restating the assumptions introduced in Section 4. Unless otherwise specified, these assumptions are assumed to hold throughout this section.

**Assumption A.1** (Compact state and action spaces). Let the state space $\mathcal{S}$ and the action space $\mathcal{A}$ be compact, and let $\mathcal{K} \subseteq \mathcal{S}$ denote the feasible (obstacle-free) subset.

**Assumption A.2** (Lipschitz dynamics). The transition map $\mathcal{T} : \mathcal{S} \times \mathcal{A} \to \mathcal{S}$ is deterministic and Lipschitz continuous in $s$. That is, there exists a constant $L_\mathcal{T} > 0$ such that for every $s, s' \in \mathcal{K}$ and $a \in \mathcal{A}$: $\|\mathcal{T}(s,a) - \mathcal{T}(s',a)\| \le L_\mathcal{T} \|s - s'\|$. Equivalently, in the continuous-time limit $\Delta t \to 0$, we have $\mathcal{T}(s,a) = s + f(s,a)\,\Delta t + o(\Delta t)$, where $f(\cdot, a)$ is Lipschitz continuous in $s$ (uniformly in $a$).

**Assumption A.3** (Cost regularity). The cost $c : \mathcal{S} \times \mathcal{G} \to \mathbb{R}_{\ge 0}$ is continuous, bounded, and nonnegative. Moreover, $c(g, g) = 0$ for all $g \in \mathcal{G}$, consistent with the shortest-path interpretation.

**Assumption A.4** (Value function regularity). For each fixed goal $g \in \mathcal{K}$, the optimal value function $d^*(\cdot, g)$ exists, is finite on $\mathcal{K}$, and is locally Lipschitz.

## A.2   LEMMA 4.7 RESTATED

**Lemma A.5** (1-Lipschitz property of the optimal value function). *Let $K \subset \mathcal{K}$ be compact and convex, and fix any $g \in \mathcal{K}$ such that $d^*(\cdot, g)$ is finite on $K$. Suppose the dynamics on $K$ are given by the unit-speed integrator*

$$\dot{s}(t) = a(t), \quad \|a(t)\| \le 1,$$

*equivalently, $\mathcal{T}(s,a) = s + a\,\Delta t + o(\Delta t)$ on $K$. Assume the running cost is constant, with $c(s,g) \equiv 1$ for $s \ne g$ and $c(g,g) = 0$. Then $d^*(\cdot, g)$ is 1-Lipschitz on $K$, i.e., for all $s, s' \in K$,*

$$|d^*(s, g) - d^*(s', g)| \le \|s - s'\|.$$

*Proof.* By the triangle inequality for $d^*$,

$$|d^*(s, g) - d^*(s', g)| \ \le \ \max\{d^*(s, s'),\, d^*(s', s)\}.$$

Since $K$ is convex and the dynamics are $\dot{s} = a$ with $\|a\| \le 1$, the straight-line trajectory from $s$ to $s'$ is feasible at unit speed and requires time $\|s - s'\|$. Hence $d^*(s, s') \le \|s - s'\|$ and $d^*(s', s) \le \|s - s'\|$. Substituting into the inequality above proves the claim. $\qquad\square$

## A.3   THEOREM 4.8 RESTATED

Before restating and proving Theorem 4.8 we introduce the following auxiliary lemma.

**Lemma A.6** (Bound for Gradient-Regularized Quasimetrics). *Define the feasible region*

$$\mathcal{F} = \{(s, g) \in \mathcal{K} \times \mathcal{K} \ : \ V^*(s, g) > -\infty\}.$$

*Suppose the conditions of Lemma A.5 hold, and therefore, we can assume at the same time:*

(A1) ***Gradient constraint:*** *$d_\theta \in \mathcal{Q}\mathrm{met}(\mathcal{S})$ is continuously differentiable in $s$ and satisfies*

$$\|\nabla_s d_\theta(s, g)\| \le 1 + \epsilon, \quad \forall s, g \in \mathcal{K}.$$

(A2) ***Approximation of relaxed value:*** *$d_\theta$ is a universal approximator of $\mathcal{Q}\mathrm{met}(\mathcal{K})$ in the $L_\infty$ norm. Therefore, it approximates the value function $\widetilde{V}^*$ of a relaxed-cost MDP with reward $\widetilde{\mathcal{R}}(s,a) = -1 - \epsilon/2$, up to additive error $\epsilon/2$:*

$$\left| d_\theta(s, g) + \widetilde{V}^*(s, g) \right| \le \frac{\epsilon}{2}, \quad \forall s, g \in \mathcal{S}.$$

*Then, for all $(s, g) \in \mathcal{F}$, the following bound holds:*

$$\boxed{-V^*(s, g) \leq d_\theta(s, g) \leq -(1 + \epsilon)V^*(s, g)}.$$

*Proof.* **Lower bound.** This part of the proof follows from Lemma 1 in Wang et al. (2023). In our case, however, assumptions $(A1)$ and $(A2)$ can only be satisfied simultaneously under the conditions stated in Lemma A.5.

Let $\widetilde{V}^*$ denote the value function of the relaxed-cost MDP with reward $\widetilde{\mathcal{R}}(s, s') = -1 - \epsilon/2$. Then:

$$\widetilde{V}^*(s, g) = (1 + \epsilon/2)V^*(s, g).$$

By assumption $(A2)$,

$$d_\theta(s, g) \geq -\widetilde{V}^*(s, g) - \frac{\epsilon}{2} = -(1 + \epsilon/2)V^*(s, g) - \frac{\epsilon}{2}.$$

If $s \neq g$, then $V^*(s, g) \leq -1$, so:

$$-(1 + \epsilon/2)V^*(s, g) - \frac{\epsilon}{2} \geq -V^*(s, g),$$

hence

$$d_\theta(s, g) \geq -V^*(s, g).$$

If $s = g$, then $V^*(s, g) = 0$, and by definition of a quasimetric and goal-reaching cost, we also have $d_\theta(s, g) = \widetilde{V}^*(s, g) = 0$. Thus, the inequality holds with equality.

**Upper bound.** Let $s_0 = s \to s_1 \to \cdots \to s_T = g$ be an optimal path of length $T = -V^*(s, g)$. Using the triangle inequality:

$$d_\theta(s, g) \leq \sum_{t=0}^{T-1} d_\theta(s_t, s_{t+1}).$$

Define the interpolation $\gamma_t(u) = s_t + u(s_{t+1} - s_t)$, for $u \in [0, 1]$. Note that the whole one-step interpolation path lies within the feasible space $\mathcal{K}$. Then:

$$d_\theta(s_t, s_{t+1}) = \int_0^1 \nabla_s d_\theta(\gamma_t(u), g)^\top (s_{t+1} - s_t) \, du \leq \int_0^1 \|\nabla_s d_\theta(\gamma_t(u), g)\| \cdot \|s_{t+1} - s_t\| \, du.$$

Since $\|\nabla_s d_\theta\| \leq 1 + \epsilon$, and for $\Delta t$ small enough $\|s_{t+1} - s_t\| \leq 1$, we get:

$$d_\theta(s_t, s_{t+1}) \leq (1 + \epsilon).$$

Summing over the $T$ steps we obtain:

$$d_\theta(s, g) \leq T \cdot (1 + \epsilon) = (1 + \epsilon)(-V^*(s, g)).$$

Leading to the expected bound:

$$d_\theta(s, g) \leq -(1 + \epsilon)V^*(s, g).$$

$\square$

Provided Lemma A.6, we can now formally restate and prove Theorem 4.8.

**Theorem A.7** (Approximate Value Recovery under Gradient Constraint). *Define the feasible region*

$$\mathcal{F} = \{(s, g) \in \mathcal{K} \times \mathcal{K} \; : \; V^*(s, g) > -\infty\}.$$

*Suppose the conditions of Lemma A.5 hold and furthermore*

(i) *$\{d_\theta\}_\theta$ is a family of quasimetric models, i.e., a set of universal approximators of $\mathcal{Q}\mathrm{met}(\mathcal{S})$ in the $L_\infty$ norm;*

(ii) *The model $d_\theta^* \in \{d_\theta\}_\theta$ is a solution to the constrained optimization problem:*

$$\min_\theta \max_{\lambda > 0} \mathbb{E}_{s,g} \left[ -\zeta(d_\theta(s, g)) \right] + \lambda \, \mathbb{E}_{s,g} \left[ (\|\nabla_s d_\theta(s, g)\| - 1)^2 \right] - \lambda \epsilon^2,$$

*for some strictly increasing convex function $\zeta$, such that $d_\theta^* \in \mathcal{C}_{\mathrm{grad}}^{(\epsilon)} := \{d_\theta : \|\nabla_s d_\theta(s, g)\| \leq 1 + \epsilon \; \forall (s, g) \in \mathcal{F}\}$;*

*Then for all $(s, g) \in \mathcal{F}$, it holds that:*

$$\boxed{-V^*(s, g) \;\leq\; d_{\theta^*}(s, g) \;\leq\; -(1 + \epsilon)V^*(s, g)}.$$

*Moreover, for any $\eta > 0$, and $s \sim \mathcal{P}_{\mathrm{state}}, g \sim \mathcal{P}_{\mathrm{goal}}$, we have:*

$$|d_{\theta^*}(s, g) + (1 + \epsilon)V^*(s, g)| \leq \eta \quad \text{with probability at least} \quad 1 - \mathcal{O}\left(\frac{\epsilon}{\eta} \cdot \left(-\mathbb{E}_{s,g}[V^*(s, g)]\right)\right).$$

*Proof.* By Lemma A.6, and under the assumptions of the theorem, we have for all $(s, g) \in \mathcal{F}$:

$$-V^*(s, g) \leq d_{\theta^*}(s, g) \leq -(1 + \epsilon)V^*(s, g). \tag{9}$$

This bound means the learned quasimetric overestimates the cost (i.e., is more pessimistic than $-V^*$), but not more than a factor $1 + \epsilon$.

We now quantify how close $d_{\theta^*}(s, g)$ is to the upper bound $-(1 + \epsilon)V^*(s, g)$ with high probability.

**Define the bad event.** Let $\eta > 0$ be a small tolerance. Define the event:

$$E := \{(s, g) : d_{\theta^*}(s, g) < -(1 + \epsilon)V^*(s, g) - \eta\}.$$

This is the "bad" event where the quasimetric overshoots the pessimistic bound by more than $\eta$. Let $p$ be the bad event probability $p = \mathbb{P}[E]$ for all $s \sim \mathcal{P}_{\mathrm{state}}$ and $g \sim \mathcal{P}_{\mathrm{goal}}$.

**Upper-bound the expectation.** Split the expectation of $d_{\theta^*}$ over the event $E$ and its complement $E^c$:

$$\mathbb{E}[d_{\theta^*}] = \mathbb{E}[d_{\theta^*} \mid E] \cdot p + \mathbb{E}[d_{\theta^*} \mid E^c] \cdot (1 - p).$$

On $E$, by definition of the event:

$$d_{\theta^*}(s, g) \leq -(1 + \epsilon)V^*(s, g) - \eta \quad \Rightarrow \quad \mathbb{E}[d_{\theta^*} \mid E] \leq -(1 + \epsilon)\mathbb{E}[V^*] - \eta.$$

On $E^c$, by the upper bound from equation 9:

$$d_{\theta^*}(s, g) \leq -(1 + \epsilon)V^*(s, g) \quad \Rightarrow \quad \mathbb{E}[d_{\theta^*} \mid E^c] \leq -(1 + \epsilon)\mathbb{E}[V^*].$$

Putting these together:

$$\mathbb{E}[d_{\theta^*}] \leq p \cdot (-(1 + \epsilon)\mathbb{E}[V^*] - \eta) + (1 - p) \cdot (-(1 + \epsilon)\mathbb{E}[V^*])$$
$$= -(1 + \epsilon)\mathbb{E}[V^*] - p\eta.$$

**Lower-bound the expectation.** From the lower bound of equation 9:

$$d_{\theta^*}(s, g) \geq -V^*(s, g) \quad \Rightarrow \quad \mathbb{E}[d_{\theta^*}] \geq -\mathbb{E}[V^*].$$

**Conclusion.** Putting both bounds on $\mathbb{E}[d_{\theta^*}]$ together:

$$-(1 + \epsilon)\mathbb{E}[V^*] - p\eta \geq -\mathbb{E}[V^*] \quad \Rightarrow \quad p \leq \frac{\epsilon}{\eta} \cdot (-\mathbb{E}[V^*]).$$

With probability at least

$$1 - \frac{\epsilon}{\eta} \cdot (-\mathbb{E}[V^*]),$$

the bad event $E$ does not occur. That is,

$$d_{\theta^*}(s, g) \geq -(1 + \epsilon)V^*(s, g) - \eta.$$

Combined with the always-valid upper bound $d_{\theta^*}(s, g) \leq -(1 + \epsilon)V^*(s, g)$, we conclude:

$$|d_{\theta^*}(s, g) + (1 + \epsilon)V^*(s, g)| \in [-\eta, 0] \quad \text{with probability at least} \quad 1 - \mathcal{O}\left(\frac{\epsilon}{\eta} \cdot (-\mathbb{E}[V^*])\right).$$

$\square$

## B   DIDACTIC EXAMPLE: JOINT BENEFITS OF HIERARCHY AND QUASIMETRIC

In the following we provide additional theoretical results that justify the use of a quasimetric value function together with a hierarchical actor.

We consider a goal-conditioned value function $V(s, g)$ in the 1D environment in Fig. 5, where an agent moves toward a fixed goal state $g$ from an initial state $s$, making decisions based on $V(s, g)$. The true optimal value function (assuming a step cost of $-1$) is given by:

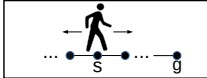

Figure 5: Toy environment.

$$V^*(s, g) = -|s - g| = -T.$$

The learned (unprojected) noisy value function follows a multiplicative Gaussian noise model:

$$\hat{V}(s, g) = V^*(s, g)(1 + \sigma Z_{s,g}), \quad Z_{s,g} \sim \mathcal{N}(0, 1), \tag{10}$$

with noise magnitude $\sigma > 0$.

We then apply an approximate quasimetric projection to enforce triangle inequality constraints:

$$\tilde{V}(s, g) \geq \tilde{V}(s, s_1) + \tilde{V}(s_1, g), \quad \forall s, s_1, g.$$

This projection modifies the noise structure by decomposing long-horizon estimates into 1-step segments:

$$\tilde{V}(s, g) = \sum_{i=s}^{g-1} \tilde{v}(i), \quad \tilde{v}(i) = -1 - \sigma Z_i, \quad Z_i \sim \mathcal{N}(0, 1). \tag{11}$$

In the following, we state Proposition B.1, which provides a novel bound on the probability of actor's error under quasimetric value function parameterization. We provide a complete proof for Proposition B.1, and include a brief comparison with the corresponding results obtained without quasimetric projection in Park et al. (2024b).

**Proposition B.1.** *In the environment of Fig. 5, after applying the quasimetric projection described above, the error probabilities of the flat policy $\pi$ and the hierarchical policy $\pi^l \circ \pi^h$ are respectively bounded by:*

$$\mathcal{E}(\pi) \leq \Phi\left(-\frac{\sqrt{2}}{\sigma\sqrt{T(1 - \rho_{flat})}}\right),$$

$$\mathcal{E}(\pi^l \circ \pi^h) \leq \Phi\left(-\frac{\sqrt{2}k}{\sigma\sqrt{T(1 - \rho_{high})}}\right) + \Phi\left(-\frac{\sqrt{2}}{\sigma\sqrt{k(1 - \rho_{low})}}\right),$$

*where $T$ is the full horizon, $k$ is the number of steps after which $\pi^h$ selects a subgoal, $\Phi$ denotes the cumulative distribution function of the standard normal distribution, $\Phi(x) = \mathbb{P}[Z \leq x] = \frac{1}{\sqrt{2\pi}}\int_{-\infty}^{x} e^{-t^2/2}\,dt$, and $\rho \in [0, 1]$ is the correlation coefficient between the total noise accumulated along the two neighboring values that the policy compares.*

*Proof.* The proof follows similarly to (Park et al., 2024b, Proposition 4.1), however under the approximate quasimetric projection model.

**Flat policy.** The flat policy chooses between $\tilde{V}(s - 1, g)$ and $\tilde{V}(s + 1, g)$, so define the difference:

$$\Delta_{\text{flat}} = \tilde{V}(s + 1, g) - \tilde{V}(s - 1, g) = 2 - \sigma(S_2 - S_1),$$

where $S_1 = \sum_{i=1}^{T+1} Z_i$, $S_2 = \sum_{i=1}^{T-1} Z_i'$, and $\rho_{\text{flat}} = \text{Corr}(S_1, S_2)$. Then:

$$\text{Var}(\Delta_{\text{flat}}) = \sigma^2\left[(T + 1) + (T - 1) - 2\rho_{\text{flat}}\sqrt{(T + 1)(T - 1)}\right] \approx 2\sigma^2 T(1 - \rho_{\text{flat}}),$$

yielding:

$$\mathcal{E}(\pi) = \mathbb{P}[\Delta_{\text{flat}} < 0] \leq \Phi\left(-\frac{2}{\sigma\sqrt{2T(1 - \rho_{\text{flat}})}}\right).$$

**Hierarchical policy (high-level).** Let the high-level policy compare subgoals at $s - k$ and $s + k$. Define:
$$\Delta_{\text{high}} = \tilde{V}(s + k, g) - \tilde{V}(s - k, g) = 2k - \sigma(S_2 - S_1),$$
Let $S_1 = \sum_{i=1}^{T+k} Z_i$, $S_2 = \sum_{i=1}^{T-k} Z_i'$, and assume $\text{Corr}(S_1, S_2) = \rho_{\text{high}}$. Then:
$$\text{Var}(\Delta_{\text{high}}) = \sigma^2 \text{Var}(S_2 - S_1) = \sigma^2 \left[ (T + k) + (T - k) - 2\rho_{\text{high}} \sqrt{(T + k)(T - k)} \right],$$
$$\text{Var}(\Delta_{\text{high}}) = \sigma^2 \left[ 2T - 2\rho_{\text{high}} \sqrt{T^2 - k^2} \right].$$
When $T \gg k$, we use $\sqrt{T^2 - k^2} \approx T - \frac{k^2}{2T}$, yielding:
$$\text{Var}(\Delta_{\text{high}}) \approx 2\sigma^2 T(1 - \rho_{\text{high}}).$$
so:
$$\mathcal{E}(\pi^h) \leq \Phi\left( -\frac{2k}{\sigma\sqrt{2T(1 - \rho_{\text{high}})}} \right).$$

**Hierarchical policy (low-level).** For reaching a chosen subgoal (horizon $k$), define:
$$\Delta_{\text{low}} = \tilde{V}(s + 1, s + k) - \tilde{V}(s - 1, s + k) = 2 - \sigma(S_2 - S_1),$$
where each sum is over $k \pm 1$ terms. Then:
$$\text{Var}(\Delta_{\text{low}}) \approx 2\sigma^2 k(1 - \rho_{\text{low}}),$$
and:
$$\mathcal{E}(\pi^l) \leq \Phi\left( -\frac{2}{\sigma\sqrt{2k(1 - \rho_{\text{low}})}} \right).$$

**Total hierarchical error.** Using a union bound:
$$\mathcal{E}(\pi^l \circ \pi^h) \leq \mathcal{E}(\pi^h) + \mathcal{E}(\pi^l),$$
which concludes the result. $\qquad\square$

In summary, under quasimetric projection we get from Proposition B.1:
$$\tilde{\mathcal{E}}(\pi) \leq \Phi\left( -\frac{\sqrt{2}}{\sigma\sqrt{T(1 - \rho_{\text{flat}})}} \right), \tag{12}$$
$$\tilde{\mathcal{E}}(\pi^l \circ \pi^h) \leq \Phi\left( -\frac{\sqrt{2}k}{\sigma\sqrt{T(1 - \rho_{\text{high}})}} \right) + \Phi\left( -\frac{\sqrt{2}}{\sigma\sqrt{k(1 - \rho_{\text{low}})}} \right). \tag{13}$$

Without quasimetric projection from (Park et al., 2024b, Proposition 4.1) we get:
$$\mathcal{E}(\pi) \leq \Phi\left( -\frac{\sqrt{2}}{\sigma\sqrt{T^2 + 1}} \right), \tag{14}$$
$$\mathcal{E}(\pi^l \circ \pi^h) \leq \Phi\left( -\frac{\sqrt{2}k}{\sigma\sqrt{T^2 + k^2}} \right) + \Phi\left( -\frac{\sqrt{2}}{\sigma\sqrt{k^2 + 1}} \right). \tag{15}$$

Eik-HiQRL gets
$$\mathcal{E}_{\text{Eik-HiQRL}}(\pi^l \circ \pi^h) \leq \Phi\left( -\frac{\sqrt{2}k}{\sigma\sqrt{T(1 - \rho_{\text{high}})}} \right) + \Phi\left( -\frac{\sqrt{2}}{\sigma\sqrt{k^2 + 1}} \right). \tag{16}$$

From Proposition B.1, the tightest upper bound is obtained when quasimetric projection is used for both $\pi^l$ and $\pi^h$, corresponding to the bound in Eq. (13). This strategy is implemented in our Eik-QRL and QRL:hi variants (see Appendix D), and it explains the strong empirical results observed in the idealized `pointmaze` setup (Table 5 in Appendix H). In higher-dimensional spaces such

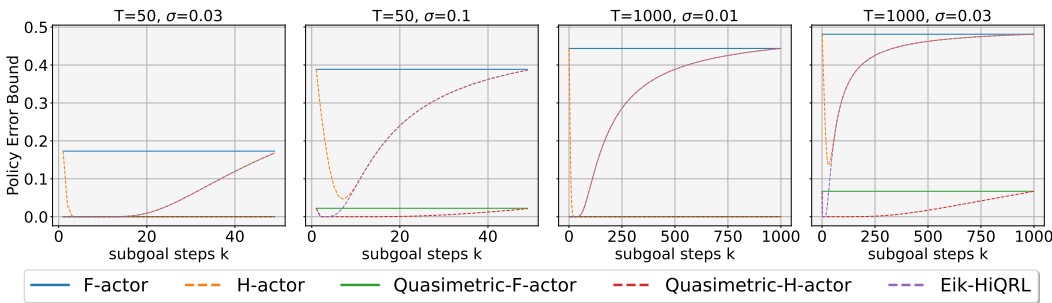

Figure 6: Comparison of policy errors in the didactic environment in Fig. 5. In the figure legend, F-actor corresponds to the bound in (14), H-actor to (15), Quasimetric-F-actor to (12), Quasimetric-H-actor to (13), and Eik-HiQRL to (16).

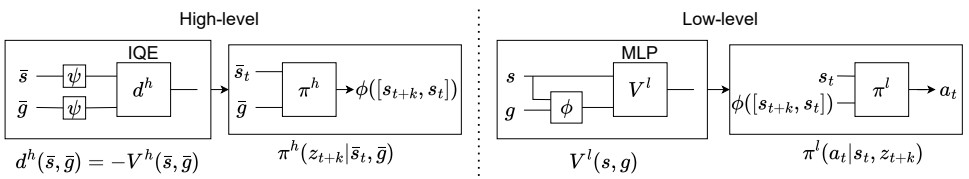

Figure 7: Diagram of the Eik-HiQRL architecture. The high-level planning module consists of a quasimetric model $d^h$, parameterized as an IQE (Wang & Isola), which is trained to optimize the physics-informed objective in (17), and a high-level policy $\pi^h$ trained via the advantage-weighted objective in (19). The low-level module comprises a value function $V^l$ and a goal representation network $\phi$, both instantiated as MLPs and trained to minimize the Implicit V-Learning loss in (18). The low-level policy $\pi^l$ is trained to maximize the advantage-weighted regression objective in (20).

as `antmaze`, however, accurate quasimetric projections become substantially harder to obtain. This difficulty manifests as increased noise $\sigma$ in Eq. (11), which in turn explains the challenges faced by all purely quasimetric-based algorithms, whether PDE-constrained or not, when applied to high-dimensional state spaces.

In this analysis, the correlation coefficient $\rho$ is a modeling parameter, and, similarly to $\sigma$ in Eqs. (10)–(11), it is typically not estimated from data in practical environments. In Proposition B.1, we introduce $\rho$ for correctness: after quasimetric projection, the noise terms on neighboring segments become correlated, and this correlation must be reflected in the variance of the value difference. In the illustrative example of Fig. 6, we simply fix $\rho = 0.01$. Choosing a small correlation coefficient makes the bound more conservative, corresponding to a harder regime where shared noise between neighboring estimates is limited, and random fluctuations have a stronger impact on value comparisons.

Finally, Fig. 6 compares the bounds derived above. Provided an appropriately chosen subgoal step $k$, Eik-HiQRL achieves performance comparable to fully quasimetric agents. Moreover, hierarchical policies consistently outperform their flat counterparts, underscoring the benefits of temporal abstraction in long-horizon tasks.

## C  EIK-HIQRL

In the following, we provide a comprehensive overview of the Eik-HiQRL algorithm, including full implementation details, complete pseudo-code, and the main hyperparameters used in our experiments. Additional resources and extended documentation are available in our GitHub repository.

## C.1 OVERVIEW

Refer to Fig. 7 for a graphical illustration of the Eik-HiQRL architecture. The high-level planning module consists of a quasimetric model parameterized by an IQE (Wang & Isola), together with a high-level policy that outputs a subgoal embedding $z_{t+k} = \phi([s_{t+k}, s_t])$. The low-level control module includes a value function $V^l$ parameterized by an MLP, as well as a goal representation network $\phi$, also instantiated as an MLP with length normalization applied to its final layer. The low-level policy takes the current state and the subgoal embedding as input and outputs an action to interact with the environment.

Recall that the value decomposition described above enables the construction of a high-level abstract space $\bar{\mathcal{S}}$. Accordingly, in Fig. 7, we denote the inputs to the high-level module by $\bar{s}, \bar{g} \in \bar{\mathcal{S}}$.

The high-level value function $d_{\boldsymbol{\theta}}$ (denoted as $d^h$ in Fig. 7) is trained to solve the following Eik-QRL optimization problem:

$$\max_{\boldsymbol{\theta}} \underbrace{\mathbb{E}_{s,g}\left[\zeta(d_{\boldsymbol{\theta}}(s,g))\right]}_{\text{Global Relationships (GRs)}} \quad \text{s.t.} \quad \underbrace{\mathbb{E}_{s\sim\mathcal{P}_{\text{state}},\, g\sim\mathcal{P}_{\text{goal}}}\left[(\|\nabla_s d_{\boldsymbol{\theta}}(s,g)\| - 1)^2\right]}_{\text{Eikonal-Local Relationships (Eik-LRs)}} \leq \epsilon^2, \tag{17}$$

where $\mathcal{P}_{\text{state}}$ and $\mathcal{P}_{\text{goal}}$ denote the marginal state and goal distributions induced by the offline dataset, and $\zeta$ is a monotonically increasing transformation that dampens large distances $d_{\boldsymbol{\theta}}(s,g)$.

The low-level value function $V_{\boldsymbol{\theta}_V}$ (referred to as $V^l$ in Fig. 7) and the goal representation network $\phi_{\boldsymbol{\theta}_\phi}$ ($\phi$ in Fig. 7) are trained by minimizing the following Implicit V-Learning objective:

$$\mathcal{L}_V(\boldsymbol{\theta}_V, \boldsymbol{\theta}_\phi) = \mathbb{E}_{(s,s')\sim\mathcal{D},\, g\sim\mathcal{P}_g}\left[L_2^\iota\big(\mathcal{R}(s,g) + \gamma V_{\bar{\boldsymbol{\theta}}_V}(s', z'_\phi) - V_{\boldsymbol{\theta}_V}(s, z_\phi)\big)\right], \tag{18}$$

where $z_\phi$ and $z'_\phi$ are the goal embeddings produced by $\phi_{\boldsymbol{\theta}_\phi}$, $\bar{\boldsymbol{\theta}}_V$ are the parameters of the target value network (Mnih, 2013), and $L_2^\iota(x) = |\iota - \mathbb{1}(x < 0)|x^2$ is the expectile loss with $\iota \in [0.5, 1]$.

Policy extraction is performed via advantage-weighted regression for both the high-level policy $\pi_{\boldsymbol{\theta}_h}^h$ and the low-level policy $\pi_{\boldsymbol{\theta}_l}^l$, using the following objectives:

$$J_{\pi^h}(\boldsymbol{\theta}_h) = \mathbb{E}_{(s_t,s_{t+k})\sim\mathcal{D},\, g\sim\mathcal{P}_g}\left[\exp\big(\beta \cdot \tilde{A}^h(s_t, s_{t+k}, g)\big) \log \pi_{\boldsymbol{\theta}_h}^h(s_{t+k} \mid s_t, g)\right], \tag{19}$$

$$J_{\pi^l}(\boldsymbol{\theta}_l) = \mathbb{E}_{(s_t,a_t,s_{t+1},s_{t+k})\sim\mathcal{D}}\left[\exp\big(\beta \cdot \tilde{A}^l(s_t, a_t, s_{t+k})\big) \log \pi_{\boldsymbol{\theta}_l}^l(a_t \mid s_t, s_{t+k})\right], \tag{20}$$

where $\beta$ is an inverse temperature hyperparameter, and the advantages are approximated as $\tilde{A}^h(s_t, s_{t+k}, g) = d_{\boldsymbol{\theta}}(s_t, g) - d_{\boldsymbol{\theta}}(s_{t+k}, g)$ and $\tilde{A}^l(s_t, a_t, s_{t+k}) = V_{\boldsymbol{\theta}_V}(s_{t+1}, s_{t+k}) - V_{\boldsymbol{\theta}_V}(s_t, s_{t+k})$.

## C.2 IMPLEMENTATION DETAILS

In this section, we describe how the optimization problem in (17) is solved in practice. We propose two main variants: the first, which we refer to as Eik-QRL-$\lambda$, and a simpler unconstrained version, Eik-QRL.

The Eik-QRL-$\lambda$ variant adopts a constrained optimization approach based on the Lagrangian relaxation of the original problem. Specifically, it formulates a minimax objective that introduces a dual variable $\lambda \geq 0$ to explicitly enforce the Eikonal constraint:

$$\min_{\boldsymbol{\theta}} \max_{\lambda \geq 0} \mathbb{E}_{s\sim\mathcal{P}_{\text{state}},\, g\sim\mathcal{P}_{\text{goal}}}\left[-\zeta(d_{\boldsymbol{\theta}}(s,g)) + \lambda\left(\|\nabla_s d_{\boldsymbol{\theta}}(s,g)\| - 1\right)^2 - \lambda\epsilon^2\right], \tag{21}$$

This formulation ensures a principled trade-off between maximizing global value alignment (through the first term) and enforcing local Lipschitz continuity via the constraint. However, it requires careful tuning of dual ascent steps and often leads to unstable optimization dynamics due to the interplay between the primal and dual variables.

By contrast, the Eik-QRL variant simplifies the training process by removing the explicit constraint and instead incorporating the local regularization term directly into the objective as a soft penalty:

$$\min_{\boldsymbol{\theta}} \mathbb{E}_{s\sim\mathcal{P}_{\text{state}},\, g\sim\mathcal{P}_{\text{goal}}}\left[-\zeta(d_{\boldsymbol{\theta}}(s,g)) + (\|\nabla_s d_{\boldsymbol{\theta}}(s,g)\| - 1)^2\right]. \tag{22}$$

Table 3: Hyperparameter values for Eik-HiQRL.

| Hyperparameter Name | Value |
|---|---|
| Actor $(\pi^h_{\boldsymbol{\theta}_h}, \pi^l_{\boldsymbol{\theta}_l})$ hidden dims | $(512, 512, 512)$ |
| Values $(d_{\boldsymbol{\theta}}, V_{\boldsymbol{\theta}_V})$ hidden dims | $(512, 512, 512)$ |
| Goal representation network $(\phi_{\boldsymbol{\theta}_\phi})$ hidden dims | $(512, 512, 512)$ |
| Quasimetric type | IQE |
| Quasimetric latent dim $(m)$ | 512 |
| Subgoal representation dim $(\dim(\mathcal{Z}))$ | 10 |
| Discount factor $(\gamma)$ | 0.99 |
| Batch size $(B)$ | 1024 |
| Optimizer | Adam |
| Learning rates $\alpha_d, \alpha_V, \alpha_h, \alpha_l$ | $3 \cdot 10^{-4}$ |
| Target update rate $(\tau)$ | 0.005 |
| Expectile factor $(\iota)$ | 0.7 |
| Inverse temperature parameter $(\beta)$ | 3.0 |

This relaxed formulation eliminates the need for dual variable updates and simplifies implementation, while still encouraging the learned quasimetric to satisfy the Eikonal constraint approximately. Although it lacks explicit constraint satisfaction guarantees, we find it to be more robust in practice and easier to tune.

We evaluate both implementations in the ablation study reported in Section H.2. Our final algorithm adopts the simpler formulation in (22), as it demonstrated superior training stability and consistently higher performance with both flat and hierarchical actors. Refer to Table 6 for a summary of these results.

### C.3 PSEUDO-CODE AND HYPERPARAMETERS

In this section, we provide implementation-level details necessary to reproduce our results. We first present, in Algorithm 1, the full pseudo-code for Eik-HiQRL, covering both the high-level planning and low-level control modules. This includes training routines for the quasimetric model, value functions, and policies. We then summarize in Table 3 the key hyperparameters used in our experiments. These details reflect the final configuration that achieved the best performance across our benchmarks. For further information and code, please refer to our GitHub repository.

---

**Algorithm 1** Eikonal-Constrained Hierarchical Quasimetric RL (Eik-HiQRL)

---

**Input:** Offline dataset $\mathcal{D}$, high-level quasimetric model $d_{\boldsymbol{\theta}}$, low-level value function $V_{\boldsymbol{\theta}_V}$, target value function $V_{\bar{\boldsymbol{\theta}}_V}$, goal representation network $\phi_{\boldsymbol{\theta}_\phi}$, high-level policy $\pi^h_{\boldsymbol{\theta}_h}$, low-level policy $\pi^l_{\boldsymbol{\theta}_l}$, expectile factor $\iota$, discount factor $\gamma$, inverse temperature parameter $\beta$, learning rates $\alpha_d, \alpha_V, \alpha_h, \alpha_l$, target update rate $\tau$

**while** not converged **do**

    $(s_t, s_{t+1}, g) \sim \mathcal{D}$

    Update $d_{\boldsymbol{\theta}}$ optimizing the problem in (22) with learning rate $\alpha_d$

    Update $V_{\boldsymbol{\theta}_V}$ and $\phi_{\boldsymbol{\theta}_\phi}$ minimizing $\mathcal{L}_V(\boldsymbol{\theta}_V, \boldsymbol{\theta}_\phi)$ in (18) with learning rate $\alpha_V$

    $\bar{\boldsymbol{\theta}}_V \leftarrow (1 - \tau)\bar{\boldsymbol{\theta}}_V + \tau\boldsymbol{\theta}_V$

**end while**

**while** not converged **do**

    $(s_t, s_{t+k}, g) \sim \mathcal{D}$

    Update $\pi^h_{\boldsymbol{\theta}_h}$ maximizing $J_{\pi^h}(\boldsymbol{\theta}_h)$ in (19) with learning rate $\alpha_h$

**end while**

**while** not converged **do**

    $(s_t, a_t, s_{t+1}, s_{t+k}) \sim \mathcal{D}$

    Update $\pi^l_{\boldsymbol{\theta}_l}$ maximizing $J_{\pi^l}(\boldsymbol{\theta}_l)$ in (20) with learning rate $\alpha_l$

**end while**

---

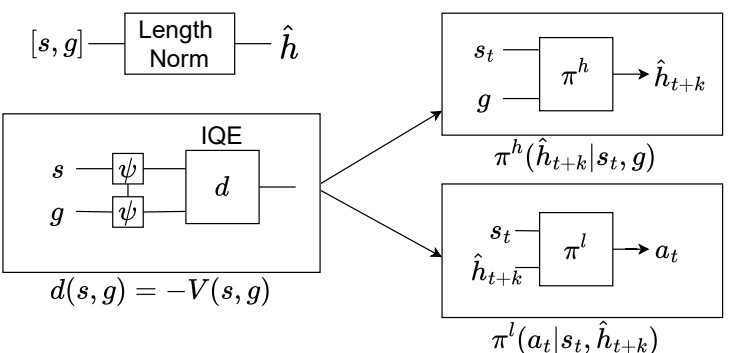

Figure 8: Diagram for the the hierarchical Eik-QRL, Eik-QRL-$\lambda$ and QRL:hi used for the experiments in Table 5 and Table 6.

## D    EIK-QRL, EIK-QRL-$\lambda$ AND QRL:HI

In this section, we describe the hierarchical variants Eik-QRL, Eik-QRL-$\lambda$ and QRL:hi, which maintain a single value function but incorporate hierarchical actors that operate over subgoals. Introducing hierarchical policies requires additional care, primarily due to the need for subgoal encoding to enable effective high-level planning. An illustrative diagram of this hierarchical architecture is provided in Fig. 8.

Each of these variants employs a single quasimetric value function trained to optimize a different objective: QRL:hi minimizes (1), Eik-QRL minimizes (22), and Eik-QRL-$\lambda$ minimizes the constrained form (21). Subgoal representations are implemented using length normalization, and both levels of the hierarchical actor are trained via advantage-weighted regression using the objectives in (19) and (20). The corresponding advantages are computed as

$$\tilde{A}^h(s_t, s_{t+k}, g) = d_{\boldsymbol{\theta}}(s_t, g) - d_{\boldsymbol{\theta}}(s_{t+k}, g), \quad \tilde{A}^l(s_t, a_t, s_{t+k}) = d_{\boldsymbol{\theta}}(s_t, s_{t+k}) - d_{\boldsymbol{\theta}}(s_{t+1}, s_{t+k}).$$

Note that when Eik-QRL is not paired with a hierarchical actor, it follows the standard QRL pipeline described in the preliminaries (Fig. 1 in Section 3), optimizing the unconstrained objective in (22), while Eik-QRL-$\lambda$ optimizes the constrained formulation in (21). We denote these flat variants as Eik-QRL:flat and Eik-QRL-$\lambda$:flat respectively and experimental results for these versions are reported in Table 6.

## E    OFFLINE GCRL ENVIRONMENTS USED IN SECTION 6

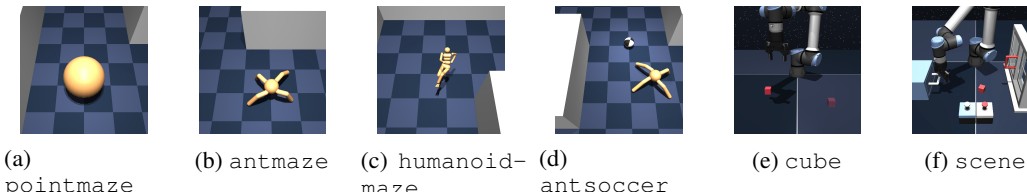

(a)
pointmaze
    (b) antmaze
    (c) humanoid-
maze
    (d)
antsoccer
    (e) cube
    (f) scene

Figure 9: Environments from OGbench (Park et al., 2024a) used in our experiments in Section 6.

In the following, we provide a brief description of the agents used in our experiments (Fig. 9). Our goal is to highlight to what extent their dynamics comply with the regularity assumptions in Section 4. From an experimental standpoint, two assumptions are particularly relevant: Assumption 4.2, which requires Lipschitz continuity of the dynamics, and Assumption 4.4, which prescribes Lipschitz continuity of the optimal goal-conditioned value function.

**pointmaze**   This task involves position control of a 2-D point mass. The state dimension for this agent is 2 (the $(x, y)$ coordinates), and the action space is also 2-dimensional. The dynamics in this environment are isotropic and therefore Lipschitz continuous, so Assumption 4.2 holds. Moreover, in this setting Assumption 4.2 is a sufficient condition for Assumption 4.4, and thus both assumptions are satisfied.

**antmaze**   This task involves controlling a quadrupedal ant agent with 8 degrees of freedom. The state dimension for this agent is 29, and the action dimension is 8. The dynamics involve intermittent contact between the feet and the ground for locomotion, so Assumption 4.2 is not strictly satisfied. However, due to the absence of explicit categorical variables in the state space and external manipulable objects, we expect the optimal goal-conditioned value function to remain reasonably regular in practice, making Assumption 4.4 a plausible approximation.

**humanoidmaze**   This task involves controlling a humanoid agent with 21 degrees of freedom. The state dimension for this agent is 69, and the action dimension is 21. As in antmaze, the dynamics involve intermittent contact with the ground, so Assumption 4.2 does not strictly hold. Nonetheless, the absence of categorical state variables and external manipulable objects suggests that the optimal goal-conditioned value function is still relatively smooth, and Assumption 4.4 is again a reasonable approximation.

**antsoccer**   This task involves controlling an ant agent to dribble a soccer ball. The overall state dimension is 42 (29 for the ant and 13 for the ball: 7 position/orientation coordinates and 6 linear and angular velocity coordinates), and the action dimension is 8. The dynamics involve both ground contacts and interactions with a separate ball object, which introduce discontinuities and non-smooth transitions. Consequently, Assumption 4.2 does not hold in this setting. Moreover, due to these contact-rich interactions with an external object, we do not expect the optimal goal-conditioned value function to be globally Lipschitz, and Assumption 4.4 is generally violated.

**cube**   This task involves pick-and-place manipulation of cube blocks, where a robot arm must arrange cubes into designated configurations. The state dimension for this agent is 28, and the action dimension is 5. The dynamics involve contact-rich interactions between the end-effector and the cubes, which break Lipschitz continuity, so Assumption 4.2 does not hold. In addition, the presence of multiple interacting objects and discrete mode switches (e.g., grasp / no-grasp) leads to value function discontinuities, so Assumption 4.4 is also violated in practice.

**scene**   This task involves manipulating diverse everyday objects, such as a cube block, a window, a drawer, and two button locks, where pressing a button toggles the lock status of the corresponding object (the drawer or window). The state dimension for this agent is 40, and the action dimension is 5. As in cube, the dynamics are contact-rich, and button presses induce discrete mode switches, so Assumption 4.2 does not hold. The combination of external objects and categorical state variables (e.g., lock status) introduces sharp discontinuities in the optimal goal-conditioned value function, leading to violations of Assumption 4.4.

## F   OFFLINE GCRL BASELINES USED IN SECTION 6

Table 4: Structural comparison of the considered goal-conditioned RL methods. A checkmark (✓) indicates that the corresponding component is present in the method, while a cross (✗) indicates its absence.

|  | Eik-HiQRL | Eik-QRL | QRL | HIQL | Eik-HIQL | CRL |
|---|---|---|---|---|---|---|
| Quasimetric Value | ✓ | ✓ | ✓ | ✗ | ✗ | ✗ |
| Eikonal Regularization | ✓ | ✓ | ✗ | ✗ | ✓ | ✗ |
| Hierarchical Actor | ✓ | ✓ | ✗ | ✓ | ✓ | ✗ |
| Hierarchical Value | ✓ | ✗ | ✗ | ✗ | ✗ | ✗ |

Table 4 summarizes the main structural differences among Eik-QRL (8), QRL (Wang et al., 2023), HIQL (Park et al., 2024b), Eik-HIQL (Giammarino et al., 2025), CRL (Eysenbach et al., 2022) and Eik-HiQRL (Fig. 2).

QRL learns a quasimetric goal-conditioned value function without any PDE-based constraints. Eik-QRL builds on QRL by adding Eikonal constraints, which provides similar theoretical guarantees under our regularity assumptions and a smoother value landscape, while also introducing a hierarchical actor. HIQL, in contrast, uses a hierarchical actor but learns a standard TD-based value (rather than a quasimetric) and does not impose any PDE constraints. Eik-HIQL augments HIQL with Eikonal regularization, yet it maintains a non-quasimetric value and does not decompose the value hierarchically. Our Eik-HiQRL combines the strengths of these approaches: it uses a quasimetric value constrained by the Eikonal PDE at the high level, together with both a hierarchical actor and a hierarchical value structure. This design preserves the theoretical benefits of Eik-QRL where its assumptions are most plausible, while mitigating its practical limitations through hierarchy and enabling improved scaling to long-horizon tasks. Finally, CRL (Eysenbach et al., 2022) is a substantially different method compared to the previous algorithms which learns a critic based on contrastive learning principles. We include it here for the sake of completeness.

## G  LEARNING CURVES FOR EXPERIMENTS IN SECTION 6

In this section, we present the full set of learning curves corresponding to the experiments in the main text. Fig. 11, 12, 13, 14 show success rate and collision rate as functions of training steps for the results summarized in Table 1, covering respectively Eik-HiQRL, Eik-QRL, HJB-QRL, and QRL.

Fig. 15 and 16 report the learning curves for the `antsoccer` and `manipulation` experiments in Table 2.

All experiments were conducted on a single NVIDIA RTX 3090 GPU (24 GB VRAM), using a local server equipped with a 12th Gen Intel i7-12700F CPU, 32 GB RAM. No cloud services or compute clusters were used. Each individual experimental run required approximately 4 hours of compute time on the GPU.

A natural concern is the additional computational overhead introduced by the Eikonal constraint, which requires automatic differentiation with respect to the inputs of the value network. In our implementation, this amounts to one extra backward pass per minibatch, which can be fully parallelized on modern hardware. To quantify the overhead, we compare wall-clock training time for Eik-QRL and QRL over 100k gradient steps with batch size 1024. The average time for 100 training steps (mean $\pm$ standard deviation across training) is $6.37 \pm 5.69$ ms for Eik-QRL and $6.20 \pm 5.55$ ms for QRL, i.e., a difference of less than 3%, well within the observed variability. In practice, we also observe comparable memory usage for the two methods. These results indicate that the Eikonal regularization has negligible impact on training efficiency in our setting (Cf. Fig. 10).

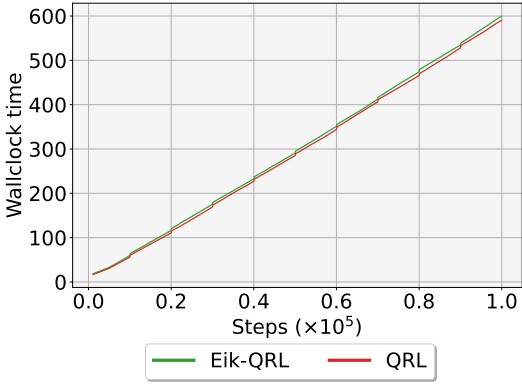

Figure 10: Wall-clock training time as a function of training steps for Eik-QRL and QRL over 100k gradient steps with batch size 1024. The curves closely overlap, indicating that the Eikonal term in Eik-QRL introduces negligible computational overhead compared to QRL.

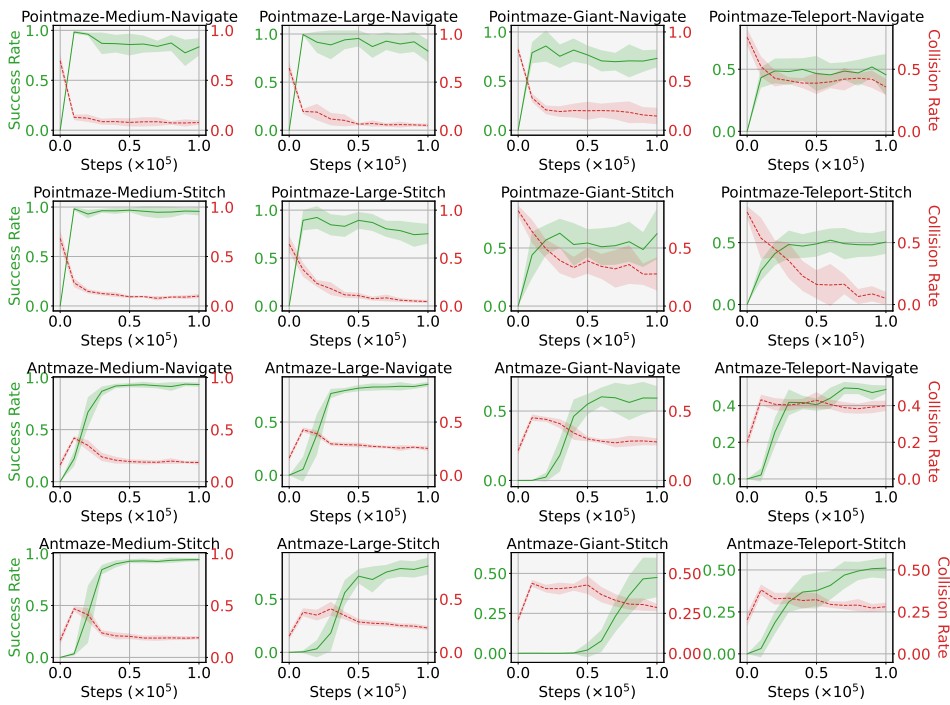

Figure 11: Learning curves for Eik-HiQRL in the experiments in Table 1. Plots show the average success rate and collision rate per evaluation across seeds as a function of training steps.

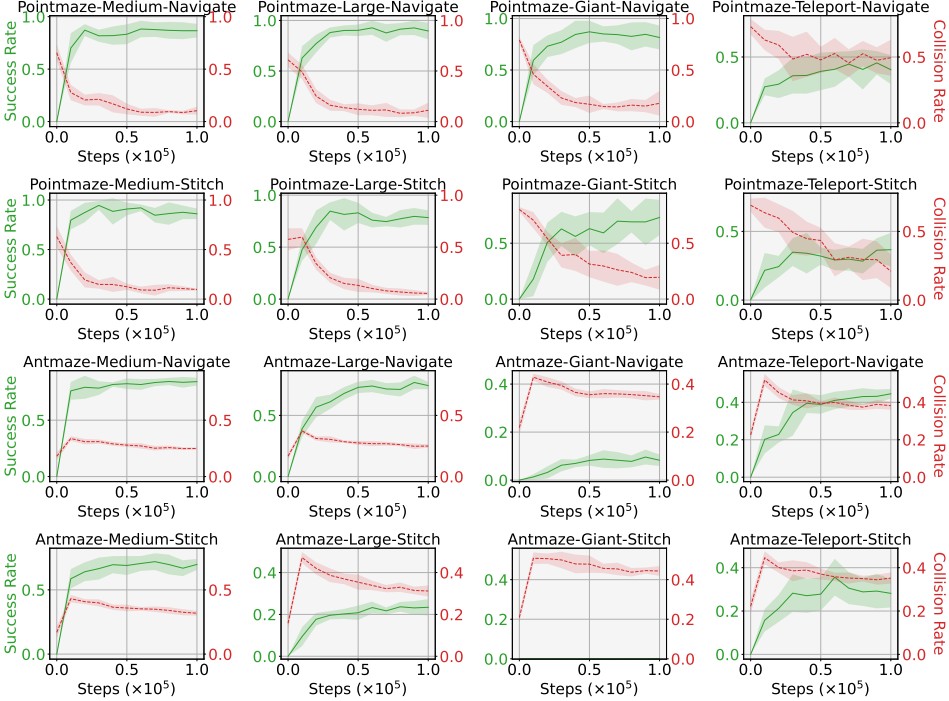

Figure 12: Learning curves for Eik-QRL in the experiments in Table 1. Plots show the average success rate and collision rate per evaluation across seeds as a function of training steps.

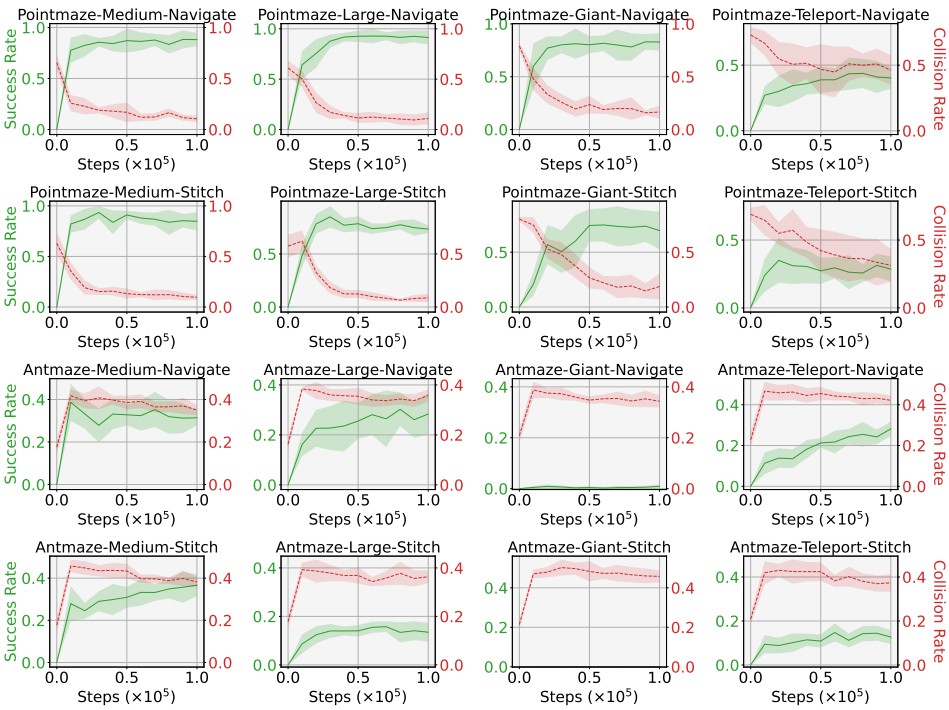

Figure 13: Learning curves for HJB-QRL in the experiments in Table 1. Plots show the average success rate and collision rate per evaluation across seeds as a function of training steps.

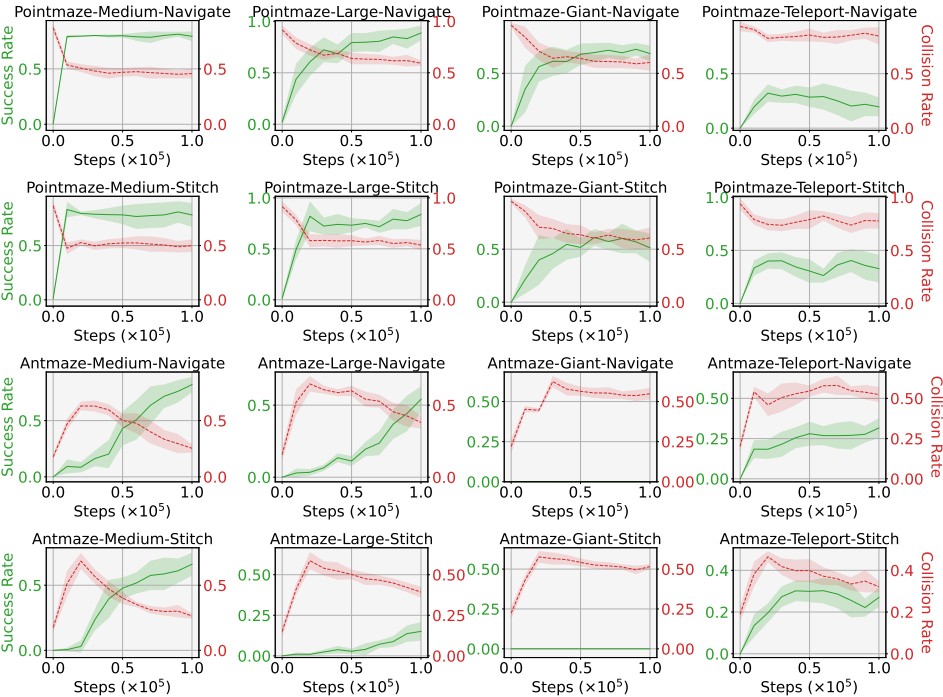

Figure 14: Learning curves for QRL in the experiments in Table 1. Plots show the average success rate and collision rate per evaluation across seeds as a function of training steps.

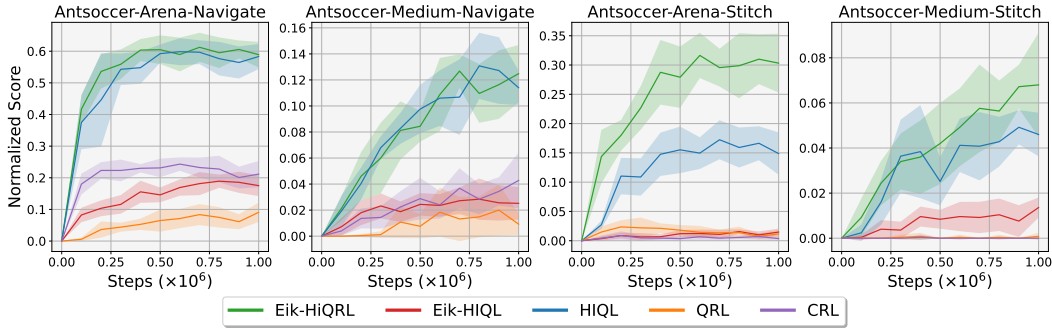

Figure 15: Learning curves for the `antsoccer` experiments in Table 2. Plots show the average success percentage per evaluation across seeds as a function of training steps.

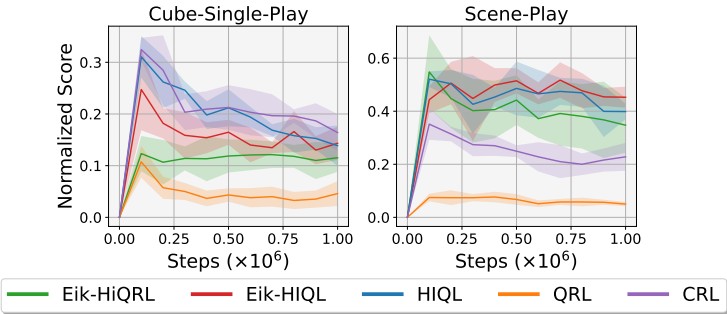

Figure 16: Learning curves for the `manipulation` experiments in Table 2. Plots show the average success percentage per evaluation across seeds as a function of training steps.

## H    ADDITIONAL EXPERIMENTS

In the following, we present additional experi-
ments and ablations that complement the results in the main paper. Section H.1 reports a comparison analogous to Table 1, but with QRL replaced by QRL:hi as introduced in Section D.

Section H.2 presents an ablation over actors and optimization losses to validate our design choices.

Section H.3 provides a full comparison between Eik-HiQRL and Offline GCRL baselines, extending the experiments from the main text.

Section H.4 reports an ablation between Eik-HiQRL and HiQRL, contrasting the Eikonal-constrained optimization pipeline in Eq. (8) with the standard QRL formulation in Eq. (1) within our hierarchical framework.

Section H.5 investigates the trajectory-free setting and presents an empirical study demonstrating the potential of Eik-QRL in this regime.

Finally, Section H.6 concludes this Section by showing that Eik-QRL can also be applied in the online RL setting.

### H.1    COMPARISON WITH QRL:HI

In the following, we report a comparison analogous to Table 1, but with QRL replaced by QRL:hi as presented in Appendix D. We include these experiments for completeness and to better disentangle the contributions of hierarchy from those of the PDE-based constraints. The results, summarized in Table 5, emphasize the strong regularizing effect of PDE-based constraints, which yield tangible improvements in the `pointmaze-giant` environment under both the `navigate` and `stitch` data regimes. In the `antmaze` setting, the considerations discussed in the main text remain valid

Table 5: Summary of the comparison among the QRL formulations. All agents are trained for $10^5$ steps using 10 seeds and evaluated every $10^4$ steps. We report the mean and standard deviation across seeds for the last evaluation. For each seed, evaluations are conducted over 5 different random goals, with the learned policy tested for 50 episodes per goal. For success rate ($\mathcal{R}$), results within 95% of the best value are written in **bold**. For collision avoidance ($\kappa$) the lowest average is colored.

| Environment | Dataset | Dim | Eik-HiQRL $\mathcal{R}$ ($\uparrow$) | Eik-HiQRL $\kappa$ ($\downarrow$) | Eik-QRL $\mathcal{R}$ ($\uparrow$) | Eik-QRL $\kappa$ ($\downarrow$) | HJB-QRL $\mathcal{R}$ ($\uparrow$) | HJB-QRL $\kappa$ ($\downarrow$) | QRL:hi $\mathcal{R}$ ($\uparrow$) | QRL:hi $\kappa$ ($\downarrow$) |
|---|---|---|---|---|---|---|---|---|---|---|
| pointmaze | navigate | medium | $\mathbf{83 \pm 9}$ | $8 \pm 2$ | $\mathbf{87 \pm 7}$ | $10 \pm 4$ | $\mathbf{88 \pm 6}$ | $10 \pm 3$ | $\mathbf{85 \pm 6}$ | $13 \pm 4$ |
| | | large | $82 \pm 12$ | $5 \pm 2$ | $\mathbf{90 \pm 8}$ | $11 \pm 8$ | $\mathbf{91 \pm 7}$ | $11 \pm 6$ | $\mathbf{94 \pm 7}$ | $7 \pm 2$ |
| | | giant | $73 \pm 9$ | $14 \pm 8$ | $\mathbf{82 \pm 12}$ | $18 \pm 12$ | $\mathbf{83 \pm 8}$ | $17 \pm 6$ | $69 \pm 13$ | $20 \pm 5$ |
| | | teleport | $\mathbf{46 \pm 17}$ | $36 \pm 6$ | $40 \pm 11$ | $49 \pm 14$ | $40 \pm 8$ | $46 \pm 11$ | $23 \pm 8$ | $58 \pm 11$ |
| | stitch | medium | $\mathbf{96 \pm 4}$ | $10 \pm 2$ | $86 \pm 5$ | $10 \pm 1$ | $85 \pm 9$ | $10 \pm 2$ | $89 \pm 9$ | $11 \pm 2$ |
| | | large | $75 \pm 10$ | $5 \pm 2$ | $78 \pm 7$ | $5 \pm 1$ | $73 \pm 4$ | $9 \pm 4$ | $\mathbf{91 \pm 6}$ | $11 \pm 5$ |
| | | giant | $62 \pm 22$ | $28 \pm 14$ | $\mathbf{73 \pm 16}$ | $19 \pm 11$ | $\mathbf{70 \pm 17}$ | $19 \pm 12$ | $54 \pm 12$ | $33 \pm 14$ |
| | | teleport | $\mathbf{50 \pm 10}$ | $5 \pm 5$ | $37 \pm 12$ | $21 \pm 12$ | $29 \pm 9$ | $31 \pm 12$ | $28 \pm 6$ | $45 \pm 15$ |
| antmaze | navigate | medium | $\mathbf{93 \pm 2}$ | $18 \pm 2$ | $84 \pm 3$ | $25 \pm 1$ | $31 \pm 4$ | $35 \pm 3$ | $86 \pm 3$ | $24 \pm 2$ |
| | | large | $\mathbf{86 \pm 2}$ | $25 \pm 2$ | $74 \pm 3$ | $25 \pm 1$ | $28 \pm 8$ | $36 \pm 2$ | $\mathbf{85 \pm 2}$ | $24 \pm 1$ |
| | | giant | $\mathbf{59 \pm 7}$ | $28 \pm 3$ | $8 \pm 2$ | $35 \pm 1$ | $0 \pm 0$ | $34 \pm 2$ | $23 \pm 2$ | $32 \pm 1$ |
| | | teleport | $\mathbf{49 \pm 2}$ | $40 \pm 3$ | $45 \pm 3$ | $38 \pm 2$ | $28 \pm 4$ | $42 \pm 2$ | $40 \pm 4$ | $43 \pm 2$ |
| | stitch | medium | $\mathbf{94 \pm 1}$ | $19 \pm 2$ | $70 \pm 4$ | $32 \pm 2$ | $37 \pm 5$ | $38 \pm 2$ | $83 \pm 5$ | $31 \pm 3$ |
| | | large | $\mathbf{81 \pm 8}$ | $23 \pm 2$ | $23 \pm 4$ | $31 \pm 3$ | $13 \pm 4$ | $36 \pm 3$ | $36 \pm 4$ | $31 \pm 0$ |
| | | giant | $\mathbf{47 \pm 12}$ | $29 \pm 2$ | $0 \pm 0$ | $44 \pm 2$ | $0 \pm 0$ | $46 \pm 3$ | $0 \pm 0$ | $47 \pm 2$ |
| | | teleport | $\mathbf{51 \pm 6}$ | $28 \pm 2$ | $28 \pm 7$ | $35 \pm 3$ | $13 \pm 3$ | $37 \pm 4$ | $38 \pm 3$ | $34 \pm 3$ |

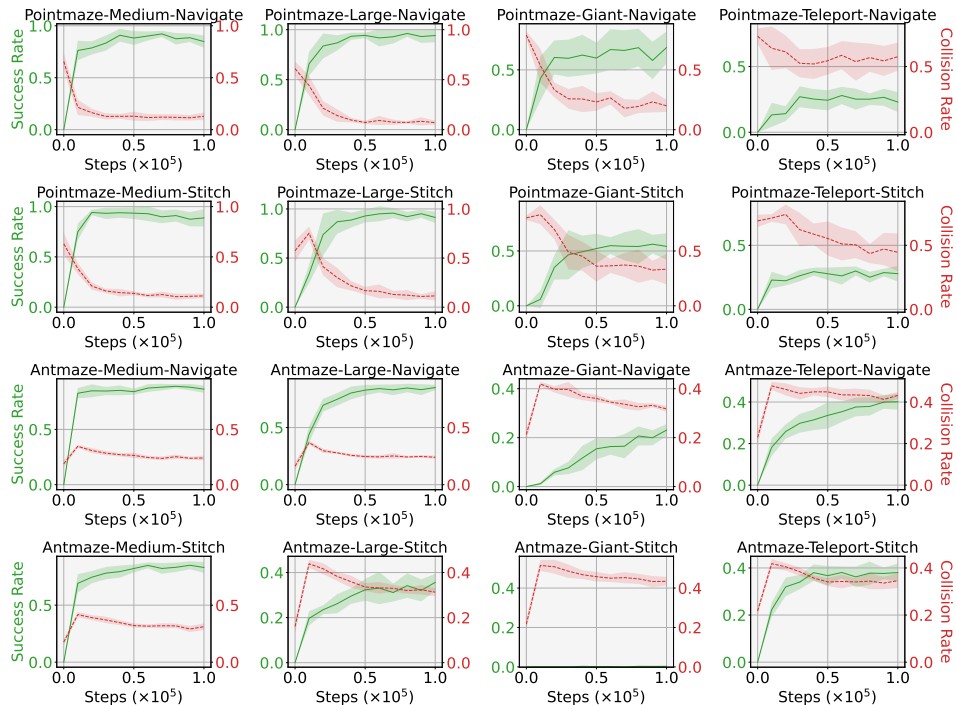

Figure 17: Learning curves for QRL:hi in the experiments in Table 5. Plots show the average success rate and collision rate per evaluation across seeds as a function of training steps.

here as well. The corresponding learning curves are shown in Fig. 11, 12, 13, and 17 for Eik-HiQRL, Eik-QRL, HJB-QRL, and QRL:hi, respectively.

## H.2 ADDITIONAL ABLATION

This section reports an additional ablation comparing quasimetric-based algorithms across actor types (flat vs. hierarchical) and optimization variants (constrained in Eq. 21 vs. unconstrained in Eq. 22). We include this experiment to further justify our design choices and to provide a more compact comparison across algorithms. The results, summarized in Table 6, confirm the benefits of combining

Table 6: Additional ablation. All agents are trained for $10^5$ steps using 10 seeds and evaluated every $10^4$ steps. We report the mean and standard deviation across seeds for the best evaluation across training. For each seed, evaluations are conducted over 5 different random goals, with the learned policy tested for 50 episodes per goal. Results within $95\%$ of the best value are written in **bold**.

| Env | Dataset | Dimension | Eik-HiQRL | QRL | QRL:hi | Eik-QRL:flat | Eik-QRL-$\lambda$:flat | Eik-QRL | Eik-QRL-$\lambda$ |
|---|---|---|---|---|---|---|---|---|---|
| pointmaze | navigate | medium | **99 ± 1** | 83 ± 3 | 93 ± 4 | 92 ± 12 | 84 ± 6 | 90 ± 5 | 78 ± 15 |
| | | large | **99 ± 1** | 90 ± 6 | **95 ± 4** | 76 ± 15 | **98 ± 6** | **94 ± 4** | 87 ± 14 |
| | | giant | **89 ± 8** | 72 ± 7 | 73 ± 10 | 63 ± 21 | 68 ± 10 | **90 ± 6** | 57 ± 13 |
| | | teleport | **53 ± 6** | 34 ± 7 | 28 ± 6 | 25 ± 18 | 14 ± 8 | 49 ± 12 | 27 ± 9 |
| | stitch | medium | **99 ± 2** | 80 ± 10 | **96 ± 4** | 93 ± 8 | **98 ± 4** | 94 ± 7 | 85 ± 7 |
| | | large | **95 ± 8** | 85 ± 11 | **96 ± 3** | 70 ± 12 | **96 ± 7** | 87 ± 9 | 75 ± 13 |
| | | giant | 57 ± 14 | 56 ± 9 | 67 ± 8 | 45 ± 18 | 58 ± 15 | **76 ± 11** | 32 ± 24 |
| | | teleport | **52 ± 8** | 42 ± 6 | 30 ± 4 | 39 ± 8 | 29 ± 12 | 32 ± 10 | 35 ± 9 |
| antmaze | navigate | medium | **94 ± 2** | 81 ± 5 | **89 ± 3** | 51 ± 6 | 10 ± 6 | 85 ± 3 | 69 ± 4 |
| | | large | **84 ± 4** | 62 ± 9 | **84 ± 7** | 9 ± 5 | 5 ± 2 | 73 ± 4 | 68 ± 10 |
| | | giant | **61 ± 13** | 0 ± 0 | 19 ± 5 | 0 ± 0 | 0 ± 0 | 9 ± 3 | 6 ± 2 |
| | | teleport | **52 ± 2** | 30 ± 8 | 39 ± 3 | 32 ± 4 | 29 ± 4 | 43 ± 6 | 43 ± 5 |
| | stitch | medium | **95 ± 2** | 68 ± 6 | 88 ± 2 | 8 ± 4 | 1 ± 1 | 72 ± 6 | 67 ± 8 |
| | | large | **83 ± 3** | 14 ± 5 | 35 ± 5 | 5 ± 4 | 2 ± 2 | 25 ± 3 | 24 ± 7 |
| | | giant | **51 ± 10** | 0 ± 0 | 0 ± 0 | 0 ± 0 | 0 ± 0 | 0 ± 0 | 0 ± 0 |
| | | teleport | **53 ± 5** | 31 ± 4 | 43 ± 6 | 14 ± 4 | 17 ± 5 | 33 ± 5 | 38 ± 8 |

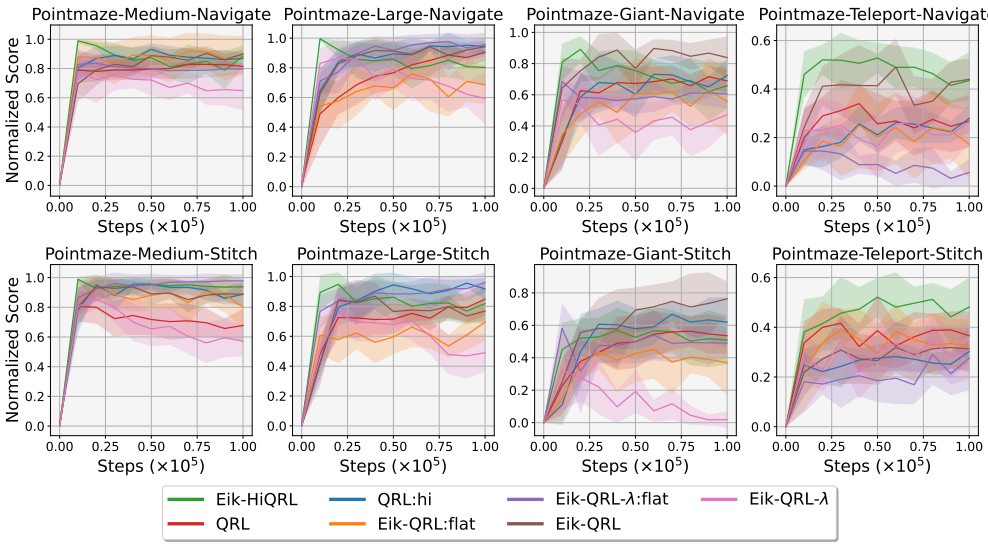

Figure 18: Learning curves for the `pointmaze` experiments in Table 6. Plots show the average success percentage per evaluation across seeds as a function of training steps.

hierarchical actors with PDE-based constraints. The corresponding learning curves are shown in Fig. 18 and Fig. 19.

## H.3 FULL COMPARISON EIK-HIQRL VS BASELINES

This section provides a thorough evaluation of Eik-HiQRL against the Offline GCRL baselines introduced in the main text, complementing and summarizing the earlier comparisons. The experiments are reported in Table 7, and the corresponding learning curves are shown in Fig. 20, 21, 22, 4, 15, and 16 for the `pointmaze`, `antmaze`, `antmaze-explore`, `humanoidmaze`, `antsoccer`, and `manipulation` environments, respectively. Consistent with the conclusions of the main text, Eik-HiQRL comfortably outperforms or matches all baselines across tasks. As also noted, performance gains are smaller in the `manipulation` setting, which represents an interesting testbed for future research.

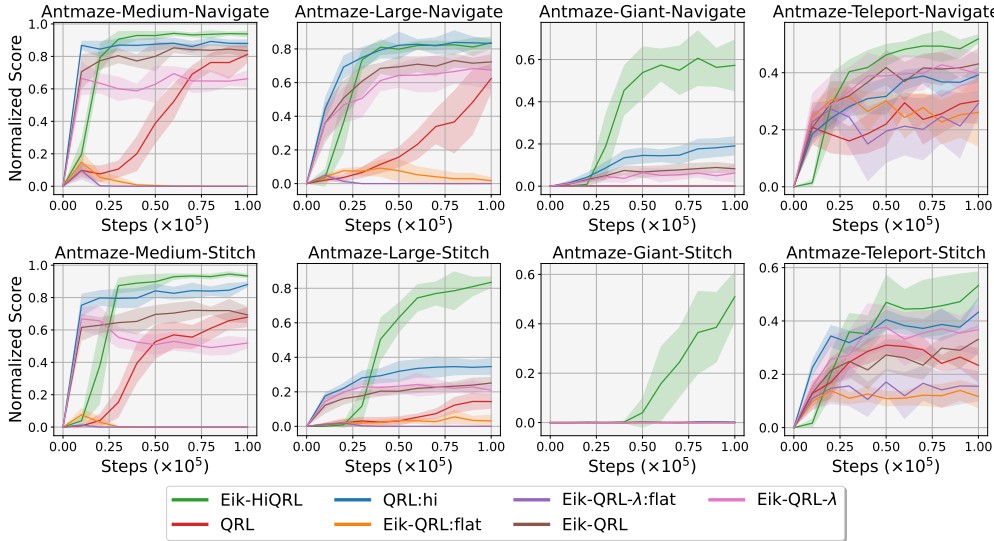

Figure 19: Learning curves for the `antmaze` experiments in Table 6. Plots show the average success percentage per evaluation across seeds as a function of training steps.

Table 7: Summary of our experimental results. All agents are trained for $10^6$ steps using 10 seeds and evaluated every $10^5$ steps. We report the mean and standard deviation across seeds for the best evaluation. For each seed, evaluations are conducted over 5 different random goals, as designed in Park et al. (2024a), with the learned policy tested for 50 episodes per goal. Results within 95% of the best value are written in **bold**.

| Environment | Dataset | Dimension | Eik-HiQRL (ours) | Eik-HIQL | HIQL | QRL | CRL |
|---|---|---|---|---|---|---|---|
| pointmaze | navigate | medium | **99 ± 1** | **93 ± 5** | 92 ± 2 | 83 ± 3 | 54 ± 19 |
| | | large | **99 ± 1** | 83 ± 9 | 49 ± 13 | 90 ± 5 | 56 ± 9 |
| | | giant | **89 ± 8** | 79 ± 13 | 7 ± 8 | 72 ± 7 | 37 ± 17 |
| | | teleport | **53 ± 6** | 47 ± 10 | 29 ± 7 | 34 ± 7 | **50 ± 5** |
| | stitch | medium | **99 ± 2** | **96 ± 3** | 76 ± 8 | 80 ± 10 | 3 ± 5 |
| | | large | **95 ± 8** | 73 ± 6 | 19 ± 7 | 85 ± 11 | 4 ± 6 |
| | | giant | 57 ± 14 | 22 ± 10 | 1 ± 4 | **56 ± 9** | 0 ± 0 |
| | | teleport | **52 ± 8** | 43 ± 9 | 38 ± 5 | 42 ± 6 | 12 ± 6 |
| antmaze | navigate | medium | **95 ± 2** | **95 ± 1** | **96 ± 1** | 87 ± 5 | **94 ± 2** |
| | | large | **86 ± 2** | **86 ± 2** | **90 ± 6** | 80 ± 5 | **86 ± 3** |
| | | giant | **66 ± 1** | **67 ± 5** | **69 ± 3** | 14 ± 6 | 18 ± 4 |
| | | teleport | **53 ± 4** | **52 ± 4** | 43 ± 3 | 39 ± 4 | **55 ± 4** |
| | stitch | medium | **94 ± 2** | **94 ± 2** | **95 ± 14** | 68 ± 6 | 54 ± 8 |
| | | large | **88 ± 3** | **84 ± 3** | 74 ± 6 | 24 ± 5 | 13 ± 4 |
| | | giant | **61 ± 9** | 48 ± 11 | 3 ± 3 | 2 ± 2 | 0 ± 0 |
| | | teleport | **60 ± 3** | 47 ± 2 | 35 ± 3 | 29 ± 6 | 34 ± 4 |
| | explore | medium | **49 ± 20** | 43 ± 15 | 33 ± 15 | 5 ± 4 | 4 ± 2 |
| | | large | **16 ± 11** | **13 ± 1** | 6 ± 7 | 0 ± 0 | 0 ± 0 |
| | | teleport | 7 ± 4 | 15 ± 10 | **45 ± 5** | 2 ± 2 | 22 ± 5 |
| humanoidmaze | navigate | medium | **91 ± 2** | 86 ± 2 | **90 ± 3** | 22 ± 2 | 61 ± 4 |
| | | large | **74 ± 5** | 64 ± 7 | 50 ± 4 | 7 ± 3 | 22 ± 9 |
| | | giant | **83 ± 6** | 68 ± 5 | 18 ± 5 | 1 ± 1 | 4 ± 2 |
| | stitch | medium | **85 ± 3** | 79 ± 2 | **88 ± 3** | 22 ± 4 | 40 ± 7 |
| | | large | **63 ± 6** | 29 ± 7 | 28 ± 2 | 3 ± 1 | 4 ± 2 |
| | | giant | **69 ± 5** | 19 ± 5 | 3 ± 1 | 0 ± 0 | 0 ± 0 |
| antsoccer | navigate | arena | **61 ± 5** | 19 ± 2 | **60 ± 4** | 10 ± 3 | 24 ± 2 |
| | | medium | **13 ± 1** | 3 ± 2 | **13 ± 3** | 2 ± 2 | 4 ± 2 |
| | stitch | arena | **32 ± 4** | 2 ± 0 | 17 ± 3 | 2 ± 1 | 1 ± 1 |
| | | medium | **7 ± 2** | 1 ± 0 | 5 ± 1 | 0 ± 0 | 0 ± 0 |
| manipulation | cube-single-play | | 12 ± 3 | 25 ± 1 | **31 ± 4** | 11 ± 3 | **32 ± 2** |
| | scene-play | | **55 ± 14** | **52 ± 7** | **52 ± 3** | 8 ± 2 | 35 ± 6 |

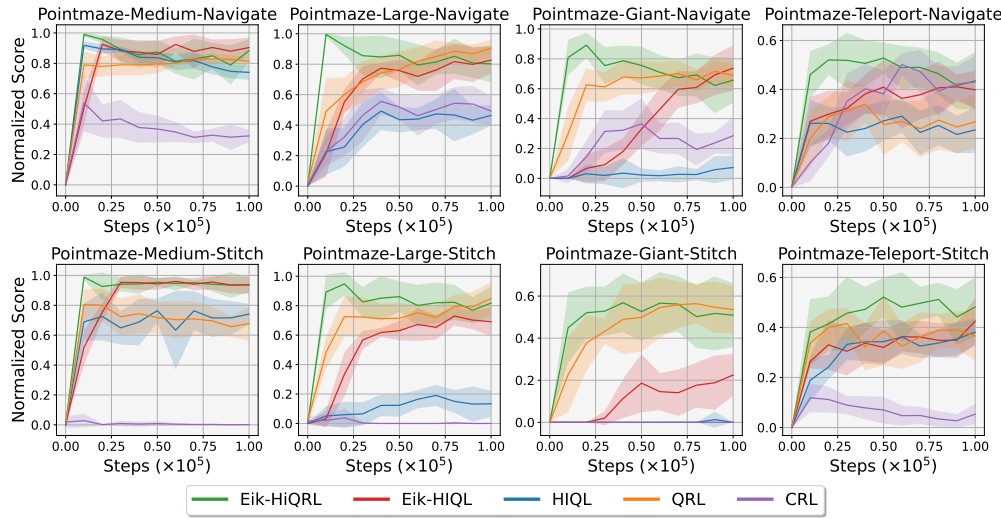

Figure 20: Learning curves for the `pointmaze` experiments in Table 7. Plots show the average success percentage per evaluation across seeds as a function of training steps.

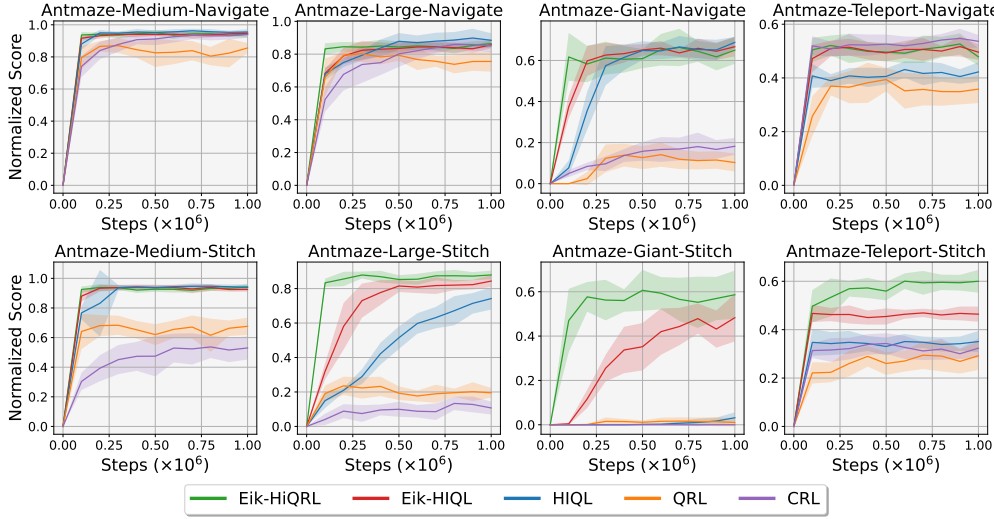

Figure 21: Learning curves for the `antmaze` experiments in Table 7. Plots show the average success percentage per evaluation across seeds as a function of training steps.

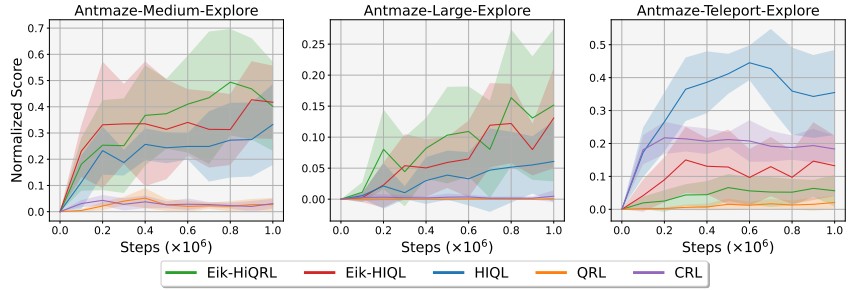

Figure 22: Learning curves for the `antmaze-explore` experiments in Table 7. Plots show the average success percentage per evaluation across seeds as a function of training steps.

Table 8: Second ablation. Training and evaluation procedures follow Table 7. Results within $95\%$ of the best value are written in **bold**.

| Environment | Dataset | Dim | Eik-HiQRL | HiQRL |
|---|---|---|---|---|
| antmaze | navigate | medium | $\mathbf{95 \pm 2}$ | $93 \pm 2$ |
| | | large | $86 \pm 2$ | $\mathbf{87 \pm 3}$ |
| | | giant | $66 \pm 1$ | $\mathbf{67 \pm 7}$ |
| | | teleport | $\mathbf{53 \pm 4}$ | $42 \pm 4$ |
| | stitch | medium | $\mathbf{94 \pm 2}$ | $\mathbf{94 \pm 2}$ |
| | | large | $\mathbf{88 \pm 3}$ | $\mathbf{88 \pm 3}$ |
| | | giant | $\mathbf{61 \pm 9}$ | $52 \pm 9$ |
| | | teleport | $\mathbf{60 \pm 3}$ | $45 \pm 4$ |
| | explore | medium | $\mathbf{49 \pm 20}$ | $40 \pm 19$ |
| | | large | $\mathbf{16 \pm 11}$ | $13 \pm 13$ |
| | | teleport | $7 \pm 4$ | $\mathbf{17 \pm 11}$ |
| humanoidmaze | navigate | medium | $91 \pm 2$ | $\mathbf{92 \pm 2}$ |
| | | large | $\mathbf{74 \pm 5}$ | $\mathbf{74 \pm 3}$ |
| | | giant | $\mathbf{83 \pm 6}$ | $70 \pm 9$ |
| | stitch | medium | $85 \pm 3$ | $\mathbf{87 \pm 2}$ |
| | | large | $63 \pm 6$ | $\mathbf{67 \pm 7}$ |
| | | giant | $\mathbf{69 \pm 5}$ | $40 \pm 12$ |
| antsoccer | navigate | arena | $61 \pm 5$ | $\mathbf{65 \pm 3}$ |
| | | medium | $13 \pm 1$ | $\mathbf{14 \pm 3}$ |
| | stitch | arena | $\mathbf{32 \pm 4}$ | $30 \pm 4$ |
| | | medium | $7 \pm 2$ | $\mathbf{8 \pm 2}$ |

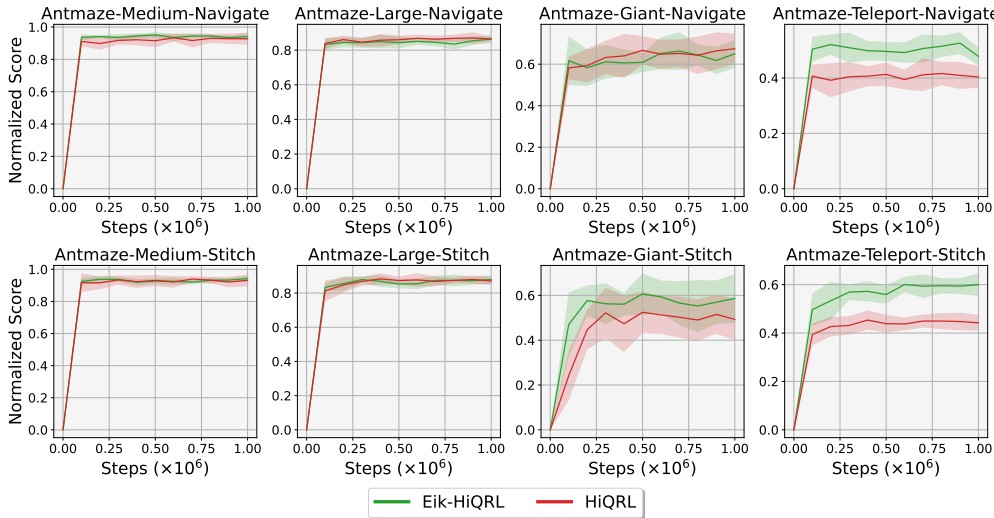

Figure 23: Learning curves for the `antmaze` experiments in Table 8. Plots show the average success percentage per evaluation across seeds as a function of training steps.

## H.4 COMPARISON EIK-HIQRL VS HIQRL

This section complements the ablation in Section H.2 and examines the impact of Eikonal constraints within our hierarchical framework, thereby closing the loop of our analysis. To this end, we compare Eik-HiQRL and HiQRL: the former employs the Eikonal-constrained optimization pipeline in Eq. 8 for high-level value function learning, while the latter relies on the standard QRL formulation in Eq. 1. Results are summarized in Table 8, with learning curves reported in Fig. 23, 24, 25, and 26. The conclusions align with those of the previous sections, once again showing substantial gains in large environments and in the `stitch` setting, underscoring the benefits of our PDE-based constraints.

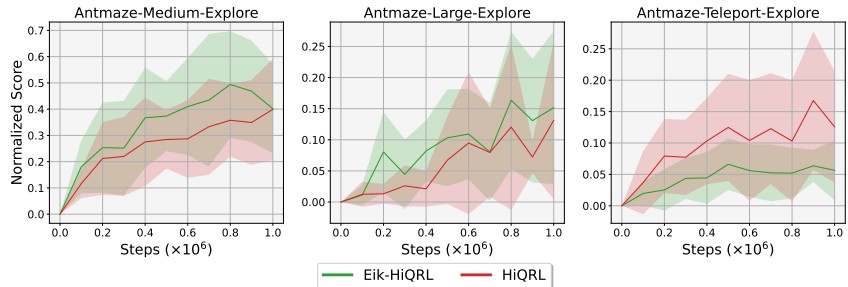

Figure 24: Learning curves for the `antmaze-explore` experiments in Table 8. Plots show the average success percentage per evaluation across seeds as a function of training steps.

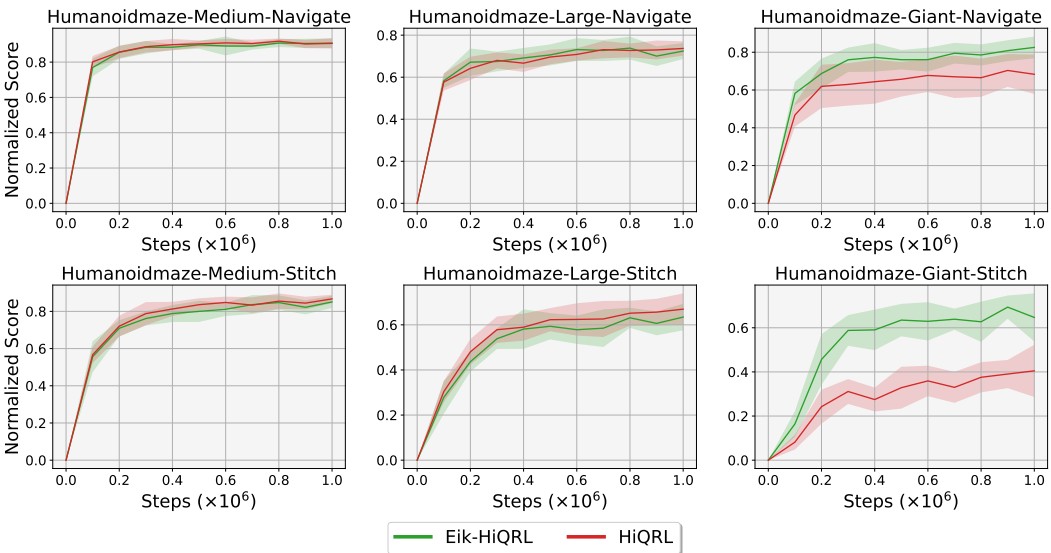

Figure 25: Learning curves for the `humanoidmaze` experiments in Table 8. Plots show the average success percentage per evaluation across seeds as a function of training steps.

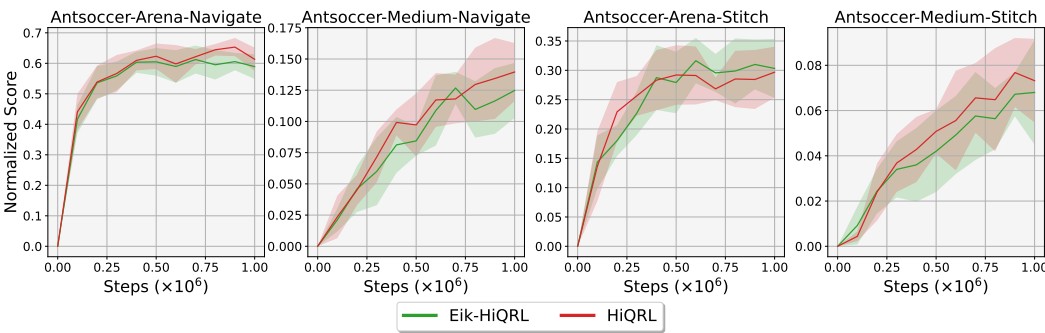

Figure 26: Learning curves for the `antsoccer` experiments in Table 8. Plots show the average success percentage per evaluation across seeds as a function of training steps.

## H.5 TRAJECTORY-FREE EXPERIMENTS

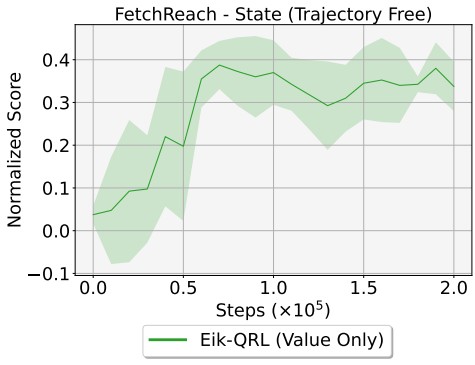
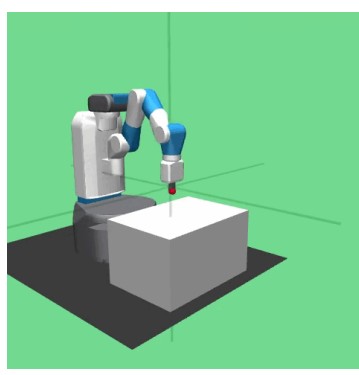

(a) Learning curve for **Eik-QRL** in a trajectory-free setting.

(b) `FetchReach` environment in Plappert et al. (2018).

Figure 27: Summary of the results in the fully trajectory-free setting. We train a goal-conditioned value function $V_{\boldsymbol{\theta}}(s, g) = -d_{\boldsymbol{\theta}}(s, g)$ using Eik-QRL as defined in Eq. (8). Training relies solely on random samples of end-effector $(x, y, z)$-coordinates from the robot workspace. The end-effector is then controlled via inverse kinematics, with its position updated by following the negative value gradient, i.e., $s_{t+1} = s_t - \eta \nabla_s V_{\boldsymbol{\theta}}(s_t, g)$ for a suitable stepsize $\eta > 0$. The corresponding learning curve is shown in Fig. 27a. We report mean and standard deviation over 8 random seeds.

To investigate the trajectory-free regime, we evaluate Eik-QRL on a robotic goal-reaching task based on the `FetchReach` environment (Plappert et al., 2018). The setup, illustrated in Fig. 27b, uses a 7-DoF Fetch mobile manipulator equipped with a two-finger parallel gripper; the underlying environment controls the robot in Cartesian space by specifying small displacements of the end-effector, while inverse kinematics are handled internally by the simulator. To make this setting fully trajectory-free, we restrict the state space to the end-effector Cartesian coordinates $(x, y, z)$, and randomly sample 100k end-effector positions from the robot workspace, aggregating them into a replay buffer without storing any transition trajectories.

Value learning is performed with Eik-QRL as in Eq. (8), where minibatches of random coordinates are independently sampled as state $s$ and goal $g$. Every 10k training steps, we evaluate the resulting controller over 50 rollouts. We do not train a separate policy network in this experiment; instead, control is obtained by directly updating the end-effector position using the gradient of the learned goal-conditioned value function,

$$s_{t+1} = s_t - \eta \nabla_s V_{\boldsymbol{\theta}}(s_t, g),$$

for a suitable stepsize $\eta > 0$ (we use $\eta = 1$ in this experiment). The robot arm is then commanded to the updated end-effector position via inverse kinematics.

Results for this experiment are reported in Fig. 27a and show that, despite the very weak form of supervision, the method reaches an average success rate of about $40\%$ after 100k training steps. We view this as an initial proof-of-concept demonstration of the potential of Eik-QRL in fully trajectory-free settings with smooth, approximately isotropic dynamics. Note that we do not perform any architecture or hyperparameter tuning in these experiments, but simply reuse the same configuration as in the previous tasks (see Table 3). Future work will focus on leveraging the learned goal-conditioned value function as initialization for subsequent value-learning or control schemes, effectively bootstrapping learning in more complex or data-scarce regimes.

## H.6 ONLINE RL EXPERIMENTS

In this section, we evaluate QRL, Eik-QRL, and Eik-HiQRL in an online GCRL setting on two continuous-control benchmarks: `Pointmaze` and `FetchReach`. For both domains, we use identical network architectures, hyperparameters, and training budgets across methods, and we report mean and standard deviation over 3 random seeds.

### H.6.1 POINTMAZE

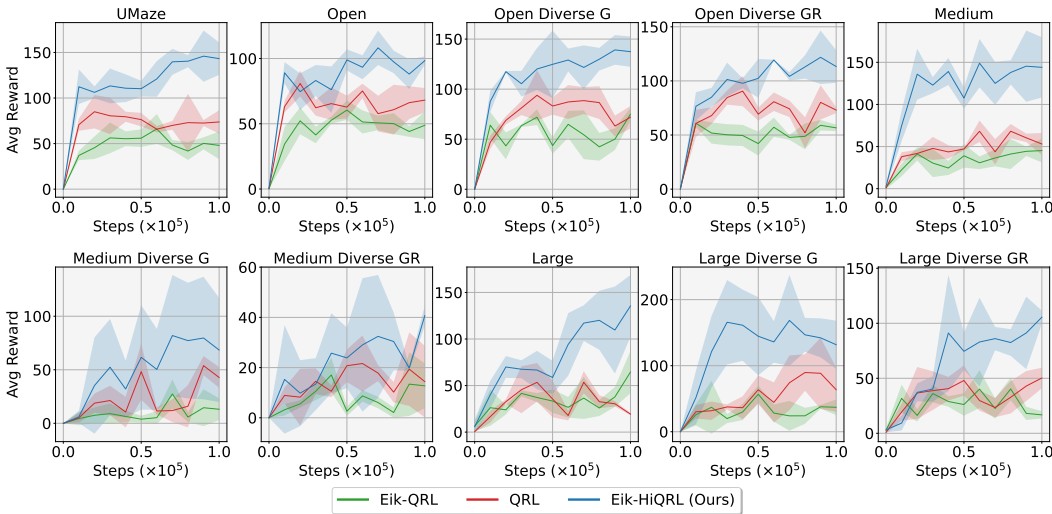

Figure 28: Online GCRL results on `Pointmaze`. We evaluate each algorithm every 10k steps over 50 evaluation episodes and report mean and standard deviation over 3 random seeds.

This `Pointmaze` environment consists of a 2-DoF point mass (a "ball") that is force-actuated in the Cartesian directions $x$ and $y$, and must reach a target goal within a closed maze (Fu et al., 2020). The environment can be instantiated with different maze layouts and increasing levels of difficulty, adapted from Fu et al. (2020). We consider multiple maze sizes and configurations, including single-goal and multi-goal variants, where the latter are referred to as *diverse* mazes. A further group of variations instantiates another class of diverse mazes in which both goal locations and initial agent states are randomly sampled at each reset. These environments share the same base identifiers as their default counterparts, with the suffix `Diverse_GR` (GR stands for *Goal and Reset*) appended.

Because the agent is controlled through forces, the underlying dynamics are closer to a double-integrator model in free space, but collisions with the maze walls and contact effects make the effective dynamics highly anisotropic and introduce strong non-smoothness in the value function. As a result, Assumption 4.2 and the induced Lipschitz regularity in Assumption 4.4 are not strictly satisfied. This setting therefore provides a stress test for Eik-QRL outside its ideal isotropic regime.

We compare QRL, Eik-QRL, and Eik-HiQRL on these online tasks in Fig. 28. All methods learn meaningful goal-reaching policies across the different maze configurations. The results highlight that Eik-QRL remains competitive even when its modeling assumptions are violated, while Eik-HiQRL is designed to better cope with such settings and indeed outperforms both QRL and Eik-QRL.

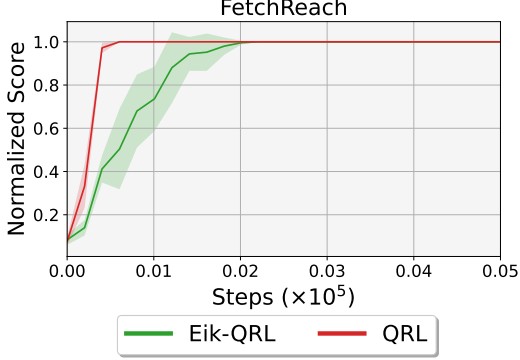

### H.6.2 FETCHREACH

We further evaluate QRL and Eik-QRL in an online GCRL setting on the `FetchReach` environment introduced in Plappert et al. (2018). The task is to move the end effector of a 7-DoF Fetch manipulator, equipped with a two-finger parallel gripper, to a randomly sampled target position within the robot workspace. The robot is controlled in Cartesian space: each action specifies a small displacement $(\Delta x, \Delta y, \Delta z)$ of

Figure 29: Learning curves for the online RL experiments on `FetchReach`. We evaluate each algorithm every 5k steps over 50 evaluation episodes and report mean and standard deviation over 5 random seeds.

the end effector together with a scalar command that opens or closes the gripper, while inverse kinematics are handled internally by the simulator. The resulting continuous dynamics are smooth and approximately isotropic in the control space, so both Assumption 4.2 (Lipschitz dynamics) and, consequently, Assumption 4.4 (Lipschitz goal-conditioned value function) are expected to hold reasonably well in this setting.

In Fig. 29, we report online training curves comparing QRL and Eik-QRL under identical architectures, hyperparameters, and training budgets.[2] Both methods rapidly achieve high success rates, and their convergence characteristics are very similar: the learning curves closely overlap, with only minor differences in transient behavior. These experiments show that, under modeling conditions that match its assumptions, Eik-QRL can be successfully extended to an online robotic control task with performance comparable to QRL.

---

[2]We follow the original QRL implementation provided at `https://github.com/quasimetric-learning/quasimetric-rl/`.

