# OpenReview forum: "Goal Reaching with Eikonal-Constrained Hierarchical Quasimetric Reinforcement Learning"
_ICLR.cc/2026/Conference — ICLR 2026 Poster_

### Official Review · Reviewer_nmvA · 2025-10-21

**Soundness:** 3
**Presentation:** 3
**Contribution:** 3
**Rating:** 6
**Confidence:** 3

**Summary:**

This paper proposes Eikonal-Constrained Quasimetric Reinforcement Learning (Eik-QRL), a novel goal-conditioned reinforcement learning framework that integrates constraints derived from partial differential equations (PDEs), along with its hierarchical extension, Eik-HiQRL. The method reformulates the discrete trajectory constraints in Quasimetric Reinforcement Learning (QRL) into continuous-time constraints governed by the Eikonal PDE, thereby providing a more theoretically grounded and smooth representation of distance-based objectives. The effectiveness of the proposed approach is empirically validated on the ogebnch benchmark.

**Strengths:**

1.The paper is well-written, with a clear motivation and a high level of completeness in both theoretical and experimental aspects.

2.The paper presents clear theoretical contributions with a rigorous derivation chain from QRL to HJB and finally to the Eikonal equation. Lemma 4.7 establishes the 1-Lipschitz property of the optimal value function, while Theorem 4.8 provides a high-probability value recovery guarantee, ensuring the theoretical soundness of the framework.

3.In terms of algorithmic design, the trajectory-free nature of the method represents a key advantage, particularly evident on stitching datasets, where traditional approaches rely heavily on trajectory reconstruction. The PDE constraint serves as an implicit regularizer, enhancing out-of-distribution estimation. Moreover, the hierarchical design is well-structured: the high-level abstraction operates in a simplified latent space to avoid complex dynamics, while the low-level temporal difference (TD) module compensates for the limitations of Eik-QRL.

4.The experiments are highly comprehensive, covering six categories of environments—PointMaze, AntMaze, HumanoidMaze, AntSoccer, and Manipulation tasks. The paper also introduces an innovative evaluation metric, the collision rate, which addresses a notable gap in the existing literature. Furthermore, extensive ablation studies are conducted, including five additional experiments in the appendix, providing thorough empirical validation of the proposed approach.

**Weaknesses:**

1.The assumptions are somewhat overly idealized—for instance, the requirement of unit-speed isotropic dynamics (i.e., f(s,a)=a) does not hold in real-world robotic systems, where manipulators typically exhibit complex, nonlinear dynamics. This limits the direct applicability of the theoretical model to practical robotic control scenarios.

2.On the manipulation tasks in the ogebnch benchmark, the proposed method does not outperform previous state-of-the-art algorithms, indicating that its advantages may be less pronounced in environments requiring fine-grained control and complex dynamics.

3.Although Eik-QRL represents a practical compromise that replaces full quasimetric projection with PDE-based constraints for handling high-dimensional and complex dynamical systems, the authors do not provide results on high-dimensional visual environments. As a result, the generality and robustness of their claims remain uncertain without further empirical validation in such settings.

4.The authors are encouraged to clearly specify the differences and relationships among Eik-QRL, QRL, HIQL, and the closely related Eik-HIQL, and to explicitly discuss the advantages and limitations of Eik-QRL relative to these methods. It is recommended to include a comparative table in the related work section to enhance clarity.

5.To the best of my knowledge, QRL has demonstrated experimental effectiveness in online settings; however, it remains unclear how Eik-QRL performs in online training scenarios.

6.Computational efficiency is not reported—for example, there is no comparison of training time or memory consumption. The PDE constraint requires automatic differentiation to compute gradients ‖∇_s d_θ(s,g)‖, which may be more expensive than discrete constraints. Appendix E only states “4 hours on RTX 3090” without providing comparisons to baseline methods.

**Questions:**

1.Regarding the fundamental issue of the isotropy assumption: Can the authors quantify how much the optimal value function deviates from the 1-Lipschitz property under realistic robotic dynamics? Can they provide a theoretical analysis or bound—even a relaxed one—for cases with non-isotropic dynamics? Since HJB-QRL is theoretically more general, why does it perform poorly in practice, and is there room for improvement?

2.Proposition B.1 — Hierarchical Analysis:The analysis is based on a 1D toy environment (Fig. 5). Can this be extended to higher-dimensional settings?How is the correlation coefficient ρ estimated or controlled in practical environments?

3.Why does the unconstrained version (Eq. 22) outperform the constrained version (Eq. 21)? Table 5 shows that Eik-QRL outperforms Eik-QRL-λ, suggesting that a soft penalty may be more effective than a hard constraint. Does this observation contradict the theoretical motivation of the method?

4.Figure 6 shows Quasimetric-H-actor has the tightest bound. Why should we use Eik-HiQRL instead?

5.The authors consider a continuous formulation of QRL. To my knowledge, a major limitation of QRL lies in its deterministic dynamics assumption. Could the authors clarify whether Eik-QRL is applicable under stochastic dynamics? If not, how could QRL be extended to handle stochastic environments?

6.Statistical significance: Many results exhibit large standard deviations (e.g., Table 1, pointmaze-giant-stitch: 62 ± 22). Were any statistical tests (such as a t-test) conducted to verify the significance of the reported improvements?

---

> ### Author Response · Authors · 2025-11-23
>
> We thank the reviewer for the careful reading of our paper and for the detailed, in-depth feedback.
>
> We have addressed the identified weaknesses one-by-one, answered all questions, and updated the manuscript accordingly. If our responses satisfactorily resolve your main concerns, we would be very grateful if you could consider updating your score. We remain available for any further clarifications or follow-up questions.
>
> ### W1
>
> > The assumptions are somewhat overly idealized. This limits the direct applicability of the theoretical model to practical robotic control scenarios.
>
> Thank you for your comment. We agree that some of our assumptions are strong from a practical perspective, as also highlighted in Remark 4.6 (lines 281-290), but they are needed to obtain a clean and rigorous theoretical analysis. Importantly, as discussed in lines 314-321, our approach can still be extended beyond this setting as long as the induced value function satisfies suitable Lipschitz properties.
>
> Regarding applicability to robot control, we would like to emphasize that our framework remains suitable even when the assumptions are not perfectly met in practice. This is supported by the results in Table 1, where Eik-QRL performs well on the *antmaze* benchmark. Moreover, in the revised version of the manuscript **we have added an additional experiment with a robot arm, where Eik-QRL and QRL achieve comparable performance (Appendix H.6)**.
>
> ### W2
>
> > On the manipulation tasks in the ogbench benchmark, the proposed method does not outperform previous state-of-the-art algorithms, indicating that its advantages may be less pronounced in environments requiring fine-grained control and complex dynamics.
>
> Thank you for this observation. We agree that on the OGbench manipulation tasks our approach does not outperform all prior methods, and we explicitly do not claim universal gains in such settings. However, the main limitation here is not fine-grained control or the dynamics per se: as long as the induced value function remains (approximately) Lipschitz, our Eikonal regularization remains appropriate (Lines 314-321).
>
> Furthermore, as discussed in lines 485-520, these manipulation tasks involve rich object interactions that can induce discontinuities in the value function, making the Eikonal PDE a less suitable inductive bias. In this sense, the empirical results are consistent with our theoretical discussion of when Eik-QRL is expected to provide the largest benefits.
>
> ### W3
>
> > The authors do not provide results on high-dimensional visual environments. As a result, the generality and robustness of their claims remain uncertain without further empirical validation in such settings.
>
> Thank you for raising this point. We would like to clarify that we do not claim generality or robustness to pixel-based environments in this work. In fact, such environments typically do not satisfy the regularity assumptions introduced in Section 4. As a result, applying our framework in pixel-based settings would require an additional, and likely non-trivial, representation learning step. We see the design of such representations as a compelling direction for future work.
>
> ### W4
>
> > The authors are encouraged to clearly specify the differences and relationships among Eik-QRL, QRL, HIQL, and the closely related Eik-HIQL, and to explicitly discuss the advantages and limitations of Eik-QRL relative to these methods. It is recommended to include a comparative table in the related work section to enhance clarity.
>
> Thank you for this suggestion.
>
> We agree that it is important to clearly spell out the relationships among Eik-QRL, QRL, HIQL, and Eik-HIQL. In the revised manuscript, **we have expanded the discussion of these relationships in Appendix F, where we also include a compact comparative table (Table 4)**. We chose to place this material in the appendix rather than the Related Work section because, by that point in the paper, the reader has already seen our formalism and is better positioned to fully appreciate the nuances between these methods.
>
> In the main text, we explicitly refer to Appendix F in Lines 463-464, immediately after introducing our Offline GCRL baselines.
>
> ### W5
> > QRL has demonstrated experimental effectiveness in online settings; however, it remains unclear how Eik-QRL performs in online training scenarios.
>
> Thank you for pointing this out. In the revised version of the manuscript, **we have added an additional comparison between Eik-QRL and QRL in an online robotic arm setting to directly evaluate the performance of Eik-QRL in online training scenarios (Appendix H.6)**.

---

> ### Author Response · Authors · 2025-11-23
>
> ### W6
> > Computational efficiency is not reported—for example, there is no comparison of training time or memory consumption. The PDE constraint requires automatic differentiation to compute gradients, which may be more expensive than discrete constraints.
>
> Thank you for raising this point. In practice, the additional computational overhead of the Eikonal term is minimal: it requires one extra per-sample backward pass, which can be fully parallelized on modern hardware. To enhance transparency, **we have added a quantitative analysis of training time in Appendix G, Lines 1378-1396**, in the revised version of the manuscript.
>
> ### Questions
> > Can the authors quantify how much the optimal value function deviates from the 1-Lipschitz property under realistic robotic dynamics?
>
> > Can they provide a theoretical analysis or bound for cases with non-isotropic dynamics?
>
> Thank you for these questions. In order to quantify how much the optimal value function deviates from the 1-Lipschitz property, one first needs a theoretical characterization of the dynamics that generate it.
>
> More precisely, answering your first question amounts to the following: given dynamics $f(s,a)$ (and a running cost), the optimal value function solves an associated Hamilton-Jacobi equation and induces a natural “control metric” on the state space. Our Eikonal regularizer, in contrast, enforces 1-Lipschitz continuity in the Euclidean metric. Quantifying the deviation from the 1-Lipschitz model can therefore be seen as studying the discrepancy between $V^*$ and its projection onto the 1-Lipschitz hypothesis class in this Euclidean metric.
>
> To derive explicit bounds, one would need to impose strong, system-specific assumptions on the dynamics $f(s,a)$ (and the induced Hamiltonian). For simple, structured systems (e.g., certain linear dynamics on a compact domain with regular costs), one could in principle relate the control metric to the Euclidean metric and obtain quantitative bounds on this discrepancy. However, for general nonlinear, contact-rich robotic dynamics this quickly becomes intractable and highly system-dependent.
>
> > Since HJB-QRL is theoretically more general, why does it perform poorly in practice, and is there room for improvement?
>
> Thank you for this question. Please also refer to our answer to **W3, Q3** above (**Reviewer oo25**). As we mention in lines 256-261, the inner product term in the HJB loss in Eq. (7) can become very small in high-dimensional spaces, which leads to vanishing gradients and unstable training. Similar numerical issues for HJB-based solvers have been documented in prior work [1,2].
>
> We believe there is room for improvement along at least two directions: (i) enforcing the HJB constraint in a learned latent space (see also **W3, Q3** to **Reviewer oo25**), and (ii) modifying the neural architecture used to parameterize the value function, for example along the lines of DeepReach [2].
>
> [1] Shilova, A., Delliaux, T., Preux, P., and Raffin, B., 2023. Revisiting continuous-time reinforcement learning: a study of HJB solvers based on PINNs and FEMs. EWRL 2023 Workshop.
>
> [2] Bansal, S., and Tomlin, C. J., 2021. DeepReach: A deep learning approach to high-dimensional reachability. ICRA.
>
> > Proposition B.1 — Hierarchical Analysis: The analysis is based on a 1D toy environment. Can this be extended to higher-dimensional settings?
>
> Thank you for this question. In principle, the argument in Proposition B.1 can be extended to higher-dimensional settings by replacing the 1D chain with a more general graph and carefully tracking how noise accumulates along multiple paths. However, this would make the analysis substantially harder, without changing the main qualitative message: hierarchy and quasimetric projection reduce the impact of noise on value comparisons and, consequently, the probability of policy error. Since Proposition B.1 is intended as a didactic example rather than a central theorem, we chose a 1D setting for clarity and tractability.
>
> > How is the correlation coefficient $\rho$ estimated or controlled in practical environments?
>
> Thank you for this question. In our analysis, $\rho$ is a modeling parameter and, similarly to $\sigma$ in (10) and (11), it is typically not estimated from data in practical environments. In Proposition B.1, we introduce $\rho$ for the sake of correctness, since in (11) the noise terms on neighboring segments are correlated after the quasimetric projection, and this correlation must be reflected in the variance of the value difference.
>
> However, for computing the bounds of the illustrative example in Figure 6 we simply fix $\rho$ = 0.01. Choosing a small correlation coefficient makes the bound more conservative, as it corresponds to a harder regime where the shared noise between neighboring estimates is limited and random fluctuations have a stronger impact on value comparisons. **We have clarified this aspect at the end of Appendix B (Lines 1115-1122)**.

---

> ### Author Response · Authors · 2025-11-23
>
> > Figure 6 shows Quasimetric-H-actor has the tightest bound. Why should we use Eik-HiQRL instead?
>
> Thank you for this question. We discuss this phenomenon in lines 1077-1114. Figure 6 reports the bound obtained in a 1D didactic setting, where the quasimetric projection can be computed almost exactly.
>
> However, in higher-dimensional state spaces quasimetric projections become substantially harder to obtain in practice: it can be shown that the approximation error of the quasimetric projection scales exponentially with the state dimension (Footnote in Line 377). This difficulty effectively translates into a larger noise level $\sigma$ in Eq. (11), which explains why purely quasimetric-based algorithms struggle in high-dimensional environments (see, e.g., QRL on the antmaze and humanoidmaze tasks in Table 6).
>
> The bound obtained by Eik-HiQRL matches that of Quasimetric-H-actor for suitable choices of the subgoal step $k$. Moreover, Eik-HiQRL shows better empirical performance in high-dimensional settings than purely quasimetric algorithms, as highlighted in Table 1.
>
> > Why does the unconstrained version (Eq. 22) outperform the constrained version (Eq. 21)? Table 5 shows that Eik-QRL outperforms Eik-QRL-λ, suggesting that a soft penalty may be more effective than a hard constraint. Does this observation contradict the theoretical motivation of the method?
>
> Thank you for bringing this up. Theorem 4.8 is explicitly designed to account for approximate enforcement of the Eikonal constraint: the result is stated in terms of an $\epsilon$ term that bounds the Eikonal residual. In other words, the theorem guarantees that if the constraint is satisfied up to an error $\epsilon$, then the learned value function is $\epsilon$-close to the ideal constrained solution. This already covers the soft-penalty case.
>
> Regarding the empirical finding that the “unconstrained’’ formulation in Eq.(22) (Eik-QRL) outperforms the more explicitly constrained variant in Eq.(21) (Eik-QRL-$\lambda$), this does not contradict the theoretical motivation. As illustrated in Table 5, Eik-QRL-$\lambda$:flat and Eik-QRL-$\lambda$ perform well in the isotropic *pointmaze* setting. The deterioration in the *antmaze* setting is in fact consistent with our theory, since the ant dynamics are not isotropic and thus less aligned with the Eikonal constraint. In addition, we observe higher sensitivity to hyperparameters and increased training instability for the Lagrangian variant. For these reasons, we adopt the simpler unconstrained Eik-QRL formulation in our extended set of experiments.
>
> > To my knowledge, a major limitation of QRL lies in its deterministic dynamics assumption. Could the authors clarify whether Eik-QRL is applicable under stochastic dynamics? If not, how could QRL be extended to handle stochastic environments?
>
> Thank you for this question. We have successfully deployed Eik-QRL in stochastic settings, and it is indeed applicable in practice. Please refer to the *teleport* setting in Table 1 for Eik-QRL and Table 6 for Eik-HiQRL.
>
> Similarly to QRL, we assume deterministic dynamics in the theory (Assumption 4.2) in order to simplify the analysis. Extending QRL and Eik-QRL to the stochastic setting is conceptually straightforward but technically involved: one would move from a deterministic shortest-path formulation to a stochastic control formulation, where $d(s,g)$ is defined via an expected hitting cost, and the PDE counterpart becomes a second-order HJB equation. We see developing such a stochastic extension as a natural but nontrivial direction for future work.
>
> > Statistical significance: Many results exhibit large standard deviations (e.g., Table 1, pointmaze-giant-stitch: 62 ± 22). Were any statistical tests (such as a t-test) conducted to verify the significance of the reported improvements?
>
> Thank you for raising this point. We respectfully disagree with the statement that *many* results exhibit large standard deviations: most entries in Table 1 have relatively small standard deviations, with higher variability appearing mainly in the *giant* maze settings, which are intrinsically harder.
>
> Moreover, Table 1 is not presented to claim improvements; rather, it is used to show that Eik-HiQRL addresses the limitations of Eik-QRL and is a strong candidate for thorough comparison against state-of-the-art Offline GCRL algorithms.
>
> For the settings where we do claim clear improvements over the baselines, we performed statistical significance testing using Welch’s t-test. In particular, for *Humanoidmaze-Large-Stitch* and *Humanoidmaze-Giant-Stitch*, Eik-HiQRL achieves statistically significant gains over the best-performing baseline, with $t = 11.7$, $p \approx 10^{-9}$ and $t = 22$, $p \approx 10^{-14}$, respectively. These values indicate that the reported improvements are highly statistically significant. **We have added these details in the revised manuscript for clarity (Lines 467-470)**.

---

> > ### Comment · Reviewer_nmvA · 2025-11-26
> >
> > Thank you for the clarification. I have two follow-up questions:
> >
> > 1. Given that your method is based on the assumption of deterministic dynamics, could you elaborate on why it still performs well in the teleport setting? A more detailed explanation of the underlying reasons would be appreciated.
> >
> > 2. Could the authors provide additional experimental results across more environments for the online setting? Based on the current results in the FetchReach environment alone, I do not observe a clear performance distinction between QRL and Eik-QRL. In other words, the advantages of Eik-QRL in the online setting remain unclear to me.

---

> ### Author Response · Authors · 2025-11-27
>
> Thank you for the follow-up questions.
>
> **Deterministic Assumption**: The deterministic dynamics assumption is introduced to obtain a clean and concise theoretical analysis and is shared with a large portion of the RL literature [1, 2, 3].
>
> In practice, our method performs reasonably well, but it is not optimal since it does not explicitly model stochastic dynamics. In other words, it exhibits *empirical robustness* to stochasticity. A plausible intuition is that, under such regimes, the learned value function approximates the expectation of the value under the stochastic dynamics, so the effect of randomness is largely averaged out during training. Guaranteeing that this behavior holds more generally is an interesting direction.
>
> **Online RL experiments**: **We have added 10 additional experiments in the online RL setting**, where the beneficial effects of our approach are clearly evident. Please refer to Appendix H.6.1 and Figure 28 in the updated version of the manuscript.
>
>
>
> **Bibliography**
>
> [1] Kumar, A., Peng, X.B. and Levine, S., 2019. Reward-conditioned policies. arXiv preprint arXiv:1912.13465.
>
> [2] Chen, L., Lu, K., Rajeswaran, A., Lee, K., Grover, A., Laskin, M., Abbeel, P., Srinivas, A. and Mordatch, I., 2021. Decision transformer: Reinforcement learning via sequence modeling. Advances in neural information processing systems, 34, pp.15084-15097.
>
> [3] Ghosh, D., Gupta, A., Reddy, A., Fu, J., Devin, C., Eysenbach, B. and Levine, S., 2019. Learning to reach goals via iterated supervised learning. arXiv preprint arXiv:1912.06088.

---

### Official Review · Reviewer_oo25 · 2025-10-31

**Soundness:** 2
**Presentation:** 2
**Contribution:** 2
**Rating:** 4
**Confidence:** 2

**Summary:**

This paper introduces Eikonal-Constrained Quasimetric RL (Eik-QRL), a novel approach to Goal-Conditioned Reinforcement Learning (GCRL). It builds upon Quasimetric RL (QRL), which frames GCRL as learning a quasimetric (a shortest-path distance function) $d(s, g)$. The key insight of this paper is to reformulate QRL's discrete, trajectory-based local consistency constraint (i.e., $d(s, s') \le \text{cost}$) into a continuous-time Partial Differential Equation (PDE) constraint.

By assuming simplified unit-speed, isotropic dynamics ($f(s,a) = a$), this PDE constraint reduces to the Eikonal equation: $||\nabla_s d_\theta(s, g)|| = 1$. This new formulation, Eik-QRL, is trajectory-free, meaning it only requires sampling states ($s$) and goals ($g$) rather than full state-action-next-state transitions. This makes it an effective regularizer and highly suitable for offline RL.

To address the limitations of the isotropic dynamics assumption in complex environments, the authors propose Eik-Hierarchical QRL (Eik-HiQRL). This method uses the efficient, trajectory-free Eik-QRL as a high-level planner to propose subgoals in a simple abstract space (e.g., $(x, y)$ coordinates) where the Eikonal assumption holds. A separate, standard low-level TD-learning policy is then trained to reach these subgoals in the full, complex state space.

**Strengths:**

1.  The core idea of connecting quasimetric learning's local consistency to the Eikonal PDE is a creative, insightful, and theoretically sound contribution.
2.  A major practical strength of Eik-QRL is that it is trajectory-free, only requiring i.i.d. state and goal samples. This makes it more data-efficient and better suited for offline learning from unstructured datasets than the original QRL, which requires transition tuples.
3.  The paper clearly identifies the main weakness of its own Eik-QRL formulation (the strong isotropic dynamics assumption) and proposes a very logical and effective solution: use Eik-QRL as a high-level planner in a simple abstract space where the assumption *does* hold, and use a standard model-free controller for the complex low-level dynamics.

**Weaknesses:**

1.  The paper's strongest results are in navigation tasks (pointmaze, antmaze, humanoidmaze). In the antsoccer and manipulation tasks (Table 2), the performance gains vanish, and Eik-HiQRL is only "comparable" to baselines. The paper acknowledges this but it suggests the method's applicability is currently best suited for tasks where a simple Cartesian abstract space is available.
2.  The success of Eik-HiQRL appears to be highly dependent on the choice of the high-level abstract space $\overline{\mathcal{S}}$, which must be one where the Eikonal (unit-speed) assumption holds. For navigation, this is intuitively the agent's $(x, y)$ coordinates. For manipulation, the paper states this is a "latent space learned end-to-end", but details on *how* this space is learned and *how* it is constrained to satisfy the Eikonal properties are not in the main paper. This choice seems critical and non-trivial for applying the method to new domains.
3.  The paper derives HJB-QRL (Eq. 7) as a general PDE constraint, but quickly simplifies it to Eik-QRL by assuming $f(s,a)=a$. The justification is that HJB-QRL is "ill-conditioned" and still relies on transitions. This simplification is what limits the method to isotropic dynamics. An alternative, underexplored direction would be to keep the HJB-QRL formulation and use a learned local dynamics model.

**Questions:**

1.  The performance gains are most pronounced in navigation tasks. In manipulation and antsoccer, where the abstract space $\overline{\mathcal{S}}$ is a "latent space learned end-to-end" (for manipulation), the method is only on par with baselines. Could you elaborate on how this latent space is learned? How do you ensure this learned space adheres to the 1-Lipschitz / unit-speed properties required for the Eik-QRL high-level planner to be effective?
2.  Following on from Q1: How sensitive is Eik-HiQRL to the quality of this abstract space? If the abstract space does not accurately reflect the geometry of the underlying state space (e.g., it thinks two states are close when they are separated by a wall), does the Eikonal constraint still provide a useful learning signal?
3.  The jump from HJB-QRL (Eq. 7) to Eik-QRL (Eq. 8) is a key step, motivated by numerical stability and removing the reliance on transition tuples. Did the authors experiment with a middle ground? For example, using the HJB-QRL formulation but with a *learned* local dynamics model to estimate $f(s, a)$ (or $s' - s$), rather than simplifying the dynamics to $f(s,a)=a$?
4.  In Appendix C.2, the paper discusses two optimization methods: the constrained Lagrangian (Eq. 21) and the soft penalty (Eq. 22), ultimately choosing the soft penalty for stability. Does using a soft penalty risk the Eikonal constraint not being fully enforced (i.e., $||\nabla_s d_\theta|| \neq 1$)? If so, how much does this "approximate" enforcement weaken the theoretical guarantees of Theorem 4.8, which rely on the constraint being met?

---

> ### Author Response · Authors · 2025-11-23
>
> We sincerely thank you for your thoughtful and constructive feedback, and for highlighting the main conceptual and practical strengths of our work. We greatly appreciate the time and care you dedicated to this review.
>
> We have carefully addressed all of your concerns in the rebuttal and revised the manuscript accordingly. We hope these changes resolve your remaining doubts and, if our responses are satisfactory, we would be very grateful if you could consider updating your overall score. We remain fully available for any further clarification or follow-up questions.
>
> ### W1
> > The method's applicability is currently best suited for tasks where a simple Cartesian abstract space is available.
>
> Thank you for this comment. We fully agree that our current formulation is most naturally applied when we can define an abstract state space in which the induced dynamics are approximately isotropic, such as the Cartesian space used in pointmaze in Table 1, where Eik-QRL outperforms QRL, and the abstract high-level space used in Eik-HiQRL.
>
> However, we would like to emphasize that, as long as the dynamics are Lipschitz continuous (a sufficient condition for a Lipschitz optimal value function), Eik-QRL remains a suitable approach. Please refer to the explanation in Lines 314-320, to Table 1 and Table 4 (*antmaze*, comparing Eik-QRL vs. QRL and QRL:hi), and to the new experiments in Appendix H.6 on online manipulation tasks, where an online version of Eik-QRL shows good empirical results.
>
> That said, our goal in this paper is not to claim universal applicability, but to show that when such an abstract space exists, enforcing this structure yields clear benefits for PDE-constrained quasimetric value learning. We see explicitly stating and analyzing these assumptions as an important part of responsible research practice, and we have been careful not to overclaim beyond this regime. Extending the approach to more complex or learned abstract spaces is an exciting direction for future work.
>
> ### W2, Q1
> > For manipulation, the paper states this is a "latent space learned end-to-end", but details on how this space is learned and how it is constrained to satisfy the Eikonal properties are not in the main paper. This choice seems critical and non-trivial for applying the method to new domains.
>
> > Could you elaborate on how this latent space is learned? How do you ensure this learned space adheres to the 1-Lipschitz / unit-speed properties required for the Eik-QRL high-level planner to be effective?
>
> Thank you for raising this point and for the question. In the manipulation experiments, the abstract latent space is parameterized by a multi-layer perceptron encoder $\nu: \mathcal{S} \to \mathcal{Z}$. This encoder is trained end-to-end as part of the objective in Eq. (8): we feed $\nu(s)$ and $\nu(g)$ to the quasimetric value network, and the gradients from both the Global Relationships loss and the Eikonal Local Relationships term in Eq. (8) are backpropagated through $\nu$. Importantly, we do not introduce any additional auxiliary loss to explicitly enforce our geometric assumptions in the embedding space. **We have added this explanation in the updated version of the paper (Lines 478-483)**.
>
> We agree with the reviewer that this design choice is critical and non-trivial for extending the method to new domains. We see this as an important direction for future work (Lines 531-538).

---

> ### Author Response · Authors · 2025-11-23
>
> ### W3, Q3
>
> > An alternative, underexplored direction would be to keep the HJB-QRL formulation and use a learned local dynamics model.
>
> > The jump from HJB-QRL (Eq. 7) to Eik-QRL (Eq. 8) is a key step, motivated by numerical stability and removing the reliance on transition tuples. Did the authors experiment with a middle ground? For example, using the HJB-QRL formulation but with a learned local dynamics model to estimate $f(s,a)$ or $s’-s$, rather than simplifying the dynamics to $f(s,a)=a$?
>
> Thank you for bringing this up. The main challenges with the HJB-QRL formulation are indeed numerical. As we mention in Lines 257-260, the inner product in the HJB loss in Eq. (7) can become very small in high-dimensional spaces, which leads to vanishing gradients and unstable training. Similar issues have been documented for HJB-based solvers in prior work [2,3]. We do not expect that replacing the true dynamics with a learned model would alleviate this problem, since the gradient term will still be evaluated in the same high-dimensional space and could even compound approximation errors from the learned model.
>
> An interesting approach would be to learn a lower-dimensional latent dynamics model and apply an HJB-style constraint in that latent space. However, how to design and train such a latent model so that it preserves the relevant reachability geometry is non-trivial and, in our view, deserves a dedicated study. For this reason, we chose to focus on the Eik-QRL formulation in Eq. (8), which is numerically more stable yielding better empirical performance. We consider the idea of latent-space HJB-QRL with learned dynamics as an important direction for future work.
>
> [2] Shilova, A., Delliaux, T., Preux, P., and Raffin, B., 2023. Revisiting continuous-time reinforcement learning: a study of HJB solvers based on PINNs and FEMs. EWRL 2023 Workshop.
>
> [3] Bansal, S., and Tomlin, C. J., 2021. DeepReach: A deep learning approach to high-dimensional reachability. ICRA.
>
> ### Q2
>
> > How sensitive is Eik-HiQRL to the quality of this abstract space?
>
> Thank you for this question. In our theoretical analysis, when the abstract space is isotropic and satisfies the assumptions of Eik-QRL, Eik-HiQRL inherits the guarantee of recovering the optimal goal-reaching value function. Beyond this ideal case, what matters is that the induced optimal value function in the abstract space remains reasonably well-behaved (for example, Lipschitz continuous, which holds when the dynamics $f(s,a)$ are Lipschitz or approximately Lipschitz).
>
> In other words, Eik-HiQRL does benefit from a reasonably chosen abstract space, but our experiments indicate that it is not overly fragile to moderate deviations from the ideal isotropic setting like in *antsoccer* in Table 2 for instance.
>
> > If the abstract space does not accurately reflect the geometry of the underlying state space (e.g., it thinks two states are close when they are separated by a wall), does the Eikonal constraint still provide a useful learning signal?
>
> Thank you for this question.
>
> In practice, the Eikonal constraint still provides a useful learning signal even when the abstract space does not perfectly match the true environment geometry. In Eq. (8), the Eikonal term is defined using the Euclidean norm of the gradient in the abstract coordinates. This should be interpreted as a *local* constraint: it enforces that the value function $d_\theta(s,g)$ changes at a controlled rate with respect to these coordinates, i.e., it promotes a 1-Lipschitz, distance-like behavior of $d_\theta(\cdot,g)$ in the abstract space.
>
> Importantly, in our implementation this constraint is only evaluated on states sampled from the *feasible* space, i.e., states visited in the dataset $\mathcal{D}$ where the agent can actually move. Thus, the Eikonal term shapes the value landscape locally along feasible directions of motion, without making any claim about states inside obstacles.
>
> Empirically, this interpretation is supported by the strong performance we observe across all mazes, including the most cluttered and long-horizon \texttt{giant} setting.

---

> ### Author Response · Authors · 2025-11-23
>
> ### Q4
> > Does using a soft penalty risk the Eikonal constraint not being fully enforced? If so, how much does this "approximate" enforcement weaken the theoretical guarantees of Theorem 4.8, which rely on the constraint being met?
>
> Thank you for this comment. Theorem 4.8 is explicitly designed to account for approximate enforcement of the constraint: the result is stated in terms of an $epsilon$ term that bounds the Eikonal residual. In other words, the theorem guarantees that if the Eikonal constraint is satisfied up to an error $\epsilon$, then the learned value function is correspondingly epsilon-close to the ideal solution.
>
> We also experimented with a more explicit Lagrangian dual formulation, where a learnable multiplier enforces the constraint more “hardly” (see Eq. (21) for this formulation and Appendix H.2 for the empirical results). As illustrated in Table 6, these variants perform very similarly in the isotropic *pointmaze* environment, while, the lagrangian dual formulation performs worse in the *antmaze* setting due to the fact that the ant dynamics is not actually isotropic. As a result, we adopt the simpler soft-penalty formulation in Eq. (22), which already fits naturally within the epsilon-approximate guarantees of Theorem 4.8.

---

### Official Review · Reviewer_9KNx · 2025-10-31

**Soundness:** 3
**Presentation:** 4
**Contribution:** 2
**Rating:** 6
**Confidence:** 2

**Summary:**

- This paper introduces Eikonal-Constrained Quasimetric RL (Eik-QRL), which uses a continuous Eikonal constraint instead of the standard local Quasimetric RL local constraint.
- Eik-QRL is derived and and formulated. This involves: (1) moving from QRL to a HJB-QRL continuous time variant, including bye assuming smooth dynamics and a smooth value function; (2) Moving from HJB-QRL to Eik-QRL, which is easier to optimize in practice, by assuming unit-speed, isotropic dynamics.
- Theoretical guarantees are then proved: Under unit-speed isotropic dynamics in a convex space, the optimal value function has unit gradient norm everywhere, and a universal approximator that satisfies the unit gradient constraint will recover the optimal value function.
- The paper notes the benefits of Eik-QRL include trajectory-free estimation and PDE-based regularization, at the cost of strong assumptions (including the 1-Lipschitz property) which may not hold in general MDPs
- Acknowleging the potential limitations in general MDPs, they propose a hierarchical approach that can use a high-level policy to exploit the benefits of Eik-QRL in a “a lightweight, dynamics-agnostic state-space”, while the low-level policy uses standard TD-learning that does not have restrictive assumptions.
- Their methods are empirically tested in the offline GCRL setting. In the simple pointmaze environment, they show improvements in collisions, in more complex antmaze, they show Eik-QRL is on par with QRL despite Eikonal constraints, and show EikHiQRL achieves the strongest results. In humanoidmaze with more complex dynamics still, they also outperform baselines
- In environments where 3rd party objects and categorical variables mean regularity assumptions do not hold, they observe comparable performance to baselines. They identify designing PDE-constrained algorithms for these non-regular, non-isotropic robotic settings as follow up direction.
- Overall, the paper contributes novel theory which enables a practical method that performs very well in certain settings

**Strengths:**

- The theory and formulation of Eik-QRL is novel and is an interesting perspective on QRL.
- The clarity and writing in the paper is generally very good. The theory is well written, the writing is well-balanced, limitations are well-acknowledged.
- While Eik-QRL does make strong assumptions, it seems plausible these assumptions could hold to some extent in certain scenarios.
- Eik-QRL often shows strong performance on benchmarks versus baselines, including state-of-the art humanoid maze performance. This seems to establish Eik-HiQRL as a SOTA method in the navigation benchmarks.
- Empirically, performance is reasonable in settings where the strong assumptions do not hold. This seems to mitigate some concerns regarding the limitations of the restrictive assumptions of Eik-QRL.

**Weaknesses:**

**Assumptions of Eik-QRL may limit real-world use-cases.**

The authors have well-acknowledged the limitations of some assumptions made by Eik-QRL. My general concern is that these assumptions could limit the significance and wider impact of this method. While the paper has evaluated their method on a wide range of environments, including some where their assumptions break down, many of these environments seem to be navigation-based - it seems plausible the method is overfitting to these specific toy navigation benchmarks which may not be representative of many realistic settings.

**Experiments and results could be performed and presented in a more systematic manner to help us better understand the strengths and weaknesses of Eik-QRL**

I would like to see improved benchmarking and systematic experiments that better decompose and provide evidence for exactly where and how  Eik-QRL adds value and where and how it struggles. This could add more clarity on the practical value of the method. I will give some examples below.

The authors state three main benefits that Eik-QRL provides. They state trajectory-free as a strong advantage in the introduction, but I cannot see any clear experiments validating this. While they do have evidence of the “improved state coverage” benefit, this evidence is hard to parse in the results.

The baseline methods they compare against all seem to use slightly different components, making it hard to fairly compare and hard to determine what are the core valuable contributions of Eik-HiQRL (is the performance gain simply due to the hierarchy, or is it due to the Eik method? Is Eik particularly suited to a high-level abstract-space, versus standard TD methods or standard QRL?). Ideally, we could easily compare Eik-QRL and Eik-HiQRL to an equivalent QRL and HiQRL to better distinguish where the value add is. If Eik-HiQRL is proposed as their SOTA method, it could be useful to better compare the Eik high-level policy to other standard high-level policies, while keeping the low-level policy method constant.

In general, presenting aggregated may make it easier to parse overall performance differences between methods.

Perhaps a more thorough and systematic approach could be taken to “red-teaming” the Eik method to finding and understand cases where the assumptions are limiting in practice. This could involve: (i) adding another non-regular environment such as CALVIN, or (ii) better determining the extent to which Eik performance gains are mainly due to the ‘navigation’ component in the navigation environments.

**Other**

After strong theory to introduce Eik-QRL, there is minimal theory or discussion to properly justify the choice to use Eik-HiQRL.

**Questions:**

- Could you explain why exactly PDE acts as an implicit regularizer?
- Could you propose an experiment to demonstrate why “trajectory-free” could be useful in practice?
- How correlated is the success rate metric with the collision avoidance metric? Is the collision avoidance metric providing much extra information?
- Perhaps you could better highlight your strong performance improvements over QRL in the challenging and non-regular environments?
- Could you provide more environment details in the appendix? Ideally including details regarding the extent to which the Eik assumptions hold in each environment.

---

> ### Author Response · Authors · 2025-11-23
>
> We thank the reviewer for carefully reading our work and for the thoughtful comments.
>
> We have addressed each point in detail and clarified the issues you raised. If our responses alleviate your concerns, we would be grateful if you could consider updating your score accordingly. We are, of course, happy to provide any further clarification or follow-up during the discussion phase.
>
>
> ### W1
> > The authors have well-acknowledged the limitations of some assumptions made by Eik-QRL. My general concern is that these assumptions could limit the significance and wider impact of this method.
>
> Thank you for raising this point. As discussed in our response to **Q2** for **Reviewer dSDx**, the assumptions underlying Eik-QRL are satisfied or well-approximated in a number of practically relevant settings, and we view them as concrete guidelines for when this formulation is appropriate in practice.
>
> We fully agree that practical applicability and broader impact are essential. This is precisely the motivation behind our hierarchical extension Eik-HiQRL.
>
> More broadly, we see our analysis, including stating the needed assumptions, not as a limitation, but as a *roadmap for future work*. By making explicit which geometric and regularity properties are beneficial for PDE-based value learning, our results suggest a promising direction: developing representation learning strategies that preserve effective control behavior while promoting value functions with the structure required by our theory (e.g., approximate Eikonal-like properties). We believe this kind of theory-driven representation learning is a compelling path to further bridge the gap between formal guarantees and real-world deployment. **We have added this perspective in the Conclusion of the revised version of the paper (Lines 531-538).**
>
> > It seems plausible the method is overfitting toy navigation benchmarks which may not be representative of many realistic settings.
>
> We thank the reviewer for raising this concern. We respectfully disagree, with the characterization of our evaluation domains as “toy navigation benchmarks”.
>
> OGbench [1] was designed to stress-test Offline GCRL with diverse maze geometries, datasets, and dynamics, including high-dimensional continuous-control systems and long horizons. Even within the navigation category, the problem is full GCRL rather than single-goal RL: the agent must answer arbitrary state-goal queries (including goals and start states not present in the dataset) and thus perform out-of-distribution generalization over the joint $(s,g)$ space.
>
> On complex tasks such as *humanoid-giant-stitch* in **Fig. 4**, as well as across the broader suite in **Table 6**, our hierarchical variant Eik-HiQRL yields substantial gains over strong baselines that often fail or degrade. This suggests that our algorithm makes tangible progress over state-of-the-art methods toward closing the gap to more realistic goal-conditioned RL settings.
>
> [1] Park, S., Frans, K., Eysenbach, B. and Levine, S., OGBench: Benchmarking Offline Goal-Conditioned RL. In The Thirteenth International Conference on Learning Representations.
>
> ----------------------------------------------------------------------------------------------------------------------------
> ### W2: Experiments in a more systematic manner + Q2
> > Trajectory-free as a strong advantage in the introduction, but I cannot see any clear experiments validating this.
>
> > Could you propose an experiment to demonstrate why “trajectory-free” could be useful in practice?
>
> Thank you for pointing this out.
>
> Referring to our Eik-QRL formulation in Eq. (8), the benefits of being “trajectory free” are primarily structural, which is why we did not originally design experiments targeting this aspect in isolation. In the regimes where this property is most relevant (Lines 270-277), one can sample states from the workspace of the agent (for example from a map or a geometric model) but does not yet have trajectories. In that case, Eik-QRL can already be trained, since its objective only requires state-goal samples, whereas standard Offline GCRL methods cannot even be instantiated because they rely on transition tuples $(s, s')$ (see Eq. (1)).
>
> That said, we agree that additional empirical results help illustrate this advantage more concretely. In the revised version, **we have added a new experiment on a robotic goal-reaching task (FetchReach in [2]) that explicitly targets the fully trajectory-free regime (See Appendix H.5 and Fig. 27)**. We view this as an interesting demonstration of the practical potential of trajectory-free value learning and leave a more exhaustive study to future work.
>
> [2] Plappert, M., Andrychowicz, M., Ray, A., McGrew, B., Baker, B., Powell, G., Schneider, J., Tobin, J., Chociej, M., Welinder, P. and Kumar, V., 2018. Multi-goal reinforcement learning: Challenging robotics environments and request for research. arXiv preprint arXiv:1802.09464.

---

> ### Author Response · Authors · 2025-11-23
>
> > It is hard to fairly compare and hard to determine what are the core valuable contributions of Eik-HiQRL.
>
> Thank you for this comment.
>
> We respectfully disagree with the statement that it is hard to identify the core contributions of Eik-HiQRL, as we believe that the information and experimental evidence needed to assess these contributions are already present in the paper. At the same time, we agree that in the original submission the narrative roadmap guiding the reader through these elements was not sufficiently explicit; **we have now added a clearer roadmap in Lines 391-403 to address this**.
>
> In detail, Table 1 already provides a compact comparison between Eik-QRL in (8), HJB-QRL in (7), and QRL in (1). This comparison showcases the pros and cons of introducing PDE-based constraints into quasimetric learning, since all three methods share the same value-learning design (up to their local constraints). The choice of using a hierarchical actor is discussed in Lines 416-420. As correctly noted by the reviewer, it is important to disentangle the effect of PDE constraints from the actor choice; to address this, we refer to Appendix H.1 and Table 5.
>
> Once the pros and cons of Eik-QRL are analyzed against these baselines, Table 1 also reports results for our hierarchical variant Eik-HiQRL. These results show that the proposed design overcomes the limitations of Eik-QRL on the *antmaze* suite, which motivates the subsequent comparison with other state-of-the-art baselines.
>
> **Appendix H** contains additional experiments and ablations, and **Section H.4** in particular studies the impact of PDE constraints within our hierarchical structure (Eik-HiQRL vs. HiQRL), with results summarized in Table 8. We hope that, together with the new roadmap added in the main text, this clarifies how the contributions of the PDE-based constraints and the hierarchical design are isolated and evaluated.
>
> > Perhaps a more thorough and systematic approach could be taken to “red-teaming” the Eik method to finding and understanding cases where the assumptions are limiting in practice.
>
> Thank you for pointing this out. We agree that illustrating scenarios where the assumptions become limiting in practice is important for a complete discussion. This is precisely the role of the experiments on *antmaze* in Table 1 for Eik-QRL and of the non regular environments in Table 2 for Eik-HiQRL, where contact rich dynamics and additional objects in the state space violate our assumptions.
>
> **We have expanded the discussion around these results in the revised version to make this aspect more explicit and to provide additional insight into the practical limitations of our assumptions (Lines 429-455 and 515-520)**.
>
> ------------------------------------------------------------------------------------------------------------------------------------------------------------
> ### Other
>
> > There is minimal theory or discussion to properly justify the choice to use Eik-HiQRL.
>
> We agree with the reviewer that the motivation for Eik-HiQRL should be discussed more clearly, and **we have expanded this aspect in the revised version (Section 5, Lines 343-366)**.
>
> In particular, we now provide a more thorough explanation in the main text of why a hierarchical structure becomes natural in our setting, how Eik-QRL induces a value function that is well suited for high level planning, and why separating high level goal selection from low level control is beneficial in long horizon GCRL.
>
> We also refer the reviewer to **Appendix B**, where we include further discussion and analysis to complement the main paper.
>
> ---------------------------------------------------------------------------------------------------------------------------------------------------------------
> ### Questions
> > Could you explain why exactly PDE acts as an implicit regularizer?
>
> We provide an explanation on this effect in **Remark 4.5, lines 277-281**. For $\mathcal S\subset\mathbb R^k$, each sampled pair $(s,g)$ contributes a full gradient vector $\nabla_s d_{\theta}(s,g)\in\mathbb R^k$, thereby coupling all coordinate directions at $s$. Aggregating such gradients across diverse pairs promotes global consistency. Please also refer to the following references:
>
> Lien, Y.H., Hsieh, P.C., Li, T.M. and Wang, Y.S., 2024, July. Enhancing value function estimation through first-order state-action dynamics in offline reinforcement learning. In Forty-first International Conference on Machine Learning.
>
> Giammarino, V., Ni, R. and Qureshi, A.H., Physics-informed Value Learner for Offline Goal-Conditioned Reinforcement Learning. In The Thirty-ninth Annual Conference on Neural Information Processing Systems.

---

> ### Author Response · Authors · 2025-11-23
>
> > How correlated is the success rate metric with the collision avoidance metric? Is the collision avoidance metric providing much extra information?
>
> Thank you for this question. We do observe a correlation between success rate and collision avoidance, as described in Lines 423-428. In particular, for QRL in Table 1, high collision counts are associated with degraded performance in larger environments.
>
> Table 4 provides similar insights. Even when QRL is equipped with a hierarchical actor, it exhibits more collisions in the *pointmaze-giant* setting, for instance, and this coincides with a performance drop compared to our methods. This shows that the collision metric is not merely redundant with success, but highlights how success is achieved. Two agents can have comparable success on a subset of tasks while behaving very differently in terms of safety and contact with obstacles.
>
> We find this interesting because collision avoidance is often overlooked in the RL literature. Reporting it provides complementary information and an additional axis along which we can evaluate and compare different solutions.
>
> > Could you provide more environment details in the appendix? Ideally including details regarding the extent to which the Eik assumptions hold in each environment.
>
> We have added more details on this aspect in **Appendix E** in our revised version.

---

### Official Review · Reviewer_dSDx · 2025-11-01

**Soundness:** 4
**Presentation:** 4
**Contribution:** 3
**Rating:** 8
**Confidence:** 3

**Summary:**

The paper considers the problem of goal conditioned quasimetric reinforcement learning (GCQRL). The novel contribution of the paper is to introduce a (Eikonal) PDE constrained formulation. This Eikonal constrained formulation can be viewed as a refinement of the preexisting HJB PDE constrained GCQRL problem, derived by assuming the continuous-time system dynamics to be isotropic. Although this dynamics assumption is simplistic the formulation trades off this with a very desirable trajectory-free nature, requiring only samples from state, goal distributions. The paper establishes that under some Lipschitz dynamics, cost and value function assumptions, for compact state-action spaces, it is possible for a universal quasimetric approximator to approximate the optimal value function under this Eikonal-constrained formulation. Further, to alleviate issues rising from the complex dynamics that most RL problems have and break the assumptions of the Eikonal formulation, the authors propose a hierarchical version of the Eikonal constrained formulation, where a Eikonal GCQRL operates on an abstract (higher level) state space, and a more traditional TD style RL algorithm operates in the actual (lower level) state space. The authors complement their theoretical contributions with empirical evidence demonstrating that their Eikonal formulation consistently outperforms other QRL methods on a variety of robotics tasks.

**Strengths:**

1. The presented formulation is trajectory-free and only requires one to sample (state, goals) rather than complete trajectory rollouts which I really appreciate.
2. The core part of the presented Eikonal approach boils down to a constrained optimization problem which can be readily solved through a wide suite of existing physics informed ML methods.
3. The experiments in the paper specifically highlight that the formulation performs well in settings where no theoretical statements can be made currently (i.e., complex, non-Lipschitz dynamics).

**Weaknesses:**

1. The main weakness is the relatively simplistic dynamics assumption of the formulation (i.e., Lipschitzness, and that the continuous time counterpart is unit speed isotropic). However, I feel that this weakness is adequately acknowledged and addressed by the authors.
2. One of the main assumptions for Lemma 4.7 and Theorem 4.8, stated in line 301 says $c(s,g)=1$ on $\mathcal{K}\setminus g$. Does this mean $g$ can be the only suitable goal in $\mathcal{K}$? If so, I feel that is a limitation as for continuous control tasks, several points in a continuous neighborhood can be also be goal states reaching which leads to success for an RL policy.

**Questions:**

Besides point 2. in the weakness section, please address:
1. The running cost $c(s,a)$ is introduced in the usual state-action sense in Section 3, but is used in the state-goal sense $c(s,g)$ in Section 4. Could you define what $c(s,g)$ means more clearly like you did for the rewards in goal conditioned RL?
2. Could you give some intuition on whether (local) Lipschitzness of the optimal value function in Assumption 4.4 is a reasonable assumption in practice or not?
3. (Minor formatting point) In table 1, in the row ant maze navigate large, the algorithm giving the lowest $\kappa$ is not colored.

---

> ### Author Response · Authors · 2025-11-23
>
> We want to thank the reviewer for having carefully reviewed our work and the positive feedback. Please find our answers below. We hope we were able to further clarify your doubts and questions and we remain available for any additional follow-ups. Thank you again for your help in improving our work.
>
> **W1**
> > The main weakness is the relatively simplistic dynamics assumption of the formulation. However, I feel that this weakness is adequately acknowledged and addressed by the authors.
>
> We want to thank the reviewer for this comment. Our goal was indeed to propose a structurally and theoretically novel algorithm while being explicit about the required assumptions and the practical trade offs they introduce. We are glad that our discussion of these limitations and how we address them was clear, and we view relaxing these assumptions as an important direction for future work.
>
> **W2**
> > The assumption stated in line 301 says: $c(s,g) = 1$ on $\mathcal{K} \setminus \{g\}$. Does this mean $g$ can be the only suitable goal in $\mathcal{K}$? If so, I feel this is a limitation.
>
> The assumption $c(s,g) = 1$ on $\mathcal{K} \setminus \{g\}$ (and $c(g,g) = 0$) is inherited from the standard GCRL / shortest-path formulation in lines 137-141: it encodes a unit running cost until the system reaches the goal associated with $g$, where the cost becomes zero. In other words, for a trajectory $(s_t)_{t \ge 0}$ targeting $g$, we have a running cost $c(s_t, g) = 1$ that is independent of the action and satisfies $c(s_T, g) = 0$ once $s_T = g$.
>
> Moreover, the statement of **Lemma 4.7** (see also Lemma A.5, lines 829-843) extends directly to the case where the goal is not a single state but a goal set, e.g., a ball $B_{\varepsilon}(g)$ around the nominal goal. This is exactly how GCRL is typically implemented in practice, where any state in a small neighborhood of $g$ is treated as successful. Replacing the singleton $\{g\}$ with such a goal set leaves the proof unchanged.
>
> **Q1**
> > Could you define what $c(s,g)$ means more clearly?
>
> In our setting, the cost function is $c : \mathcal{S} \times \mathcal{G} \to \mathbb{R}_{\geq 0}$, where $c(s,g)$ denotes the instantaneous running cost when the agent is in state $s$ while pursuing goal $g$. As mentioned above, we use a unit running cost away from the goal, that is, $c(s,g) = 0$ if $s = g$ and $c(s,g) = 1$ if $s \neq g$, for all $g \in \mathcal{G}$.
>
> We have clarified the definition of the running cost in **Assumption 4.3** (lines 206-210) in the revised version of the paper (highlighted in red).
>
> **Q2**
> > Could you give some intuition on whether (local) Lipschitzness of the optimal value function in Assumption 4.4 is a reasonable assumption in practice or not?
>
> Thank you for this question.
> When the state $s$ encodes smooth robot configurations (e.g., joint positions and velocities) and the dynamics are governed by smooth ODEs or their time-discretized versions, the optimal cost-to-go $V^\star(s,g)$ behaves like a shortest-path function on a compact set $\mathcal{K}$: small perturbations of $(s,g)$ induce only small changes in the minimal path length. In continuous-control tasks such as navigation in free space or reaching with smooth dynamics, this kind of local Lipschitzness is therefore a mild assumption.
>
> However, the assumption becomes more fragile in contact-rich settings where the state explicitly includes contact modes and external objects, as in our *AntSoccer* and *manipulation* experiments (see Table 2). Contacts and discontinuous mode switches can introduce sharp transitions in $V^\star$, so local Lipschitzness may only hold approximately, or may fail altogether near contact boundaries.
>
> In our benchmarks, this means that the assumption is well justified for smooth tasks such as *pointmaze*, and plausibly holds approximately for locomotion tasks such as *antmaze* and *humanoidmaze*, whereas it is generally violated in contact-rich domains such as *AntSoccer* and the *manipulation* environments.
>
> We now make this distinction explicit in **Appendix E**, where we detail, for each environment, to what extent Assumptions 4.2 and 4.4 are expected to hold or fail in practice.
>
> **Minor**
> > Formatting issue in Table 1.
>
> Thank you for bringing this up. We have corrected Table 1 in the updated version of the paper.

---

### Author Response · Authors · 2025-11-23

We thank the Area Chair and the Reviewers for reading our paper and for the constructive feedback.

We have carefully gone through all the raised points and provide concise, point-by-point responses in the rebuttal. In parallel, we have revised the manuscript to incorporate new experiments and expanded discussion, with **all changes highlighted in red** in the updated version.

Below we summarize the main updates to the manuscript (line numbers refer to the revised version):

1. Clarified the definition of the running cost in Assumption 4.3 (Lines 206-210).  [**Reviewer dSDx**, **Q1**]
2. Strengthened the Conclusion to more clearly highlight the value and implications of our analysis (Lines 531-538). [**Reviewer 9KNx**, **W1**]
3. Added new experiments illustrating the potential of the trajectory-free formulation (Appendix H.5 and Fig. 27). [**Reviewer 9KNx**, **W2 + Q2**]
4. Added a short roadmap of the experiments and ablations to make the contributions of Eik-HiQRL easier to parse (Lines 391-403). [**Reviewer 9KNx**, **W2 + Q2**]
5. Expanded the discussion of when our assumptions may be limiting in practice (Lines 429-455 and 515-520). [**Reviewer 9KNx**, **W2 + Q2**]
6. Expanded the section on the introduction of hierarchy to clarify the design and role of Eik-HiQRL (Section 5, Lines 343-366). [**Reviewer 9KNx**, **Other**]
7. Added more details on the tested environments including whether they comply or not with our Assumptions (Appendix E). [**Reviewer 9KNx**, **Q3**]
8. Included new experiments on online RL (Appendix H.6). [**Reviewer nmvA**, **W5 + Follow up Question**]
9. Added more details on how we encode the states to learn the embedding in the manipulation experiments (Lines 478-483). [**Reviewer oo25**, **W2 + Q1**]
10. Added a table comparing Eik-HiQRL against the relevant baselines (Appendix F). [**Reviewer nmvA**, **W4**]
11. Added an analysis of training time for Eik-QRL vs. QRL (Appendix G, Lines 1378-1396). [**Reviewer nmvA**, **W6**]
12. Clarified the role of $\rho$ in Appendix B (Lines 1115-1122). [**Reviewer nmvA**, **Q2**]
13. Added details on the Welch’s t-test for the humanoidmaze results of Eik-HiQRL (Lines 467-470). [**Reviewer nmvA**, **Q6**]

We are grateful for the opportunity to improve the quality and clarity of our work. We hope that our responses and the revised manuscript satisfactorily address the main concerns and we remain happy to provide any additional clarification if needed.

---

### Author Response · Authors · 2025-11-27

We would like to once again sincerely thank all reviewers for their time, effort, and constructive feedback throughout the review process.

As the discussion period approaches its end, we kindly invite you to consider whether our rebuttal and the updated version of the manuscript adequately address your concerns and, if you feel it is appropriate, to update your scores accordingly.

We remain available and happy to answer any further questions or provide additional clarifications.

---

### Meta-Review · Area_Chair_xJLC · 2025-12-29

**Summary:**

The main concern brought up in the reviews was that the method seems restricted to 2D navigation tasks. The OGBench experiments seem to underscore this point: the settings where the proposed method outperforms baselines are ones that most closely resemble 2D navigation. That said, the authors argued in their response that "theory-driven representation learning is a compelling path to further bridge the gap between formal guarantees and real-world deployment", and have added a discussion on this point to the appendix. Overall, as the AC, I do think this is an important limitation of the paper, but am wary of penalizing the authors for stating their assumptions more clearly than other papers. The paper appears to provide a new theoretical perspective on an important problem; none of the reviewers raised concerns about the correctness of the Eikonal perspective or its novelty.

Overall, I think that RL researchers (especially those with an algorithmic/theoretical bent) will enjoy learning about this new perspective. I therefore recommend that the paper be accepted.

**Reviewer Concerns:**

(see below)

**Reviewer Scores:**

dSDx: 8 --> 8
* [/] relatively simplistic dynamics assumption of the formulation: The original paper already acknowledges these limitations.
* [+] how are goals defined in continuous spaces? Authors clarified that the proof also holds for an epsilon-Ball formulation.

9KNx: 6 --> 6
* [+] Assumptions of Eik-QRL may limit real-world use-cases ... Finding and understand cases where the assumptions are limiting in practice: Authors argue that the assumptions are "satisfied or well-approximated in a number of practically relevant settings." They also argue that "theory-driven representation learning is a compelling path to further bridge the gap between formal guarantees and real-world deployment", and have added a discussion on this point to the appendix. As the AC, I find this argument fairly compelling.
* [+] exactly where and how Eik-QRL adds value and where and how it struggles
* [+] evidence that the method is "trajectory free"
 * [+] when comparing with baselines, using as many shared components as possible
* [-] are the gains limited to navigation tasks? The authors argue that benchmark has many different maze geometries, dynamics, datasets, and dimensionality. As the AC, I don't find this argument particularly compelling: most of the tasks where the proposed method outperforms prior work are effectively navigation tasks. There are some tasks used that aren't navigation tasks (e.g., antsoccer and manipulation in Table 2), however for those tasks it's unclear if the proposed method provides any benefits relative to the HIQL baseline.**

**  If the benefits of quasimetrics and/or the proposed method (Eik-QRL) are limited to 2D navigation tasks, I think that could actually be an interesting discovery. It currently is unclear in the literature how and when dynamic programming is necessary, and this paper could help shed some light on that.

oo25: 4 --> 4
* [/] method's applicability is currently best suited for tasks where a simple Cartesian abstract space is available. Authors acknowledged this limitation, but argue that the Lipschitz assumption is slightly weaker than assuming a Cartesian abstract space.
* [+] how this space is learned and how it is constrained to satisfy the Eikonal properties are not in the main paper. Authors clarified this and revised the paper.
* [/] keep the HJB-QRL formulation and use a learned local dynamics model. Authors discussed this point but didn't run experiments on it.

---

### Decision · Program_Chairs · 2026-01-26

Accept (Poster)